# When and How Human Curation Backfires: Preference Alignment under Multi-Model Self-Consuming Loop

Yang Zhang [1]   Xiukun Wei [1]   Xueru Zhang [1]

## Abstract

Foundation models are increasingly trained on synthetic data generated by prior model iterations rather than exclusively on real data. This *self-consuming* training paradigm can lead to model collapse, divergence, or bias amplification. Recent work (Ferbach et al., 2024) shows that incorporating human curation into the loop can steer a self-consuming model toward human-aligned behavior, but these analyses focus on a single, *isolated* model that solely consumes its own outputs. In practice, however, models often interact and train on input–output pairs produced by other models. This paper studies self-consuming training in the *multi-model* regime. We first formalize a framework for interacting self-consuming models and characterize when the resulting dynamical system converges to a stable point. We then examine how human curation of one model affects its own alignment (self-influence) and how such effects propagate to other models (cross-influence). Unlike isolated settings where human curation always enhances model alignment, we show that cross-model interactions can dampen or even invert this effect, ultimately degrading long-term alignment.

## 1. Introduction

Modern foundation models are no longer just consumers of data; they have become large-scale producers whose outputs circulate throughout the broader data ecosystem, appearing on the web, entering annotation pipelines, and being incorporated into downstream finetuning datasets. As a result, synthetic (often curated) data is routinely mixed with real data during training. This creates a *self-consuming loop* in which models are iteratively updated on synthetic data generated by earlier model iterations. For example, in language, synthetic instruction data generated by one LLM is used to finetune another model, reducing reliance on costly human annotation (Taori et al., 2023). In vision, diffusion models generate labeled images that are merged into large-scale datasets for training or finetuning recognition models (Shumailov et al., 2024). In multimodal settings, auto-captioning models produce synthetic text that accompanies scraped images, and these image–caption pairs are later reused when training image–text models at scale (Yu et al., 2024).

Motivated by this, a growing body of work has examined how models evolve under self-consuming training loops (Shumailov et al., 2024; Gao & Li, 2025; Bertrand et al., 2024; Cai et al., 2026). Prior work has shown, both theoretically and empirically, that when model-generated content is reused to train successive models, such loops can induce degenerative dynamics and lead to unintended outcomes, including model collapse (Shumailov et al., 2024; Bertrand et al., 2024; Fu et al., 2024), divergence (Shumailov et al., 2024), and bias amplification (Wang et al., 2026; Wyllie et al., 2024; Xie & Zhang, 2024). Recent works (Ferbach et al., 2024; Zhao et al., 2025) further explore the role of human curation in this process and show that when synthetic data is curated by humans according to their preferences at each iteration before being incorporated into the training set, models trained under self-consuming loops gradually align with human preferences and their outputs can converge to a distribution that maximizes human expected reward. Conversely, when human curation is noisy or adversarial, this alignment process may be disrupted (Wei & Zhang, 2025).

However, existing work on self-consuming models is largely limited to a single, **isolated** model that consumes only its own outputs, whereas real-world data ecosystems are inherently **multi-model** (Pressman et al., 2022). In web-scale corpora (e.g., LAION-5B (Schuhmann et al., 2022)), model-generated data can be scraped and incorporated into future training pipelines, effectively coupling different models through recycled data (Alemohammad et al., 2023). In such systems, updating one model reshapes the data distribution used to train others, which may then feed back into the first, forming a network of implicit interactions.

[1]Department of Computer Science and Engineering, The Ohio State University, Columbus, Ohio. Correspondence to: Xueru Zhang <zhang.12807@osu.edu>.

*Proceedings of the 43rd International Conference on Machine Learning*, Seoul, South Korea. PMLR 306, 2026. Copyright 2026 by the author(s).

To the best of our knowledge, only a few studies consider multi-model self-consuming systems (Hu et al., 2025; Gao & Li, 2025). However, these works are either empirical—highlighting new collapse patterns under multi-model recursive reuse (Hu et al., 2025)—or theoretically tractable only under highly simplified distributional assumptions and update rules (Gao & Li, 2025). As a result, we still lack a unified theoretical framework to address several practically motivated questions: How does a multi-model ecosystem evolve under iterative self-consuming loops? Under what conditions does such a system converge to a stable state? How does human curation applied to one model influence its own alignment and that of other models in the long run?

In this paper, we study a multi-model setting in which models iteratively update on mixtures of real data and synthetic data generated and curated by themselves and by other models. We first formalize an analytical framework for interacting self-consuming models and characterize conditions under which the system converges to a stable point. Given convergence, we then analyze the impact of human curation on the system's long-term alignment. Specifically, we conduct local sensitivity analyses to quantify how increasing the fraction of human-curated synthetic data for one model affects its own alignment (self-influence) and how these effects propagate to other models (cross-influence). Our results highlight a key difference from the single-model setting: whereas increasing human curation consistently improves long-term alignment in isolated self-consuming loops (Ferbach et al., 2024), its effect becomes non-monotonic in the multi-model regime. Cross-model interactions may amplify, dampen, or even reverse the impact of human curation, and our analysis identifies conditions under which human curation can negatively affect alignment in the long run. Beyond these local effects, we also quantify the global alignment gap by comparing mixed real–synthetic training to purely real-data training. We discuss more related works in Appendix A and summarize key contributions as follows:

- We formalize a framework for interacting multi-model self-consuming systems, which captures a wide range of cross-model interactions and incorporates human preference feedback into the self-consuming loop, without relying on stylized distributional assumptions (Section 2).

- We provide a rigorous analysis to identify conditions under which the multi-model system converges to a stable point, and examine how the real-data ratio contributes to stability under cross-model interactions (Section 3).

- We quantify the long-term impact of human curation on the model alignment. The results are interpretable and decompose the effects into *self-influence* (the model's direct response to curation) and *cross-influence* (the influence propagated through cross-model interactions). This allows us to identify conditions under which human curation improves, or degrades, long-term alignment (Section 4).

- We validate our theoretical findings through real experiments, demonstrating that increasing human curation does not necessarily improve model alignment and the non-monotonic behavior aligns with theorems (Section 5).

## 2. Problem Formulation

Consider a multi-model ecosystem in which each model is iteratively updated on a mixture of real data and synthetic data generated by itself or by other models. For simplicity, we focus on two models that, at each round $t$, are parametrized by $\theta_t \in \Theta$ and $\phi_t \in \Phi$ respectively, where both $\Theta, \Phi$ are closed convex sets. Let $x \in \mathcal{X}$ and $y \in \mathcal{Y}$ denote the outputs of models $\theta_t$ and $\phi_t$, respectively. We assume that the input (resp. output) data type of $\theta_t$ matches the output (resp. input) data type of $\phi_t$. Under this assumption, the corresponding conditional data distributions are defined as

$$x \sim p_{\theta_t}(x|y) \ \text{ and } \ y \sim q_{\phi_t}(y|x)$$

This formulation captures a wide range of cross-model interactions, including interactions among generative models of the same modality (e.g., text models whose outputs are reused as training data for other text models), interactions across different modalities (e.g., image-to-text and text-to-image models that feed into one another), and interactions between generative and discriminative models (e.g., classifiers that predict category labels and generative models that produce images or text conditioned on those categories).

**Sources of training data.** To maintain strong performance and adapt to evolving environments, both models $\theta_t$ and $\phi_t$ are repeatedly updated using their most recently acquired training datasets, $\mathcal{D}_{t+1}^{\theta}$ and $\mathcal{D}_{t+1}^{\phi}$, respectively. These datasets are typically drawn from multiple sources:

1. *Real data* $\mathcal{R}^{\theta}, \mathcal{R}^{\phi}$ whose distributions are fixed.

2. *Self-consuming synthetic data* $\mathcal{S}_{t+1}^{\theta}, \mathcal{S}_{t+1}^{\phi}$, consisting of samples generated either by the model itself or by other models. Let $P_{t+1}^{x,y}$ and $Q_{t+1}^{x,y}$ denote the distributions over data pairs $(x, y)$ used to obtain the updated models $\theta_{t+1}$, $\phi_{t+1}$ from $\theta_t$ and $\phi_t$, respectively. Outputs generated by the models at $t$ may be incorporated into the training data of future model generations. In particular, model $\theta_t$ can generate synthetic samples of the form

$$\{(x', y') : y' \sim P_t^y, x' \sim p_{\theta_t}(x|y')\}, \qquad (1)$$

while model $\phi_t$ can generate samples

$$\{(x', y') : x' \sim Q_t^x, y' \sim q_{\phi_t}(y|x')\}, \qquad (2)$$

where $P_t^y$ and $Q_t^x$ denote the corresponding marginals of $P_t^{x,y}$ and $Q_t^{x,y}$. To capture both self-model and cross-model influence, we allow $\mathcal{S}_{t+1}^{\theta}, \mathcal{S}_{t+1}^{\phi}$ to include samples generated by either model, with cross-model-generated

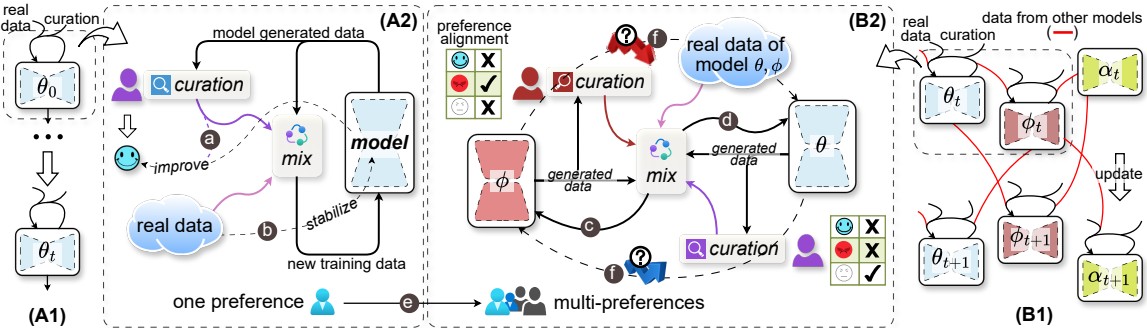

*Figure 1.* (A1) and (A2) illustrate the framework of a *single* self-consuming model and how to generate mixture data in each round. Real data helps to stabilize model updating (ⓑ) (Bertrand et al., 2024), and human curation benefits the reward improving (ⓐ) (Ferbach et al., 2024). (B1) presents our multi-model interaction framework. Different models may be updated in different orders. (B2) details the one-round iteration between two models. Under model interaction, multiple preferences exist within the system (ⓔ), and one model may use data generated/curated by other models for further training (ⓒⓓ), thus affecting its own preference alignment. It remains unclear how interactions affect each model under multi-model self-consuming learning (ⓕ).

fractions $\lambda_\theta^\phi$ and $\lambda_\phi^\theta$, respectively, and self-generated fractions $1 - \lambda_\theta^\phi$ and $1 - \lambda_\phi^\theta$, respectively.

3. *Human-curated synthetic data $\mathcal{H}_{t+1}^\theta, \mathcal{H}_{t+1}^\phi$.* To align models with human preferences, synthetic data is often refined using human feedback and curated by users based on their preferences. This process is commonly modeled using the generalized Bradley-Terry model (Bradley & Terry, 1952; Luce et al., 1959) or Direct Preference Optimization (Rafailov et al., 2023). For example, given $K$ samples $\mathcal{O} = \{o_1, \cdots, o_K\}$, $o_k \sim p_{\theta_t}(x|i)$ (or $p_{\phi_t}(y|i)$) generated by model $\theta_t$ (or $\phi_t$) for a given input $i \sim P_t^y$ (or $Q_t^x$), the probability that a sample $\hat{o} \in \mathcal{O}$ is selected by the user under generalized Bradley-Terry model is:

$$\mathbb{P}(\hat{o} = o_s|i, \mathcal{O}) = \frac{e^{r(i,o_s)}}{\sum_{k=1}^K e^{r(i,o_k)}} \qquad (3)$$

where $r$ is the underlying reward function capturing the user's preference for one sample over another. The curated datasets $\mathcal{H}_{t+1}^\theta, \mathcal{H}_{t+1}^\phi$ also contain samples curated from either model, and we assume that their cross-model data fractions match those of $\mathcal{S}_{t+1}^\theta, \mathcal{S}_{t+1}^\phi$.

For each model $j \in \{\theta, \phi\}$, the distribution underlying the training data $\mathcal{D}_{t+1}^j$ is a mixture over $\mathcal{R}_{t+1}^j, \mathcal{S}_{t+1}^j, \mathcal{H}_{t+1}^j$. The corresponding **mixing weights** are denoted by $\lambda_\mathcal{R}^j, \lambda_\mathcal{S}^j, \lambda_\mathcal{H}^j$, respectively, and satisfy $\lambda_\mathcal{R}^j + \lambda_\mathcal{S}^j + \lambda_\mathcal{H}^j = 1$.

**Iterative training loop.** Since both self-consuming and human-curated synthetic data depend on $\theta, \phi$, we denote the resulting training data distributions by $P_{t+1}^{x,y} := P(\theta, \phi)$ and $Q_{t+1}^{x,y} := Q(\theta, \phi)$ for $\mathcal{D}_{t+1}^\theta$ and $\mathcal{D}_{t+1}^\phi$, respectively, to explicitly emphasize this dependency. Depending on whether the model updates occur synchronously or asynchronously (Fig. 1(B1)), we consider the following update regimes:

1. **Synchronous updates**, where both models $\theta_t$ and $\phi_t$ are updated simultaneously. In this case, $P_{t+1}^{x,y} := P(\theta_t, \phi_t)$ and $Q_{t+1}^{x,y} := Q(\theta_t, \phi_t)$.

2. **Asynchronous updates**, where the update of one model (e.g., $\theta_{t+1}$) occurs before the other (e.g., $\phi_{t+1}$). In this setting, the self-consuming synthetic data of the later-updated model (e.g., $\mathcal{S}_{t+1}^\phi$) includes samples generated by the already-updated model (e.g., $\theta_{t+1}$). As a result, $P_{t+1}^{x,y} := P(\theta_t, \phi_t)$ and $Q_{t+1}^{x,y} := Q(\theta_{t+1}, \phi_t)$.

The detailed sampling process is presented in Algorithm 1. Given the new training datasets, both models are updated to minimize the expected loss under certain loss functions.

$$\theta_{t+1} = \arg\min_\theta \mathbb{E}_{(x,y) \sim P_{t+1}^{x,y}}[\ell_\theta(x, y)];$$
$$\phi_{t+1} = \arg\min_\phi \mathbb{E}_{(x,y) \sim Q_{t+1}^{x,y}}[\ell_\phi(x, y)]. \qquad (4)$$

**Objectives.** Recent studies have analyzed the evolution of self-consuming generative models and the role of human curation in steering their behavior (Ferbach et al., 2024; Wei & Zhang, 2025; Zhao et al., 2025). However, prior work primarily focuses on a single, isolated model. In multi-model settings, it remains unclear how such an ecosystem evolves and how human curation at one model influences the overall system and its alignment in the long run.

In this paper, we address these questions. We first analyze the evolution of the self-consuming multi-model ecosystem introduced in this section, identifying conditions for the convergence and stability of the system (Section 3). We then examine the impact of multi-model interactions and human curation on convergence, with a particular focus on preference alignment (Section 4).

## 3. Evolution of Multi-model Ecosystem

Next, we investigate how self-consuming iterative training and cross-model interactions drive the evolution of the multi-

model ecosystem. Our goal is to characterize the conditions under which this ecosystem stabilizes and converges. We first formalize the notion of stability for an evolving system.

**Definition 3.1** (Stability). The multi-model ecosystem reaches a stable point $(\theta^*, \phi^*)$ if the following holds.

$$
\begin{aligned}
\theta^* &= \arg\min_{\theta} \mathbb{E}_{(x,y)\sim P(\theta^*,\phi^*)}[\ell_\theta(x,y)]; \\
\phi^* &= \arg\min_{\phi} \mathbb{E}_{(x,y)\sim Q(\theta^*,\phi^*)}[\ell_\phi(x,y)].
\end{aligned}
\tag{5}
$$

At $(\theta^*, \phi^*)$, both models and the self-consuming synthetic data they induce no longer change. Next, we examine the evolution of $(\theta_t, \phi_t)$ and identify conditions for the existence of a stable point and for convergence of the system.

**Assumption 3.2** (Strong convexity). Loss function $\ell_\theta$ is $\gamma_\theta$-strongly convex in $\theta$: for any $\theta, \theta'$, we have

$$
\ell_\theta(\theta) \geq \ell_\theta(\theta') + \nabla_\theta \ell_\theta(\theta')^T(\theta - \theta') + \frac{\gamma_\theta}{2}||\theta - \theta'||^2 \tag{6}
$$

Similarly, $\ell_\phi$ is $\gamma_\phi$-strongly convex in $\phi$.

**Assumption 3.3** (Smoothness). $\ell_\theta(x,y)$ is $L_\theta$-smooth in $(x,y)$ and $\theta$: for any $(x,y), (x',y'), \theta, \theta'$, we have

$$
\begin{aligned}
||\nabla_\theta \ell_\theta(x,y) - \nabla_\theta \ell_\theta(x',y')|| &\leq L_\theta||(x,y) - (x',y')||, \\
||\nabla_\theta \ell_\theta(\theta) - \nabla_\theta \ell_\theta(\theta')|| &\leq L_\theta||\theta - \theta'||.
\end{aligned}
\tag{7}
$$

Similarly, $\ell_\phi(x,y)$ is $L_\phi$-smooth in $(x,y)$ and $\phi$.

Note that Assumptions 3.2 and 3.3 are standard and widely used in the literature on analyzing self-consuming models (Bertrand et al., 2024; Ferbach et al., 2024) and performative prediction (Perdomo et al., 2020). These assumptions are sufficient to derive theorems, and empirical results in Section 5 suggest that the findings continue to hold even when these assumptions are relaxed or violated.

**Proposition 3.4** (Convergence of multi-model system). *Suppose Assumptions 3.2 and 3.3 hold, and data spaces $\mathcal{X}, \mathcal{Y}$ are bounded. If both models are Lipschitz in their inputs and parameters $\theta, \phi$, and reward functions are Lipschitz in inputs, then there exists $\tau \in (0,1)$ such that if the fraction of real data in each round of model training is sufficiently large, i.e., $\min(\lambda_\mathcal{R}^\theta, \lambda_\mathcal{R}^\phi) > \tau$, then a unique stable point $(\theta^*, \phi^*)$ exists. Moreover, the iterative training loop (4) will drive $(\theta_t, \phi_t)$ to converge to $(\theta^*, \phi^*)$, and the training data distributions will also converge, i.e., $\lim_{t\to\infty} P_t^{x,y} = P(\theta^*, \phi^*)$, $\lim_{t\to\infty} Q_t^{x,y} = Q(\theta^*, \phi^*)$.*

The convergence of a single, isolated self-consuming model has been established by Bertrand et al. (2024). Proposition 3.4 implies that cross-model interactions will not disrupt such convergence, provided that a sufficient amount of real data is included in each round of model updates. The precise quantity of $\tau$ is given in Appendix F.8. Indeed, the fractions of real data, $\lambda_\mathcal{R}^\theta$ and $\lambda_\mathcal{R}^\phi$, directly affect a property of this

dynamic system known as *distribution sensitivity*. Rather than imposing a restriction on $\lambda_\mathcal{R}^\theta$ and $\lambda_\mathcal{R}^\phi$, we can instead formulate an alternative condition in terms of distribution sensitivity that also guarantees the stability and convergence.

**Assumption 3.5** (Distribution sensitivity). The distribution $P(\theta, \phi)$ is $\varepsilon_\theta$-sensitive: for any $(\theta, \phi), (\theta', \phi')$, we have

$$
W(P(\theta, \phi), P(\theta', \phi')) \leq \varepsilon_\theta||(\theta, \phi) - (\theta', \phi')||.
$$

where $W(\cdot, \cdot)$ is the Wasserstein distance. Similarly, $Q(\theta, \beta)$ is $\varepsilon_\phi$-sensitive.

**Theorem 3.6** (Convergence of multi-model system). *Define*

$$
\kappa \triangleq \max\left(\frac{\gamma_\theta L_\phi \varepsilon_\phi + 2\gamma_\phi L_\theta \varepsilon_\theta}{\gamma_\phi(\gamma_\theta - L_\theta \varepsilon_\theta)}, \frac{\gamma_\phi L_\theta \varepsilon_\theta + 2\gamma_\theta L_\phi \varepsilon_\phi}{\gamma_\theta(\gamma_\phi - L_\phi \varepsilon_\phi)}, \frac{L_\theta \varepsilon_\theta}{\gamma_\theta} + \frac{L_\phi \varepsilon_\phi}{\gamma_\phi}\right).
$$

*Under Assumptions 3.2, 3.3, and 3.5, if $\kappa < 1$, then the stable point $(\theta^*, \phi^*)$ exists, and iterative training loop (4) will drive $(\theta_t, \phi_t)$ to converge to $(\theta^*, \phi^*)$ at a linear rate:*

$$
||(\theta_t, \phi_t) - (\theta^*, \phi^*)|| \leq \kappa^t ||(\theta_0, \phi_0) - (\theta^*, \phi^*)||.
$$

In Theorem 3.6, we consider all possible update orderings of $\theta_t$ and $\phi_t$ in each round, including both synchronous and asynchronous updates. The form of $\kappa$, taking the maximum over three terms, ensures the convergence regardless of the update scheme. We can relate Proposition 3.4 to Theorem 3.6: larger amounts of real data (higher fractions $\lambda_\mathcal{R}^\theta$ and $\lambda_\mathcal{R}^\phi$) lead to less sensitive distributions (smaller $\varepsilon_\theta$ and $\varepsilon_\phi$) and, consequently, a smaller $\kappa$.

# 4. Impact of Human Curation

Given the convergence to stable point $(\theta^*, \phi^*)$, we next examine the role of human curation in shaping the system's behavior. For a single, isolated model, Ferbach et al. (2024) showed that when synthetic data is curated by humans according to their preferences in each round, the model's outputs can gradually align with human preferences, eventually converging to a distribution that maximizes user reward. However, in a multi-model ecosystem, it remains unclear how alignment is influenced by curation. Through inter-model interactions, the influence of human curation applied to one model can propagate to others in complex and nontrivial ways, potentially distorting or even reversing the intended effects of human curation on model alignment.

In this section, we systematically examine this by analyzing how varying the fraction of human-curated synthetic data for one model at each round affects the multi-model system in the long run. For models $\theta_t$ and $\phi_t$, we define the *alignment* as

$$
\begin{aligned}
J_p(\theta_t, \mathcal{E}_\theta) &:= \mathbb{E}_{y\sim\mathcal{E}_\theta, x\sim p_{\theta_t}(x|y)}[r_\theta(x,y)], \\
J_q(\phi_t, \mathcal{E}_\phi) &:= \mathbb{E}_{x\sim\mathcal{E}_\phi, y\sim q_{\phi_t}(y|x)}[r_\phi(x,y)].
\end{aligned}
\tag{8}
$$

where reward functions $r_\theta(x,y), r_\phi(x,y)$ quantify how well the input-output pair $(x,y)$ is aligned with users of models $\theta$

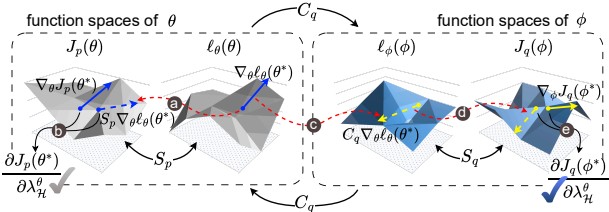

*Figure 2.* When varying only the curation strength of model $\theta$ ($\lambda_{\mathcal{H}}^{\theta}$), the $\theta$ model's update direction ($\nabla_\theta \ell_\theta$) is mapped into the reward alignment space via $S_p$ (**a**). The projected vector, together with $\nabla_\theta J_p$, characterizes the self-influence (**b**). In addition, $\nabla_\theta \ell_\theta$ is first mapped into the cross-model parameter space via $C_q$ (**c**), and then transformed by $S_q$ (**d**); combined with $\nabla_\phi J_q$, it determines the cross-influence (**e**).

and $\phi$, respectively. Importantly, $\mathcal{E}_\theta$ and $\mathcal{E}_\phi$ are generic *evaluation* distributions (e.g., test benchmarks or downstream task datasets) that may or may not match the training distributions $P_t^y$ or $Q_t^x$; this reflects standard practice in which alignment is assessed on datasets that may differ from the training environment. In our analysis, we are interested in the alignment of the models at the stable point: *How varying fraction of human-curated samples $\lambda_{\mathcal{H}}^{\theta}$ (or $\lambda_{\mathcal{H}}^{\phi}$) of one model in each round could affect alignment of its own (**self-influence**) and the other model (**cross-influence**) at the stable point, i.e., both $J_p(\theta^*)$ and $J_q(\phi^*)$.*

Because the interactions between $\phi_t$ and $\theta_t$ are symmetric, it suffices to analyze $\frac{\partial J_p(\theta^*)}{\partial \lambda_{\mathcal{H}}^{\theta}}$ (self-influence) and $\frac{\partial J_q(\phi^*)}{\partial \lambda_{\mathcal{H}}^{\theta}}$ (cross-influence). Since $\lambda_{\mathcal{H}}^{\theta}$ may vary for many reasons (e.g., changes in the curated or real-data pools), we adopt a simplified but controlled setting in which variations in $\lambda_{\mathcal{H}}^{\theta}$ arise solely from changes in the size of the curated dataset $\mathcal{H}^{\theta}$, while all other data sources remain fixed. Other cases are discussed in Appendix E. Our goal is to identify conditions under which increasing human-curation leads to improved or degraded model alignment at the stable point.

## 4.1. Quantifying Curation Impact on $J_p(\theta^*)$ and $J_q(\phi^*)$

We first quantify self-influence $\frac{\partial J_p(\theta^*)}{\partial \lambda_{\mathcal{H}}^{\theta}}$ and cross-influence $\frac{\partial J_q(\phi^*)}{\partial \lambda_{\mathcal{H}}^{\theta}}$. Since $J_q(\phi^*)$ and $J_p(\theta^*)$ are expectations measuring the generalization performance of models, the fraction of human-curated data $\lambda_{\mathcal{H}}^{\theta}$ affects $J_q(\phi^*)$ (resp. $J_p(\theta^*)$) through $\phi^*$ (resp. $\theta^*$). By the chain rule, we have

$$\frac{\partial J_p(\theta^*)}{\partial \lambda_{\mathcal{H}}^{\theta}} = \frac{\partial J_p(\theta^*)}{\partial \theta^*} \cdot \frac{\partial \theta^*}{\partial \lambda_{\mathcal{H}}^{\theta}}; \quad \frac{\partial J_q(\phi^*)}{\partial \lambda_{\mathcal{H}}^{\theta}} = \frac{\partial J_q(\phi^*)}{\partial \phi^*} \cdot \frac{\partial \phi^*}{\partial \lambda_{\mathcal{H}}^{\theta}}.$$

Note that both $\frac{\partial J_p(\theta^*)}{\partial \theta^*}$ and $\frac{\partial J_q(\phi^*)}{\partial \phi^*}$ depend only on the reward functions and the model structures. The main challenge, however, is computing $\frac{\partial \theta^*}{\partial \lambda_{\mathcal{H}}^{\theta}}$ and $\frac{\partial \phi^*}{\partial \lambda_{\mathcal{H}}^{\theta}}$, which describe how the stable point $(\theta^*, \phi^*)$ responds to changes in the fraction of human-curated data.

Define the gradients of objective functions in (5) as

$$F_p(\theta, \phi, \lambda_{\mathcal{H}}^{\theta}) \triangleq \mathbb{E}_{(x,y)\sim P(\theta,\phi)}[\nabla_\theta \ell_\theta(x, y)];$$
$$F_q(\theta, \phi, \lambda_{\mathcal{H}}^{\theta}) \triangleq \mathbb{E}_{(x,y)\sim Q(\theta,\phi)}[\nabla_\phi \ell_\phi(x, y)].$$

Note that $\theta$ and $\phi$ alone cannot fully capture the influence of $\lambda_{\mathcal{H}}^{\theta}$ on $F_q, F_q$, because $\lambda_{\mathcal{H}}^{\theta}$ also determines the mixing weights in the mixture distributions $Q(\theta, \phi)$ and $P(\theta, \phi)$.

For any value of $\lambda_{\mathcal{H}}^{\theta}$, the resulting stable point $(\theta^*, \phi^*)$ minimizes the objectives in (5). Thus, the following holds

$$\forall \lambda_{\mathcal{H}}^{\theta}, \ F_p(\theta^*, \phi^*, \lambda_{\mathcal{H}}^{\theta}) = F_q(\theta^*, \phi^*, \lambda_{\mathcal{H}}^{\theta}) = 0,$$

which yields the following proposition.

**Proposition 4.1** (Characterization of $\frac{\partial \theta^*}{\partial \lambda_{\mathcal{H}}^{\theta}}$ and $\frac{\partial \phi^*}{\partial \lambda_{\mathcal{H}}^{\theta}}$)**.**

$$\frac{\partial \theta^*}{\partial \lambda_{\mathcal{H}}^{\theta}} = -S_p \left( \frac{\partial F_p}{\partial \lambda_{\mathcal{H}}^{\theta}} + C_p \frac{\partial F_q}{\partial \lambda_{\mathcal{H}}^{\theta}} \right)$$
$$\frac{\partial \phi^*}{\partial \lambda_{\mathcal{H}}^{\theta}} = -S_q \left( C_q \frac{\partial F_p}{\partial \lambda_{\mathcal{H}}^{\theta}} + \frac{\partial F_q}{\partial \lambda_{\mathcal{H}}^{\theta}} \right)$$

Here, $S_p, S_q$ are matrices characterizing the **local sensitivity** of each model's stable point to perturbations, after accounting for the coupled dependence on the other model. The matrices $C_p, C_q$ quantify **cross-model influence**, capturing how changes in one model's objective induce parameter shifts in the other model through the coupled training dynamics. We formally define these matrices below.

**Definition 4.2** (Sensitivity matrix)**.** For $F \in \{F_p, F_q\}$, let the Jacobian matrices of the gradient mappings $F$ evaluated at the stable point $(\theta^*, \phi^*)$ be

$$\nabla_\theta \mathbf{F} := \nabla_\theta F(\theta^*, \phi^*, \lambda_{\mathcal{H}}^{\theta}); \ \ \nabla_\phi \mathbf{F} := \nabla_\phi F(\theta^*, \phi^*, \lambda_{\mathcal{H}}^{\theta}).$$

The *local sensitivity* matrices, which capture how each model's parameters respond to perturbations while accounting for cross-model interactions, are defined as

$$S_p := \left( \nabla_\theta \mathbf{F}_p - \nabla_\phi \mathbf{F}_p (\nabla_\phi \mathbf{F}_q)^{-1} \nabla_\theta \mathbf{F}_q \right)^{-1},$$
$$S_q := \left( \nabla_\phi \mathbf{F}_q - \nabla_\theta \mathbf{F}_q (\nabla_\theta \mathbf{F}_p)^{-1} \nabla_\phi \mathbf{F}_p \right)^{-1}. \quad (9)$$

**Definition 4.3** (Cross-model influence matrix)**.** Define

$$C_p := -\nabla_\phi \mathbf{F}_p (\nabla_\phi \mathbf{F}_q)^{-1}, \ \ C_q := -\nabla_\theta \mathbf{F}_q (\nabla_\theta \mathbf{F}_p)^{-1}.$$

which quantify how changes in one model's parameters propagate to the other model.

If there is no cross-model interaction (i.e., each model's input does not include data generated by the other model), the system reduces to a single-model scenario and $\lambda_\theta^\phi = \lambda_\phi^\theta = 0$. In this case, the matrices simplify to $S_q = (\nabla_\phi \mathbf{F}_q)^{-1}$, $S_p = (\nabla_\theta \mathbf{F}_p)^{-1}$, and $C_q = C_p = \mathbf{0}$.

**Proposition 4.4.** *Under Assumptions 3.2-3.5 and condition in Theorem 3.6, $\nabla_\theta \mathbf{F}_p$, $\nabla_\phi \mathbf{F}_q$, $S_p$ and $S_q$ are all invertible.*

Combining the above, we can quantify the influence of human-curated data on the alignment of the multi-model system at the stable point, as shown in Theorem 4.5.

**Theorem 4.5** (Self/cross-influence quantification). *Under Assumptions 3.2-3.5 and condition in Theorem 3.6, the self-consuming multi-model system converges. Let $P_\mathcal{H}$ be the distribution of human-curated data $\mathcal{H}_t^\theta$ after convergence.*

$$\frac{\partial J_p(\theta^*)}{\partial \lambda_\mathcal{H}^\theta} = \frac{1}{1 - \lambda_\mathcal{H}^\theta} \langle \nabla_\theta J_p(\theta^*), S_p \mathbb{E}_{P_\mathcal{H}}[-\nabla_\theta \ell_\theta(\theta^*)] \rangle,$$

$$\frac{\partial J_q(\phi^*)}{\partial \lambda_\mathcal{H}^\theta} = \frac{1}{1 - \lambda_\mathcal{H}^\theta} \langle \nabla_\phi J_q(\phi^*), S_q C_q \mathbb{E}_{P_\mathcal{H}}[-\nabla_\theta \ell_\theta(\theta^*)] \rangle.$$

*where $S_p$, $S_q$ are the sensitivity matrices, $C_q$ is the cross-model influence matrix, and $\langle \cdot, \cdot \rangle$ denotes the inner product between two vectors.*

In Theorem 4.5, $\mathbb{E}_{P_\mathcal{H}}[-\nabla_\theta \ell_\theta(\theta^*)]$ represents the direction in which $\theta$ would be updated if updated using only human-curated data. $\nabla_\theta J_p(\theta^*)$ is the direction in parameter space $\Theta$ that most increases the alignment. We can measure the alignment of these two directions using cosine similarity.

$$\rho_p \triangleq \cos\left(\nabla_\theta J_p(\theta^*), \mathbb{E}_{P_\mathcal{H}}[-\nabla_\theta \ell_\theta(\theta^*)]\right) \in [-1, 1], \quad (10)$$

Intuitively, if these two directions are aligned, i.e., $\rho_p > 0$, then the update induced by human-curated data coincides with the direction that improves the alignment metric. One might therefore expect that increasing the fraction of human-curated data would always enhance alignment. This intuition is indeed correct in the single-model setting when curation directly optimizes the alignment objective, in which case $\rho_p \approx 1$; see Corollary E.1 in Section E.1.

However, in a multi-model ecosystem, this intuition does not necessarily hold, for two distinct reasons. First, models can interact whenever they are trained on data of the same modality, even if the curation data distribution does not overlap with the evaluation distribution that defines $J_p$ and $J_q$. We refer to this phenomenon as **preference domain mismatch** (PDM). Under PDM, model interactions can substantially alter parameters without inducing corresponding changes in $J_p$ and $J_q$; see Section D.2 for further discussion.

Second, even without PDM, model interactions can introduce conflicting curation signals: $\mathcal{H}_t^\theta$ may include cross-model data whose induced update direction is misaligned with the alignment objective, resulting in $\rho_p < 0$. Moreover, Theorem 4.5 shows that cross-model interactions and self-consuming dynamics can further distort the impact of human curation through the sensitivity matrix $S_p$. In Section 4.2, we illustrate that even when $\rho_p > 0$, human curation can sometimes reduce alignment across the ecosystem.

## 4.2. Distortion Effect of Sensitivity Matrix

Next, we illustrate how $S_p$ can distort the impact of human-curated data. In particular, we provide an example showing that even when $\rho_p > 0$, this distortion can result in $\frac{\partial J_p(\theta^*)}{\partial \lambda_\mathcal{H}^\theta} < 0$, meaning that human curation intended to improve model alignment can, counterintuitively, reduce it.

*Example* 4.6 ($\rho_p > 0 \not\Rightarrow$ positive effect of human curation). Consider a simplified text-image system. For simplicity, we assume that each model's output representation coincides with its parameters:

- **Text model** generates captions $\hat{\theta}(\phi, \lambda_\mathcal{H}^\theta) = W_1 \phi + a \lambda_\mathcal{H}^\theta$ describing input images $\phi$, where $W_1$ encodes how the text representation is encouraged to align with the image representation (e.g., to produce captions that are easy for the image model to pass safety filters), and $a$ encodes the direction in representation space favored by human-curated text when $\lambda_\mathcal{H}^\theta$ increases. We assume the mixture distribution of the text model is $P = \mathcal{N}(\hat{\theta}(\beta, \lambda_\mathcal{H}^\theta), \sigma_\theta^2 I)$ with quadratic loss $\ell_\theta(z) = \frac{1}{2} \|\theta - z\|^2, z \sim P$.

- **Image model** generates images $\hat{\phi}(\theta) = W_2 \theta$ based on input captions $\theta$, where $W_2$ encodes the alignment between the image and text representations (e.g., CLIP-style alignment). The mixture data distribution is $Q = \mathcal{N}(\hat{\phi}(\theta), \sigma_\phi^2 I)$ with loss $\ell_\phi(z) = \frac{1}{2} \|\phi - z\|^2, z \sim Q$.

For this system, the gradients and Jacobians are

$$F_p = \theta - W_1 \phi - a \lambda_\mathcal{H}^\theta; \quad F_q = \phi - W_2 \theta$$

$$\nabla_\theta \mathbf{F}_p = I, \nabla_\phi \mathbf{F}_p = -W_1, \nabla_\theta \mathbf{F}_q = -W_2, \nabla_\phi \mathbf{F}_q = I$$

Thus, the sensitivity matrix $S_p = (I - W_1 W_2)^{-1}$.

Consider a specific setting with parameters defined below:

$$W_1 = I, W_2 = \frac{1}{7} \begin{bmatrix} -9 & 6 \\ 6 & 3 \end{bmatrix}, J_p(\theta) = \langle \theta, \begin{bmatrix} 1 \\ 0 \end{bmatrix} \rangle, a = \begin{bmatrix} 1 \\ -1 \end{bmatrix},$$

Since $\mathbb{E}_{P_\mathcal{H}}[-\nabla_\theta \ell_\theta(\theta^*)] = a$ and $\nabla_\theta J_p(\theta) = [1, 0]^T, \forall \theta$,

$$\rho_p = \cos\left(\nabla_\theta J_p(\theta^*), \mathbb{E}_{P_\mathcal{H}}[-\nabla_\theta \ell_\theta(\theta^*)]\right) = \frac{\sqrt{2}}{2} > 0.$$

However,

$$\frac{\partial J_p(\theta^*)}{\partial \lambda_\mathcal{H}^\theta} \propto \langle \nabla_\theta J_p(\theta^*), S_p \mathbb{E}_{P_\mathcal{H}}[-\nabla_\theta \ell_\theta(\theta^*)] \rangle = -\frac{1}{2} < 0.$$

This example illustrates that, under strong cross-model interactions via $W_1$ and $W_2$, even when the human-curated data points in a locally reward-improving direction ($\rho_p > 0$), the dynamics of the coupled text-image system can invert its effect due to distortion induced by the sensitivity matrix.

## 4.3. Positive and Negative Impact of Human Curation

Since $\rho_p > 0$ does not necessarily imply a positive effect of human curation, a natural question is whether we can

identify conditions, expressed in terms of $\rho_p$, under which increasing the fraction of human-curated data is guaranteed to improve alignment in a multi-model system.

**Self-influence.** We first examine conditions under which increasing the fraction of human-curated samples for model $\theta$ improves or degrades its own alignment $J_p(\theta^*)$.

**Corollary 4.7.** *Denote* $\tau_p = \gamma_\theta - L_\theta \varepsilon_\theta - \frac{L_\theta \varepsilon_\theta L_\phi \varepsilon_\phi}{\gamma_\phi - L_\phi \varepsilon_\phi}$ *and let* $m_p$ *be the minimal eigenvalue of* $\frac{S_p + S_p^T}{2}$, *we have* $\|S_p\| \leq \frac{1}{\tau_p}$ *and if* $|\rho_p| > \frac{1}{\sqrt{1 + m_p^2 \tau_p^2}}$, *then* $\text{sign}(\rho_p) \cdot \frac{\partial J_p(\theta^*)}{\partial \lambda_{\mathcal{H}}^\theta} > 0$.

$\|S_p\|$ quantifies the extent to which $S_p$ distorts directional relationships. A larger $\|S_p\|$ implies that discrepancies across dimensions between $\nabla_\theta J_p(\theta^*)$ and $\mathbb{E}_{P_{\mathcal{H}}}[-\nabla_\theta \ell_\theta(\theta^*)]$ may be amplified after the mapping. By Eq. (9), $\|S_p\|$ depends on the interaction settings and the fraction of data originating from other models ($\lambda_\theta^\phi, \lambda_\phi^\theta$). Consequently, increasing curation ratio is beneficial when curation update direction is sufficiently aligned with the reward-improving direction (large $\rho_p > 0$), and the coupled dynamics do not distort these directions too strongly, which is captured by $m_p, \tau_p$. Corollary 4.7 combines these two factors and gives sufficient conditions, for example, if $\rho_p > 0$ is large enough to overcome the distortion caused by coupling iterations ($\rho_p > \frac{1}{\sqrt{1 + m_p^2 \tau_p^2}}$), then $\text{sign}(\rho_p) = 1$ and $\frac{\partial J_p(\theta^*)}{\partial \lambda_{\mathcal{H}}^\theta} > 0$.

Corollary 4.7 is further validated by Example 4.6, where $|\rho_p| = \frac{\sqrt{2}}{2} < \frac{1}{\sqrt{1 + m_p^2/\|S_p\|^2}} \approx 0.997 \leq \frac{1}{\sqrt{1 + m_p^2 \tau_p^2}}$.

**Cross-model influence.** Next, we examine how human curation applied to training model $\theta$ can propagate to another model $\phi$ and improve or degrade its alignment $J_q(\phi^*)$.

Similar to self-influence, we measure the alignment between $\nabla_\phi J_p(\phi^*)$, the direction in space $\Phi$ that would most increase the alignment metric, and $C_q \mathbb{E}_{P_{\mathcal{H}}}[-\nabla_\theta \ell_\theta(\theta^*)]$, the direction by which human curation for $\theta$ indirectly shifts the parameters $\phi$. Note that we multiply $\mathbb{E}_{P_{\mathcal{H}}}[-\nabla_\theta \ell_\theta(\theta^*)]$ by cross-model influence matrix $C_q$ to translate the effect of changes in $\theta$ into the parameter space $\Phi$.

$$\rho_q \triangleq \cos\left(\nabla_\phi J_q(\phi^*), C_q \, \mathbb{E}_{P_{\mathcal{H}}}[-\nabla_\theta \ell_\theta]\right) \in [-1, 1]. \quad (11)$$

Similar to Corollary 4.7, we can then identify conditions, expressed in terms of $\rho_q$, under which human curation of $\theta$ positively or negatively affects the long-term alignment of another model $\phi$. See Corollary E.2 in Appendix E.2.

### 4.4. From Local Sensitivity to Global Deviation

Our analysis so far has focused on the local sensitivities $\frac{\partial J_q(\phi^*)}{\partial \lambda_{\mathcal{H}}^\theta}$ and $\frac{\partial J_p(\theta^*)}{\partial \lambda_{\mathcal{H}}^\theta}$, which quantify how model alignment responds *locally* to changes in the human-curation weight. A natural next question is how the *final* alignment metrics

$J_q(\phi^*)$ and $J_q(\phi^*)$ are affected at the fixed point. The following theorem characterizes, relative to a setting without self-consuming data (i.e., when all training data is real and $\lambda_{\mathcal{R}}^\theta = \lambda_{\mathcal{R}}^\phi = 1$), how much deviation in alignment synthetic data can induce at the fixed point. Denote by $\theta^*(\lambda_{\mathcal{R}}^\theta)$, $\phi^*(\lambda_{\mathcal{R}}^\phi)$ the stable points of the system obtained when the real-data weights are $\lambda_{\mathcal{R}}^\theta$ and $\lambda_{\mathcal{R}}^\phi$, respectively.

**Theorem 4.8.** *Under the conditions of Proposition 3.4 and Assumption 3.5, for any* $\widehat{\lambda}_{\mathcal{R}}^\theta, \widehat{\lambda}_{\mathcal{R}}^\phi > \tau$ *as in Proposition 3.4,*

$$\left| J_p\left(\theta^*\left(\widehat{\lambda}_{\mathcal{R}}^\theta\right)\right) - J_p\left(\theta^*\left(1\right)\right) \right| = \mathcal{O}\left(1 - \widehat{\lambda}_{\mathcal{R}}^\theta\right),$$

$$\left| J_q\left(\phi^*\left(\widehat{\lambda}_{\mathcal{R}}^\phi\right)\right) - J_q\left(\phi^*\left(1\right)\right) \right| = \mathcal{O}\left(1 - \widehat{\lambda}_{\mathcal{R}}^\phi\right).$$

Theorem 4.8 quantifies how much the final alignment scores can deviate from the ideal setting in which models are trained exclusively on real data. It shows that as long as the real-data weights remain above a stability threshold $\tau$, the deviation in alignment scales at most linearly with the amount of synthetic data.

**Discussion.** In Appendix B, we discuss more about the limitations of our framework, experiments and future works.

## 5. Experiments

We consider a two-model system initialized from base models $\theta_0$ and $\phi_0$. Following the self-consuming training paradigm described in Section 2, both models are updated iteratively using data that mix real samples with (human-curated) synthetic data. We conduct experiments on:

**(1) Gaussian models**: We extend Example 4.6 to validate the mechanism more detailed. Let $\theta, \phi \in \mathbb{R}^{24}$, and the mixture distributions $P = (1 - \lambda_{\mathcal{H}}^\theta)\mathcal{N}(\phi, \sigma^2 I) + \lambda_{\mathcal{H}}^\theta \mathcal{N}(\phi + a, \sigma^2 I)$ and $Q = \mathcal{N}(A(t)\theta, \sigma^2 I)$ where $A(t) \in \mathbb{R}^{24 \times 24}$ is a block-diagonal matrix with $12$ $2 \times 2$ blocks. The $i$-th block $A_i(t) = t\beta_i R_i$ where $t$ is the coupling scale, $\beta_i$ controls the interaction magnitude of block $i$, and $R_i$ is a 2-dimensional orthogonal matrix introduces local geometric heterogeneity. We set the alignments $J_p(\theta) = g_p^T \theta - \frac{\eta_p \|\theta\|^2}{2}$ and $J_q(\phi) = g_q^T \phi - \frac{\eta_q \|\phi\|^2}{2}$, and losses $\ell_p(\theta; z) = \frac{1}{2}\|\theta - z\|^2$, $\ell_q(\phi; z) = \frac{1}{2}\|\phi - z\|^2$. $a, g_p, g_q, R_i, \beta_i$ are all pre-set (details in Section D.3). The system is constructed from orthogonal block matrix $R_i$ with matched blockwise curation and reward directions $a, g_p, g_q$. This yields a multi-mode coupled system that remains analytically tractable, cleanly separates coupling magnitude from geometric distortion.

**(2) CIFAR-10** (Krizhevsky et al., 2009): Both $\theta$ and $\phi$ are implemented as class-conditional diffusion models (von Platen et al., 2022) that generate images conditioned on class labels. The base models $\theta_0$ and $\phi_0$ are trained using 25k real samples. Human curation is implemented via hue-based

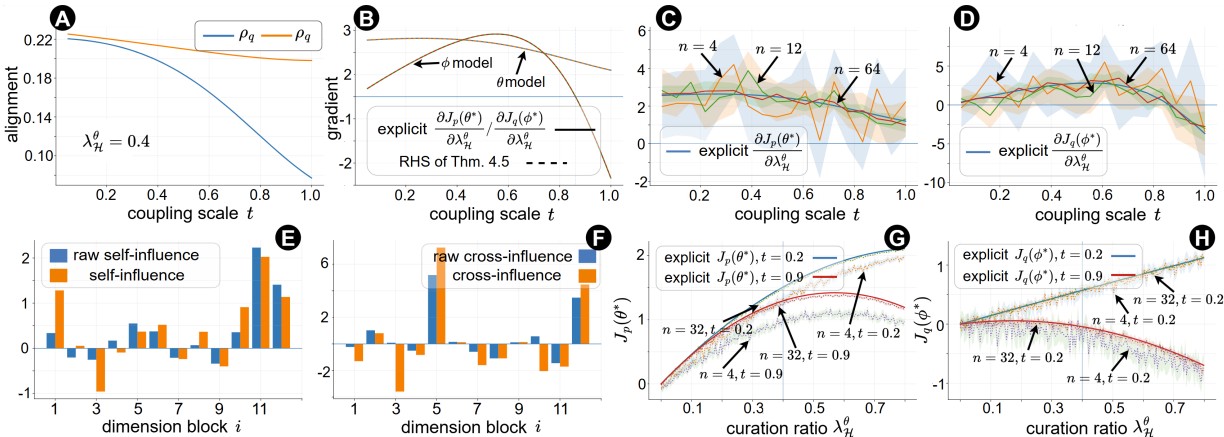

*Figure 3.* Gaussian experiment results. (A): $\rho_p, \rho_q$ in Eq. (10)-(11) for various coupling scale $t$ with fixed $\lambda_{\mathcal{H}}^{\theta} = 0.4$. (B): The theoretical explicit $\frac{\partial J_p(\theta^*)}{\partial \lambda_{\mathcal{H}}^{\theta}}, \frac{\partial J_q(\phi^*)}{\partial \lambda_{\mathcal{H}}^{\theta}}$ and RHS in Theorem 4.5 for $t \in [0.05, 1]$ with fixed $\lambda_{\mathcal{H}}^{\theta} = 0.4$. (C)(D): The theoretical explicit $\frac{\partial J_p(\theta^*)}{\partial \lambda_{\mathcal{H}}^{\theta}}, \frac{\partial J_q(\phi^*)}{\partial \lambda_{\mathcal{H}}^{\theta}}$ and their empirical values from finite samples (of size $n = 4/12/64$) with 95% confidence intervals. (E): The components of $\langle \nabla_\theta J_p(\theta^*), \mathbb{E}_{P_H}[-\nabla_\theta \ell_\theta(\theta^*)] \rangle$ and $\langle \nabla_\theta J_p(\theta^*), S_p \mathbb{E}_{P_H}[-\nabla_\theta \ell_\theta(\theta^*)] \rangle$ within specific dimension blocks. (F): The components of $\langle \nabla_\phi J_q(\phi^*), C_q \mathbb{E}_{P_H}[-\nabla_\theta \ell_\theta(\theta^*)] \rangle$ and $\langle \nabla_\phi J_q(\phi^*), S_q C_q \mathbb{E}_{P_H}[-\nabla_\theta \ell_\theta(\theta^*)] \rangle$ within specific dimension blocks. (G)(H): Explicit $J_p(\theta^*), J_q(\phi^*)$ and their finite-sample empirical estimators with $t \in \{0.2, 0.9\}$ for various $\lambda_{\mathcal{H}}^{\theta}$ with 95% confidence intervals.

reward functions $r_\theta$ and $r_\phi$ (see Appendix D.1 for details), which induce conflicting preferences: model $\theta$ prefers warm-toned images, while model $\phi$ prefers cool-toned images.

**(3) Qwen2.5-0.5B** (Team, 2024): Model $\theta$ is trained to summarize long articles, while model $\phi$ paraphrases relatively short texts. Both models are initialized from Qwen2.5-0.5B-Instruct as base models $\theta_0$ and $\phi_0$, each equipped with a separate LoRA adapter. The reward functions $r_\theta$ and $r_\phi$ are task-specific measures that evaluate how well models $\theta$ and $\phi$ perform summarization and paraphrasing, respectively.

More results and discussions are provided in Appendix D. The code is available at https://github.com/osu-srml/curationBackfire.

### 5.1. Mechanism Validation

We validate the theory mechanism in Section 4. Based on our setting and Eq. (4), the two models updates are $\theta_{t+1} = \phi_t + \lambda_{\mathcal{H}}^{\theta} a$, $\phi_{t+1} = A(t)\theta_t$. We set the iteration number to 100. As shown in Figure 3(A)(B), when $\lambda_{\mathcal{H}}^{\theta} = 0.4$ is fixed, $\rho_p, \rho_q > 0$ for all coupling scales $t \in [0.05, 1]$, while the reward gradient of model $\phi$ drops below 0 which aligns with the conclusion in Example 4.6. Moreover, the RHS of Theorem 4.5 matches the derivative obtained from the closed-form expressions of $J_p, J_q$ (Figure 3(B)), confirming the theorem in this analyzable regime. Furthermore, when the theory predicts a positive derivative (fix $\lambda_{\mathcal{H}}^{\theta} = 0.4, \frac{\partial J_p(\theta^*)}{\partial \lambda_{\mathcal{H}}^{\theta}} > 0$ at $t = 0.2$ or $0.9$ and $\frac{\partial J_q(\phi^*)}{\partial \lambda_{\mathcal{H}}^{\theta}} > 0$ at $t = 0.2$ in Figure 3(B)), the $J_p$ and $J_q$ indeed increase under small curation perturbations in $\lambda_{\mathcal{H}}^{\theta}$ (Figure 3(G)(H)); for $t = 0.9$, $\frac{\partial J_q(\phi^*)}{\partial \lambda_{\mathcal{H}}^{\theta}} < 0$ (Figure 3(B)), and $J_q$ decreases

under small curation perturbations in $\lambda_{\mathcal{H}}^{\theta}$ (Figure 3(H)).

Next, we examine the self and cross-influence in the system. We focus on the dimensional contributions of the key inner products in Theorem 4.5. Based on the blockwise Gaussian settings, the overall self and cross influence can be decomposed into the sum of influences across different dimensions in the system (details provided in Appendix D.3). Moreover, $\langle \nabla_\theta J_p(\theta^*), \mathbb{E}_{P_H}[-\nabla_\theta \ell_\theta(\theta^*)] \rangle$ and $\langle \nabla_\phi J_q(\phi^*), C_q \mathbb{E}_{P_H}[-\nabla_\theta \ell_\theta(\theta^*)] \rangle$ measure whether the curation-induced update direction is locally aligned with the reward-improving direction, while $\langle \nabla_\theta J_p(\theta^*), \mathbb{E}_{P_H}[-\nabla_\theta \ell_\theta(\theta^*)] \rangle$ and $\langle \nabla_\phi J_q(\phi^*), S_q C_q \mathbb{E}_{P_H}[-\nabla_\theta \ell_\theta(\theta^*)] \rangle$ measure the actual self and cross-influence. As shown in Figure 3(E)(F), $S_p$ and $S_q C_q$ can amplify, attenuate, or even reverse self and cross influences in different dimensions, and the dominant dimensions before and after projection can differ substantially. This provides a mechanism-level explanation for why $\rho_p > 0$ does not imply a positive curation effect.

Finally, we focus on the statistical error between the finite-sample estimates and the theoretical values. Let $n \in [4, 64]$ be the sample size used for updating models. By comparing the closed-form $J_p, J_q$ (Fig. 3(G)(H)) and their derivatives of curation ratio (Fig. 3(C)(D)) with finite-sample estimates, the statistical error decreases rapidly as $n$ increases.

### 5.2. Convergence and The Impact of Human Curation

We examine system convergence and the effect of human curation on CIFAR-10 using six settings ($A1$–$A6$). In all settings, each training round uses 8k samples with $\lambda_{\mathcal{H}}^{\theta} = \lambda_{\mathcal{H}}^{\phi} = 0.5$ and $\lambda_{\mathcal{R}}^{\theta} = \lambda_{\mathcal{R}}^{\phi} = 0.5$. To assess the impact of

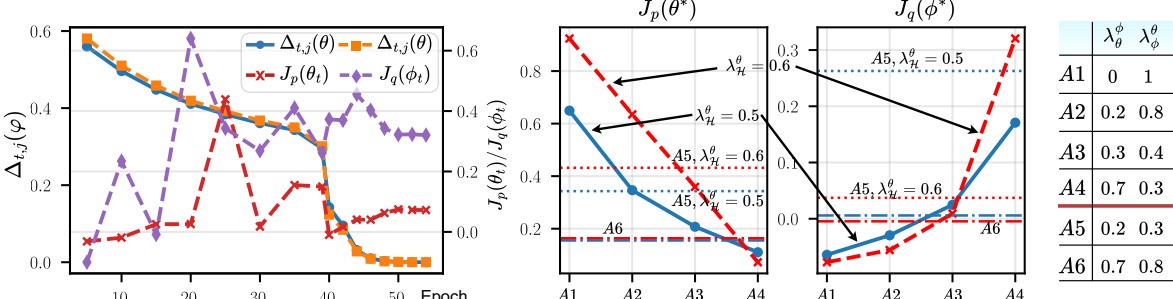

*Figure 4.* Left: The model parameter update ratios $\Delta_{t,j}(\varphi)$, $\varphi \in \{\theta, \phi\}, j \in \{2, 5\}$ and model rewards $J_p(\theta_t)$, $J_q(\phi_t)$ for $A4$ with $\lambda_{\mathcal{H}}^\theta = 0.6$ for models $\theta$ and $\phi$ at different iterations. Middle: Expected model rewards $J_p(\theta^*)$ and $J_q(\phi^*)$ under settings $A1$-$A6$ after convergence. Right: The proportions of cross-model synthetic data for models $\theta$ and $\phi$ in settings $A1$-$A6$.

human curation, we add 2,000 curated samples for model $\theta$, increasing $\lambda_{\mathcal{H}}^\theta$ to 0.6 while keeping all other proportions fixed. Six settings differ only in the fraction of cross-model synthetic data $(\lambda_\phi^\theta, \lambda_\theta^\phi)$: as shown in Figure 4 (right), from $A1$ to $A4$ this fraction increases for model $\theta$ and decreases for $\phi$, while $A5$ and $A6$ serve as control cases.

**Convergence.** We measure convergence using $\Delta_{t,j}(\varphi) = \frac{\|\varphi_t - \varphi_{t-j}\|}{\|\varphi_{t-j}\|_2}$, $\varphi \in \{\theta, \phi\}$, which quantifies the relative change in model parameters. As shown in Figure 4 (left), $\Delta_{t,j}$ for both models drops below $10^{-4}$ and the rewards stabilize, indicating convergence. We assume that $\theta^* = \theta_{54}$ and $\phi^* = \phi_{54}$. Additional results are in Appendix D.1.3.

**Self-influence.** Figure 4 (right) shows that when interactions have limited impact on model $\theta$ ($A1$–$A3$ and $A5$, with $\lambda_\theta^\phi \leq 0.3$), strengthening curation for $\theta$ increases $J_p(\theta^*)$. As $\lambda_\theta^\phi$ grows, however, increasing $\lambda_{\mathcal{H}}^\theta$ from 0.5 to 0.6 instead reduces $J_p(\theta^*)$, indicating stronger distortion from model interactions. Because $\theta$ and $\phi$ have conflicting preferences, stronger interaction causes the curation-induced update $\mathbb{E}_{P_\mathcal{H}}[-\nabla_\theta \ell_\theta]$ to deviate from $\nabla_\theta J_p$, reducing $\rho_p$ and potentially making it negative. Consistent with Corollary 4.7, this leads to $\frac{\partial J_p(\theta^*)}{\partial \lambda_{\mathcal{H}}^\theta} < 0$ in $A4$. Moreover, with the same $\lambda_\phi^\theta$, $A2$ exhibits a larger improvement in $J_p(\theta^*)$ than $A5$ because $\phi$ depends more strongly on $\theta$, which weakens $\phi$'s influence on $\theta$. The same mechanism explains the corresponding differences in $J_q(\phi^*)$.

**Cross-influence.** Due to the conflicting preferences, when model $\phi$'s curation training data is sourced entirely from model $\theta$ ($\lambda_\phi^\theta = 1$), increasing the amount of curation data tends to decreases $J_q$. Moreover, training on a large volume of preference-conflicting samples ($\lambda_\phi^\theta \geq 0.4$) can drive $J_q(\phi^*)$ close to or below 0, as observed in settings $A1 \sim A3$ and $A6$ in Figure 4.

As the influence of $\theta$ on $\phi$ decreases (i.e., as $\lambda_\phi^\theta$ is reduced), the projected curation-induced update $C_q \mathbb{E}_{P_\mathcal{H}}[-\nabla_\theta \ell_\theta]$ becomes better aligned with model $\phi$'s reward-improving direction, leading to an increase in $\rho_q$. By Corollary E.2, when

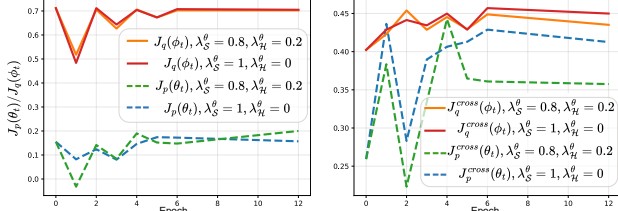

*Figure 5.* Left: models $\theta$ and $\phi$'s rewards on their own evaluation datasets. Right: Rewards on cross-domain evaluation datasets.

$\rho_q$ is sufficiently large, $\frac{\partial J_q(\phi^*)}{\partial \lambda_{\mathcal{H}}^\theta} > 0$, which explains the improvement in $J_q(\phi^*)$ observed in setting $A4$. Finally, when the two models are strongly coupled ($A6$), neither model produces high-reward samples after convergence, rendering curation ineffective; consequently, rewards remain low and increasing $\lambda_\theta^\phi$ has little effect on either $J_p(\theta^*)$ or $J_q(\phi^*)$.

### 5.3. Preference Domain Mismatch (PDM) Experiments

Next, we use Qwen2.5-0.5B to study preference domain mismatch (PDM). To isolate the effect of PDM, We consider an extreme setting with $\lambda_\theta^\phi = \lambda_\phi^\theta = 1$ and $\lambda_\mathcal{R}^\theta = \lambda_\mathcal{R}^\phi = 0$. In this case, model $\theta$ is trained exclusively on short texts but evaluated on long-article summarization, while model $\phi$ is trained on long articles but evaluated on short-text paraphrasing. Consequently, cross-model interactions induce training inputs that are mismatched with each model's target evaluation inputs, resulting in PDM.

We define cross-domain rewards $J_p^{cross}(\theta_t)$, $J_q^{cross}(\phi_t)$ as the expected rewards when each model is evaluated on the other task's evaluation inputs but scored using its own reward function. As shown in Figure 5, target rewards $J_p(\theta_t)$, $J_q(\phi_t)$ remain nearly unchanged, while the cross-domain rewards improve. This pattern indicates that cross-only updates adapt each model to the other domain's input distribution, but this adaptation transfers weakly to the target evaluation domain. As a result, although model $\theta$ tends to shorten inputs and $\phi$ tends to preserve input length, strong PDM obscures the effect of model coupling when preference alignment is measured on the target evaluation sets.

## Impact Statement

This paper presents work whose goal is to advance the field of Machine Learning. The contributions are theoretical in nature and focus on analyzing the dynamics of self-consuming multi-model iterative retraining and how the human curation influences the multi-model ecosystem. While Machine Learning systems may have broad societal impacts, we do not identify any specific ethical issues or foreseeable societal consequences that require separate discussion for this work.

## Acknowledgements

This work was funded in part by the National Science Foundation under award number IIS2202699, IIS-2416895, IIS-2301599, CMMI2301601, and DMS-2529302.

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

# A. Related work

**Self-consuming Training on Synthetic Data.** A rapid growing body of work studies iterative retraining on model-generated data from theoretical or empirical perspectives. The self-consuming training loops can lead to several degradation phenomena. Shumailov et al. (2024) show that indiscriminate recursive training can cause irreversible defects and disappearance of distributional tails. Alemohammad et al. (2023) analyze "autophagous" loops and show systematic quality-diversity deterioration without sufficient fresh real data, formalized as Model Autophagy Disorder (MAD). At the same time, "collapse" is not universal: Bertrand et al. (2024) provide conditions under which iterative retraining on mixtures of real and synthetic data can be stable, including empirical validation on diffusion models. Gerstgrasser et al. (2024); Seddik et al. (2024) both argue that accumulating synthetic data alongside real data can mitigate collapse. Theoretical refinements examine regimes of strong collapse and how model capacity and mixture proportions affect the bias-variance decomposition of errors (Dohmatob et al., 2025). Fu et al. (2025) theoretically study how both model architecture and the proportion between real and synthetic data influence recursive training loops. Meanwhile, self-consuming training induces bias amplification. Model-induced distribution shifts can amplify unfairness, and Wyllie et al. (2024) analyze fairness feedback loops over generations and propose "algorithmic reparation" as a corrective mechanism. In language, Wang et al. (2026) show that iterative retraining on synthetic data coupled with deployment-driven feedback can systematically amplify preference bias. In vision, Chen et al. (2024) evaluate the amplifying effect of self-consuming training on synthetic data on social biases. Complementary to studies of bias amplification under recursive training, recent benchmark work (Tan et al., 2026) emphases the need for harm-aware fairness evaluation in vision and multimodal models. Moreover, when systems amplify bias, Taori & Hashimoto (2023) formalize data feedback loops in conditional prediction and connect stability to calibration-like properties. Furthermore, extending to multi-model self-consuming systems, Gao & Li (2025) analyze the co-evolving dynamic while modeling the system with simplified Gaussian distribution. Hu et al. (2025) empirically explore the multi-model collapse pattern for recursive generate-train loops. However, while self-consuming training in multi-model systems is quite common (Taori et al., 2023; Yu et al., 2024; Shumailov et al., 2024), its general evolution stability remains unclear, which is one of the objectives of our explorations.

**Human Curation and Model Alignment.** Real-world synthetic data pipelines typically include user curation. For example, the JourneyDB dataset contains images generated and human-curated originating from the Midjourney Discord (Sun et al., 2023). Feng et al. (2025) study verifier-based selection on synthetic data can prevent collapse even when generation is imperfect. Ferbach et al. (2024) show that human curation can be regarded as an implicit preference optimization mechanism, and iterative retraining can improve expected rewards under suitable curation assumptions. In contrast, if the curation is adversarial, model alignment will be disrupted (Wei & Zhang, 2025). For preference optimization and alignment, RLHF is a widely used technique to align models to human preference (Christiano et al., 2017; Ouyang et al., 2022). Direct Preference Optimization (Rafailov et al., 2023), as its variant, shows that preference alignment can be achieved via a simple classification-style objective without explicit modeling. The lack of analysis on the impact of curation on preference alignment in multi-model self-consuming systems forms the basis of our work.

**Performative Prediction.** Similar to the stability results from Bertrand et al. (2024), our stability analysis in Theorem 3.6 is related to the literature on "performative prediction" (Perdomo et al., 2020). Performative prediction formalizes settings in which the model outputs affect the data-generating process, yielding a retraining dynamic whose fixed points correspond to performative stable solutions (Xie et al., 2025; Jin et al., 2024; 2026). Self-consuming and model interactions are also included in this framework. Perdomo et al. (2020) provide a foundation for treating iterative training as interacting with a decision-dependent distribution map–precisely the lens adopted in our multi-model evolution setting. Further work (Brown et al., 2022) generalizes the framework by allowing the induced distribution to depend on an evolving system state, thereby capturing path dependence and multi-stability. Moving beyond stylized convex objective, Mofakhami et al. (2023) extend stability and convergence analyses to nonconvex regimes and highlights that stability is governed by the sensitivity or predictions to deployment. Meanwhile, multi-agent performative prediction captures coupled learning dynamics where multiple decision makers jointly influence and are influenced by the evolving data distribution. Piliouras & Yu (2023) characterize regimes ranging from global stability to instability and even chaos for the multi-agent system. Li et al. (2022) develop a decentralized, coupled viewpoint in which multiple agents jointly influence data distribution and study convergence under greedy deployment and consensus mechanisms.

# B. Discussion

Our framework provides a mechanistic and analyzable understanding of multi-model self-consuming ecosystem's long-term evolution, stability and its preference alignment. Beyond the main theoretical and experiment results, we highlight several extensions and implications, and then discuss the limitations of the present analysis.

**Preference alignment along the training trajectory.** Our curation-effect analysis focuses on how alignment metrics change around the stable point after convergence, since analysis at equilibrium isolates the system-level externalities that persist after convergence caused by model interaction. The same framework can be extended to analyze alignment metrics during the iterative process by replacing the fixed evaluation marginal distribution with the iteration-dependent induced marginal distribution. In this case, $J_p(\theta_t) = \mathbb{E}_{y \sim \mathcal{E}_\theta, x \sim p_{\theta_t}(x|y)} r_\theta(x, y) \to J_p(\theta_t, \phi_t) = \mathbb{E}_{y \sim P^y(\theta_t, \phi_t), x \sim p_{\theta_t}(x|y)} r_\theta(x, y)$. Therefore, $J_p$ depends not only on model parameter $\theta_t$ but also on parameters of models that interact with model $\theta$. The remaining derivations, such as $\frac{\partial J_p(\theta_t, \phi_t)}{\partial \lambda_{\mathcal{H}}^\theta}$ and decomposition of self-influence and cross-influence, can be completed following the current framework.

**$N$-model framework.** We focus on the two-model system as the minimal unit that already reveals the core mechanism of system stability and how human curation affect the preference alignments. Human curation is not a per-model improvement in isolation, but an intervention on a coupled dynamic system where cross-model influences can dominate. Extending the framework to $N$ models ($N > 2$) is conceptually straightforward in terms of block-structured sensitivities since model interaction is achieved by linearly mixing the data generated by different models in the training dataset, but introduces additional technical challenges, such as coupling-graph topology and more complex equilibrium structure. We do not attempt to resolve here, and we expect theory-related phenomenons to remain relevant in larger ecosystems, and potentially more pronounced as indirect influence paths proliferate.

**Preference domain mismatch (PDM).** PDM highlights that coupling-induced preference conflicts can be weakly reflected, or even appear absent under a single alignment evaluation metric when the induced training distribution shifts into regions not well covered by the evaluation domain. We expect such mismatch to be common in practice, particularly when the training data is acquired or filtered from heterogeneous internet corpora, where selection pipelines may fail to capture the relevant preference data or may emphasize preferences that are effectively orthogonal to the target objective. Importantly, PDM should not be regarded as "guaranteeing safety" of internet-sourced synthetic data, but it explains how preference conflicts and distributional shifts may remain invisible to a narrow alignment evaluation metric. The internet-sourced synthetic data can appear "safe" under that metric while still inducing distributional shifts.

**Practical implications: local diagnostics and interventions.** Our theory is not primarily targeted to precisely predict end-to-end training outcomes, but to provide local diagnostics and intervention rules for coupled self-consuming ecosystems. By estimating the local and cross-model responses ($S_p, S_q, C_p, C_q$) together with the sign and decomposition of the curation derivatives ($\frac{\partial J_p(\theta)}{\partial \lambda_{\mathcal{H}}^\theta}, \frac{\partial J_q(\phi)}{\partial \lambda_{\mathcal{H}}^\phi}$), self-influence can be separated from cross-model propagation, and whether an increase in curation is likely to be neutralized or even reversed by ecosystem externalities and model interactions can be assessed without long-time simulation. In practice, these quantities can be approximated via local perturbation experiments, and serve as local indicators of stability and preference transmission. When cross-influence terms dominate or the predicted derivative sign contradicts the single-model intuition, they provide an early warning that preference conflicts are being amplified, motivating targeted mitigations such as improving the real data ratios, reducing cross-sourced curation, or isolating coupling pathways. However, as discussed in Section D.3, precise estimation of these matrices is difficult for large models. Both the assumption-level quantities and the Section 4 matrices are affected by sampling noisse, finite-sample statistical error, and approximation error from large-scale high-order matrix estimation. The development of accurate and scalable estimators for large models is also an important future research direction (Basu et al., 2021).

**Real world examples.** We provide several examples of where the problem discussed in this paper arises in practice, some of which further elaborate the examples in Section 1. A first example is modern alignment and safety pipelines, where preference signals are explicitly heterogeneous. Safe RLHF (Dai et al., 2024) formulates a tension between helpfulness and harmlessness, while Safety-Tuned LLaMAs (Bianchi et al., 2024) shows that emphasizing helpfulness alone can produce unsafe behavior, whereas stronger safety tuning can induce exaggerated refusals on benign inputs. This is precisely the kind of naturally occurring preference conflict that motivates our multi-model view: different models in the pipeline have different preferences (helpfulness/harmlessness), and their interaction can shape the final model behavior in nontrivial ways.

A second example is instruction-tuning/model-distillation pipelines such as Self-Instruct (Wang et al., 2023) and Alpaca

(Taori et al., 2023), where one language model generates instruction-response data, the outputs are filtered/curated, and another model is then fine-tuned on the resulting synthetic data. In such settings, the relevant preference differences are not artificial: they can correspond to natural trade-offs such as concise vs. detailed responses, style/formality preferences, or general helpfulness vs. safety-oriented behavior. A third example is web-scale multimodal data construction. CapsFusion (Yu et al., 2024) uses LLMs to consolidate and refine web image-text pairs and image-only data into synthetic captions for future multimodal model training, while datasets such as JourneyDB (Sun et al., 2023) further illustrate that large-scale generated image–text ecosystems already involve content- and style-sensitive curation. In such settings, natural preference differences can arise between literal captioning vs. richer descriptive captioning, style fidelity vs. semantic coverage, or aesthetic preference vs. downstream task utility.

Another realistic case is AI assisted news production in a politically polarized media ecosystem. Different outlets serve audiences with systematically different political leanings (right/liberal), while major news organizations such as AP and Reuters publicly state that generative AI is already used in news workflows, including summaries, headlines, writing, editing, and publishing (The Associated Press, 2024; Reuters, 2024). In such an environment, different outlets may naturally prefer different political framings of the same event; for example, emphasizing border security and law-and-order versus civil rights and inclusion. If AI tools are used to draft, summarize, rewrite, or prioritize content under these different editorial preferences, then even mild framing differences can affect which narrative preferences are preserved, amplified, and reused downstream. This concern is also consistent with prior work (Wang et al., 2025), which shows that recursive synthetic training can amplify political bias across generations.

### B.1. Limitations of our framework

**Finite samples' statistical error.** Our theoretical analysis does not explicitly quantify finite-sample effects, i.e., the discrepancy between the empirical distribution induced by a finite training set and the population (or iteration-dependent induced) distribution. Such discrepancies are unavoidable in both our experiments and real-world deployments. Prior work suggests that their impact can be mitigated by increasing the amount of training data (thereby reducing estimation error) or by improving algorithmic stability through conservative step-size choices and learning-rate schedules, among other standard practices (Perdomo et al., 2020; Hardt et al., 2016; Cutler et al., 2024). A more complete treatment should incorporate empirical estimation error bounds and stability analysis considering the stochastic optimization noise and training data distribution shifts due to self-consuming. We view this as complementary to our theory results, and it would quantify when stability conclusions are robust under sampling noises.

**Theory assumptions.** Following prior work on iterative self-consuming retraining and performative prediction (Perdomo et al., 2020; Bertrand et al., 2024; Ferbach et al., 2024), we adopt standard regularity assumptions (Assumption 3.2–3.5) that ensure (i) a stable learning update (e.g., smoothness and a well-behaved optimization landscape) and (ii) controlled sensitivity of the induced training distribution to model changes. These assumptions enable characterizations of stability and local response in our coupled retraining dynamics. While these assumptions facilitate a clean first-step analysis of multi-model self-consuming loops in our paper, similar to prior work's limitations, they can be difficult to verify for modern over-parameterized neural networks and may not fully reflect the nonconvex optimization landscape encountered in practice. Encouragingly, recent work has begun relaxing these requirements, e.g., by establishing guarantees under smooth nonconvex objectives or by replacing parameter-space regularity with prediction-level stability conditions that better align with neural network models (Mofakhami et al., 2023; Li & Wai, 2024; Zhao, 2022). Extending these more general analyses to strongly coupled multi-model self-consuming systems remains challenging, since model interaction introduces coupled feedback pathways and more complex equilibrium structures that are absent in single-model settings. In particular, coupling can also propagate local estimation errors across models and iterations.

**Experiments.** To cleanly expose coupling-induced preference conflicts, since our theory mainly depends on self and cross influences and preference-direction alignment, not on the particular rewards, we adopt controlled reward constructions that may produce "failure modes" which are consistent with previous conclusion (Ferbach et al., 2024). For example, model outputs visually degenerate images under extreme preferences, such as the almost monochromatic images shown in Figure 8. We treat our experiment setups as stress tests and existence proofs. Observing non-monotonicity that enhancing curation strength does not improve preference alignment in a simplified environment with explicit preferences and strong coupling is sufficient to prove that monotonic improvement from increased curation cannot be assumed as a general principle. Moreover, current experiment results also raise a theoretical question: *In a self-consuming ecosystem with multi-model interactions, how can we characterize the distance between the stable point of such system after convergence with human curation, and the optimal point training with only real data?* This problem has been solved in the single-model scenario (Bertrand et al.,

2024; Ferbach et al., 2024), but it remains unknown for multi-model systems.

We view exploring the extension of our framework and addressing these limitations as promising directions for future work.

# C. Conclusion

This paper studies the long-term evolution and preference alignment of the multi-model ecosystem in self-consuming training loops. We theoretically analyze the stability conditions in such system under multi-model interactions. Building on this foundation, we examine how human curation affects preference alignment when multiple, potentially heterogeneous preferences coexist in the multi-model system, disentangling both self-influence and cross-influence transmitted to other models through model interactions. Our findings indicate that the preference alignment of one single model is jointly shaped by self-consuming loops and model interactions, and highlight that increasing the curation strength does not necessarily improve preference alignment.

# D. Experiment Discussion And Additional Experiments

## D.1. Experiments on CIFAR-10 datasets

In this section, we present the details of the experiments in the main paper. We first detail the model architecture and training settings in Appendix D.1.1. Next, the formal definition of rewards $r_\theta, r_\phi$ are shown in Appendix D.1.2. Finally, we show more detailed experiments in Appendix D.1.3.

### D.1.1. SETTINGS

**Model architecture.** We implement the two class-conditional diffusion models $\theta$, $\phi$ using the *UNet2DModel* architectures (Ronneberger et al., 2015) from the Hugging Face Diffusers library (von Platen et al., 2022). The networks are both 4-level 2D UNet that maps a noisy RGB $x \in \mathbb{R}^{3 \times 32 \times 32}$ and timestep $t$ to an output of the same shape. We set $block\_out\_chaneels = (92, 192, 192, 384)$ with one ResNet layer per block ($layers\_per\_block = 1$). Attention blocks are used at the intermediate spatial resolutions ($16 \times 16$ and $8 \times 8$ for $32 \times 32$ inputs), while the highest and loweset resolutions are standard ResNet blocks. We train them with a DDPM noise scheduler and for sampling, we use DDIM instantiated from DDPM configuration (Ho et al., 2020; Song et al., 2021). Class conditioning is provided via integer labels $0 \sim 9$, whose embeddings are added to the timestep embeddings in the UNet forward pass. All unspecified UNet hyperparameters follow the Diffusers defaults.

**Iterative retraining process.** Algorithm 1 describes the process of iterative retraining on mixture datasets for the two-model interacting system, and model $\theta$ and $\phi$ are updated synchronously in our CIFAR-10 experiments. If models are updated asynchronously, the algorithm for asynchronous updates can be obtained by replacing the simultaneous updates of the two models in Algorithm 1 with sequential updates, and ensuring that the model parameters for cross-model samples are adjusted accordingly.

**Evaluation setting.** For model $\theta$ and $\phi$, which share the same model architecture in our experiment, we use a fixed set of 5k CIFAR-10 data samples as the common evaluation dataset to compute empirical estimates of $J_p(\theta_t)$ and $J_q(\phi_t)$.

### D.1.2. REWARD DESIGN

**HSV representation of a pixel.** HSV is a cylindrical reparameterization of RGB intended to align better with intuitive color attributes. For a pixel $u = (R, G, B) \in [0, 1]^3$ of an image, denote $V = \max(R, G, B)$, $m = \min(R, G, B)$, and $\Delta = V - m$. The Value is $V$, and the Saturation is

$$S = \begin{cases} 0, & V = 0, \\ \frac{\Delta}{V + 10^{-8}}, & V > 0. \end{cases}$$

The Hue is

$$H = \begin{cases} 0, & \Delta = 0, \\ \frac{1}{6}\left(\frac{G-B}{\Delta} \mod 6\right), & V = R, \\ \frac{1}{6}\left(\frac{B-R}{\Delta} + 2\right), & V = G, \\ \frac{1}{6}\left(\frac{R-G}{\Delta} + 4\right), & V = B. \end{cases}$$

---

**Algorithm 1** Iterative retraining of two-model ecosystem under synchronous updates with model interactions

---

**Input:** Real data $\mathcal{R}^\theta = \{(x, y)\}$ where $x$ is the image and $y$ is the label, and $\mathcal{R}^\phi$, reward functions $r_\theta$ and $r_\phi$, learning procedures $\mathcal{A}_\theta$ and $\mathcal{A}_\phi$

**Param:** For model $\theta$: training data size $N_\theta$, real data ratio $\lambda_\mathcal{H}^\theta$, synthetic data ratio $\lambda_\mathcal{S}^\theta$, synthetic curated data ratio $\lambda_\mathcal{H}^\theta$, cross-model data ratio $\lambda_\theta^\phi$. Symmetrically, for model $\phi$: training data size $N_\phi$, real data ratio $\lambda_\mathcal{H}^\phi$, synthetic data ratio $\lambda_\mathcal{S}^\phi$, synthetic curated data ratio $\lambda_\mathcal{H}^\theta$, cross-model data ratio $\lambda_\phi^\theta$. Iteration number $T$.

$\theta_0 = \mathcal{A}_\theta(\mathcal{R}^\theta)$, $\phi_0 = \mathcal{A}_\phi(\mathcal{R}^\phi)$, $\mathcal{D}_0^\theta = \mathcal{R}^\theta$, $\mathcal{D}_0^\phi = \mathcal{R}^\phi$

**for** $t = 1$ **to** $T$ **do**

    $\mathcal{S}_t^\theta = \{(x_i, y_i)\}_{i=1}^{N_\theta \lambda_\mathcal{S}^\theta (1-\lambda_\theta^\phi)} \cup \{(\widehat{x}_i, \widehat{y}_i)\}_{i=1}^{N_\theta \lambda_\mathcal{S}^\theta \lambda_\theta^\phi}$ where $y_i \sim \mathcal{D}_{t-1}^{\theta,y}$, $x_i \sim p_{\theta_{t-1}}(x|y_i)$ and $\widehat{y}_i \sim \mathcal{D}_{t-1}^{\phi,y}$, $\widehat{x}_i \sim q_{\phi_{t-1}}(x|\widehat{y}_i)$, and $\mathcal{D}_{t-1}^{\theta,y}$, $\mathcal{D}_{t-1}^{\phi,y}$ are label $y$'s empirical marginal distribution of $\mathcal{D}_{t-1}^\theta$, $\mathcal{D}_{t-1}^\phi$, respectively.

    $\mathcal{S}_t^\phi = \{(x_i, y_i)\}_{i=1}^{N_\phi \lambda_\mathcal{S}^\phi (1-\lambda_\phi^\theta)} \cup \{(\widehat{x}_i, \widehat{y}_i)\}_{i=1}^{N_\phi \lambda_\mathcal{S}^\phi \lambda_\phi^\theta}$ where $\widehat{y}_i \sim \mathcal{D}_{t-1}^{\theta,y}$, $\widehat{x}_i \sim p_{\theta_{t-1}}(x|\widehat{y}_i)$ and $y_i \sim \mathcal{D}_{t-1}^{\phi,y}$, $x_i \sim q_{\phi_{t-1}}(x|y_i)$

    **for** $j = 1$ **to** $N_\theta \lambda_\mathcal{H}^\theta$ **do**

      **if** $j < N_\theta \lambda_\mathcal{H}^\theta (1 - \lambda_\theta^\phi)$ **then**

        $\widehat{y}_j \sim \mathcal{D}_{t-1}^{\theta,y}$, $x_1, ..., x_K \sim p_{\theta_{t-1}}(x|\widehat{y}_j)$

        $x_k, 1 \le k \le K$ is selected based on Eq. (3) using reward $r_\theta$ and $\widehat{x}_j \leftarrow x_k$   {curated synthetic data generated from model $\theta$}

      **else**

        $\widehat{y}_j \sim \mathcal{D}_{t-1}^{\phi,y}$, $x_1, ..., x_K \sim q_{\phi_{t-1}}(x|\widehat{y}_j)$

        $x_k, 1 \le k \le K$ is selected based on Eq. (3) using reward $r_\phi$ and $\widehat{x}_j \leftarrow x_k$   {cross-model curated synthetic data generated from model $\phi$}

      **end if**

    **end for**

    $\mathcal{H}_t^\theta = \{(\widehat{x}_j, \widehat{y}_j)\}_{j=1}^{N_\theta \lambda_\mathcal{H}^\theta}$

    Compute $\mathcal{H}_t^\phi$ symmetrically

    $\mathcal{D}_t^\theta = \mathcal{S}_t^\theta \cup \mathcal{H}_t^\theta \cup \{(x_i, y_i) \in \mathcal{R}^\theta\}_{i=1}^{N_\theta \lambda_\mathcal{R}^\theta}$, $\mathcal{D}_t^\phi = \mathcal{S}_t^\phi \cup \mathcal{H}_t^\phi \cup \{(x_i, y_i) \in \mathcal{R}^\phi\}_{i=1}^{N_\phi \lambda_\mathcal{R}^\phi}$

    $\theta_t = \mathcal{A}_\theta(\mathcal{D}_t^\theta)$, $\phi_t = \mathcal{A}_\phi(\mathcal{D}_t^\phi)$

**end for**

---

**Why we use hue-derived rewards.** Hue is a standard, explicitly controlled color attribute. Prior image-enhancement research explicitly treats hue preservation/correction as a design objective to avoid perceptual hue distortions, including methods that keep hue constant in HSI/HSV-style processing pipelines and correction schemes designed to be applicable to deep-learning-based enhancement (Kinoshita & Kiya, 2020). Reinforcement-learning formulations for image enhancement are well established (Zhang et al., 2021; Park et al., 2018) and typically define actions as interpretable color/tonal adjustments and optimize policies using hand-crafted, non-reference reward signals. Using a hue-derived reward is a direct instantiation of this color-aware reward shaping paradigm. In our setting, we adopt hue-based rewards to better illustrate the preference conflicts of different vision models in the experiment.

**Reward Design.** Given a model generated image $x \in [-1, 1]^{3 \times 32 \times 32}$, we first map it to $[0, 1]$ via $x_{01} = \frac{clip(x, -1, 1) + 1}{2}$. For a pixel $u$ in $x_{01}$, we compute its HSV representation and obtain its hue $H(u) \in [0, 1)$ and saturation $S(u) \in [0, 1]$. Hue $H$ represents a normalized angle on the color wheel (red–yellow–green–cyan–blue–magenta–red), while saturation $S$ measures colorfulness. For near-gray pixels, $S \approx 0$ and hue becomes unstable. For a given hue interval $\mathcal{I} \subset [0, 1)$, let

$$\mathbf{1}_\mathcal{I}(t) = \begin{cases} 1, & t \in \mathcal{I}, \\ 0, & \text{otherwise}, \end{cases}$$

be the membership indicator. The hue-band occupancy score of model-generated image $x$ is:

$$Band_\mathcal{I}(x) = \frac{1}{32 \times 32} \sum_{i=1}^{32} \sum_{j=1}^{32} \mathbf{1}_\mathcal{I}\big(H(x_{01}[i,j])\big) S\big(H(x_{01}[i,j])\big)^{1.5},$$

where $x_{01}[i, j]$ is the pixel at $i$th row, $j$th column of $x_{01}$. A larger $Band_\mathcal{I}(x)$ value indicates that a larger fraction of

high-saturation pixels lies within the target hue band ($\mathcal{I}$), hence the global appearance is more aligned with the corresponding color tone.

In our experiment, we map "warm" and "cool" tones to two hue intervals on the color wheel. We implement $\mathcal{I}_{warm} = [0.92, 1) \cup [0, 0.17]$ covering red-orange-yellow, and $\mathcal{I}_{cool} = [0.5, 0.72]$ covering cyan-blue. Accordingly, we define the warm and cool scores as

$$Warm(x) = Band_{\mathcal{I}_{warm}(x)}, \ Cool(x) = Band_{\mathcal{I}_{cool}(x)}.$$

This choice aligns with common perceptual conventions: red/orange/yellow hues are typically perceived as "warm", whereas cyan/blue hues are perceived as "cool". When a larger portion of high-saturation pixels falls into the corresponding band, the overall appearance exhibits a stronger warm/cool tone.

For model generated images $x$, we define two label-free rewards consisting of a color preference term ($Warm(x)$ or $Cool(x)$) and a global-statistics regularizer ($R(x)$) as

$$r_\theta(x) = 3\, Warm(x) + 0.3\, R(x), \ r_\phi(x) = 3\, Cool(x) + 0.3\, R(x).$$

Let $(\mu_0, \sigma_0) \in \mathbb{R}^3 \times \mathbb{R}^3$ denote the channel-wise mean and standard deviation of CIFAR-10 images. $R(x)$ is the lightweight regularizer that encourages generated images to match $(\mu_0, \sigma_0)$, and $R(x) = -\big(\|\mu(x) - \mu_0\|_2 + \|\sigma(x) - \sigma_0\|_2\big)$.

In summary, the rewards $r_\theta, r_\phi$ respectively bias the generated distribution toward warm/cool hues. The rewards are used for candidate curation and for measuring preference alignment on the evaluation datasets.

### D.1.3. ADDITIONAL RESULTS OF CIFAR-10 EXPERIMENTS

**Additional results of stability and reward experiments.** In Figure 6, we show the expected reward meaning values of model $\theta$ and $\phi$, $J_p(\theta_t)$ and $J_q(\phi_t)$ respectively, at different iterations under settings $A1$-$A6$. As shown, both metrics stabilize in the later iterations, indicating that the training dynamics approach a convergent regime. Therefore, let $\theta^* = \theta_{54}, \phi^* = \phi_{54}$.

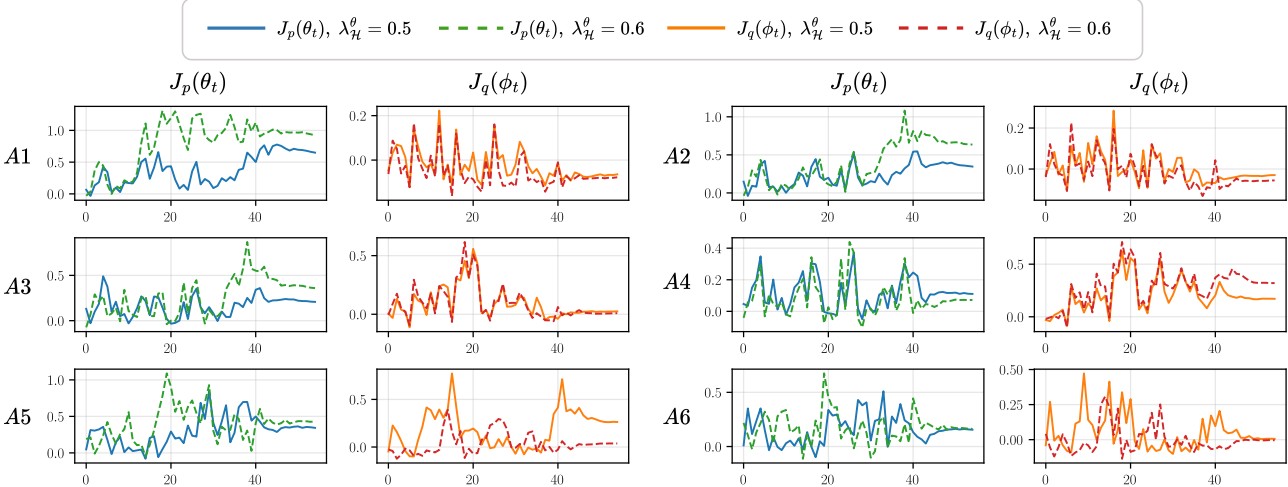

*Figure 6.* Reward meaning values $J_p(\theta_t)$ and $J_q(\phi_t)$ on the fixed evaluation dataset at different iterations for settings $A1 - A6$.

**Additional analysis of model generated images.** In Figure 7, 8, 9, we show 256 images generated by model $\theta$ and $\phi$ at different training iterations, conditioned on the labels from the fixed evaluation datasets, for the baseline as well as the $A1, A5$ and $A6$ settings. For $A1$ and $A5$, as retraining proceeds, a growing fraction of model generated samples become blurry, near-uniform color images, for which the original class label is no longer visually discernible. This behavior is a direct consequence of our experimental design: to induce strong preference conflict, we define a reward function that is easiest to maximize when an image becomes largely single-colored. The observation is also consistent with prior findings (Ferbach et al., 2024) that the model parameter related to the maximal reward value usually do not coincide with the model optimal parameter under standard training objectives.

Recall that model $\theta$ tends to generate warm-toned images, whereas model $\phi$ prefers cool-toned images. Under the $A1$ setting, model $\theta$ is trained only on its own curated data together with real data, while all curated data used to train model $\phi$

are sourced from model $\theta$. As a result, images generated by model $\phi$ gradually shift toward warm tones as training proceeds in Figure 7. Moreover, increasing the proportion of curated data in model $\theta$'s training dataset causes both models' outputs to become warm-toned earlier than before, and the effect is more pronounced (bottom 2 rows in Figure 7). This qualitative shift is consistent with the quantitative trends: model $\theta$'s reward increases, while model $\phi$'s reward decreases after improving the human curation strength of model $\theta$ in Figure 4.

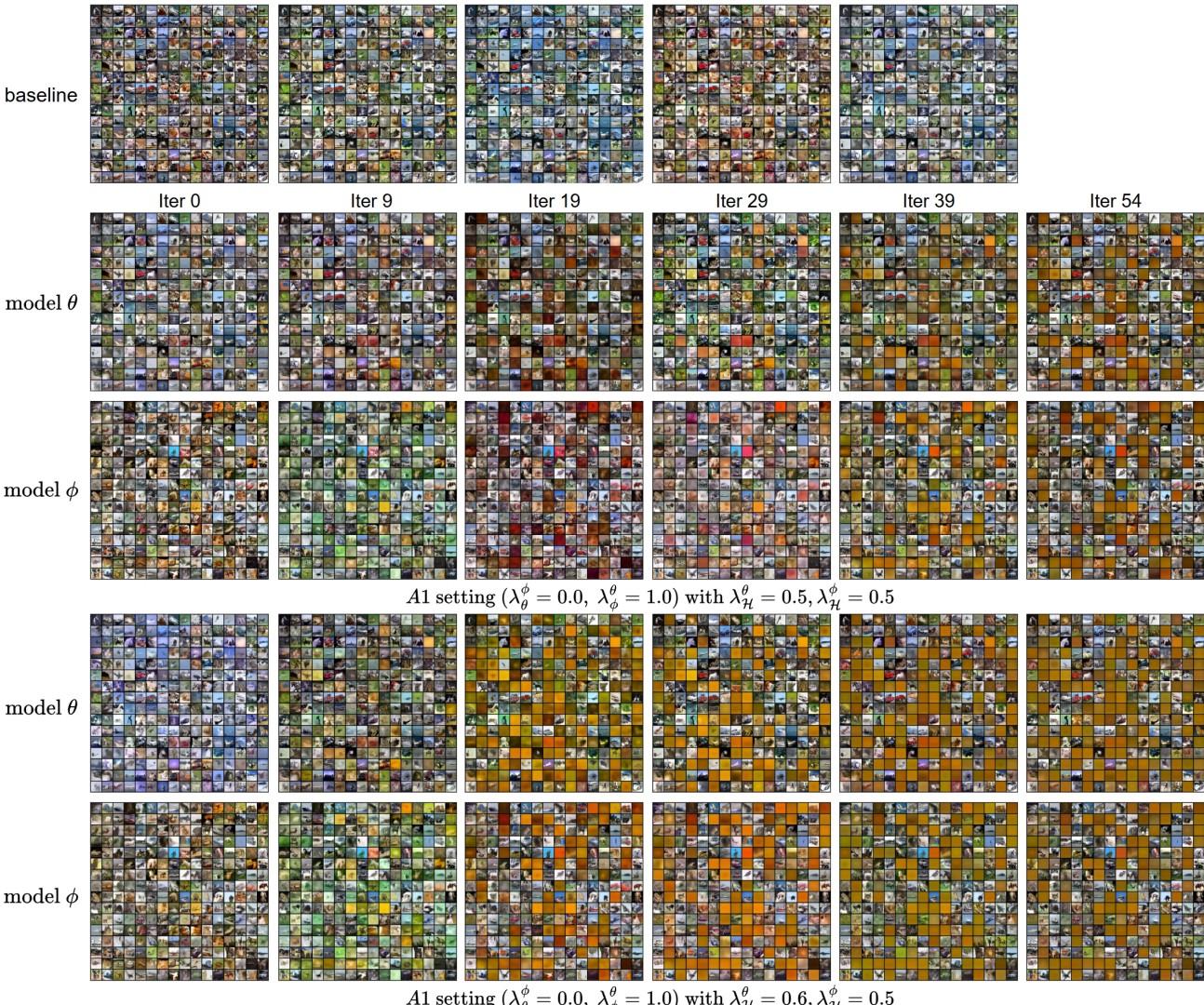

*Figure 7.* Samples generated by model $\theta$ and $\phi$ with different curation and cross-model data proportions. (1) Top: Baseline model output results after each update for 5 iteration. (2) Middle 2 rows: The output results of model $\theta$ and $\phi$ after different rounds of iterative training using 50% curation data and 50% real data under setting $A1$ ($\lambda_\theta^\phi = 0$, $\lambda_\phi^\theta = 1$). (3) Bottom 2 rows: The output results of model $\theta$ and $\phi$ under setting $A1$. Model $\theta$ uses 60% curation data and 40% real data for iterative training, while model $\phi$ uses 50% curation data and 50% real data.

Under the $A5$ setting, only a small fraction of curation data for each model is cross-sourced. The majority of curated samples are still self-generated. Compare to $A1$, model $\theta$'s generated images are therefore less strongly warm-toned, and model $\phi$'s generated images retain more cool-toned characteristics (Figure 8). Moreover, increasing model $\theta$'s curation strength ($\lambda_\mathcal{H}^\theta$) to 0.6 in $A5$ makes more warm-toned samples to be selected and reused for training. As a result, relative to the $A5$ setting with smaller $\lambda_\mathcal{H}^\theta = 0.5$, model $\theta$'s generations shift moderately toward warmer tones, while the cool-toned appearance of model $\phi$'s outputs is substantially attenuated (bottom 2 rows in Figure 8).

For $A6$, the model generated images in Figure 9 after convergence at iteration 54 are not similar to the images generated under other settings, which contain images with strong warm or cool tones approaching pure colors, and the final rewards

$J_p(\theta^*)$, $J_q(\phi^*)$ are relatively small. Due to the strong coupling between the two models ($\lambda_\theta^\phi = 0.7$, $\lambda_\phi^\theta = 0.8$), taking model $\theta$ as an example, its curation samples are largely generated by model $\phi$. Therefore, if model $\theta$ is trained to generate high-reward samples (warm-toned images) through human curation, it requires that model $\phi$ can consistently produce a sufficient number of warm-toned images. However, model $\phi$ is also inversely affected by the images generated from model $\theta$ in its own curation. After the warm-toned samples curated by model $\theta$ are used to train model $\phi$, model $\phi$ tends to generate warm-toned images, but these images are low-scored by $r_\phi$. Therefore, by Eq. (3), the curation selection is almost random, and due to the relatively high ratios of real data ($\lambda_\mathcal{R}^\theta$, $\lambda_\mathcal{R}^\phi \geq 0.4$), warm-toned images cannot be stably generated under the model interactions which leading the generated samples' hue shifts **non-monotonically and oscillates** during the iterative process. After convergence, the system tends to **generate images with tones closer to real data** in Figure 9, resulting in low $J_p(\theta^*)$, $J_q(\phi^*)$ values. In contrast, for $A1$-$A5$, there exists a stable output of curation samples, allowing both models to converge to generating strongly warm/cool-toned images.

Moreover, under the $A6$ setting, when $\lambda_\mathcal{H}^\theta = 0.5$, both models $\theta$ and $\phi$ in the early stage of iterations tend to generate cool-toned images in Figure 9 (Top 2 rows). However, after improving $\lambda_\mathcal{H}^\theta$ to 0.6, since the curation strength of model $\theta$ is stronger, the models tend to generate warm-toned images in the early stage of iterations shown in Figure 9 (Bottom 2 rows).

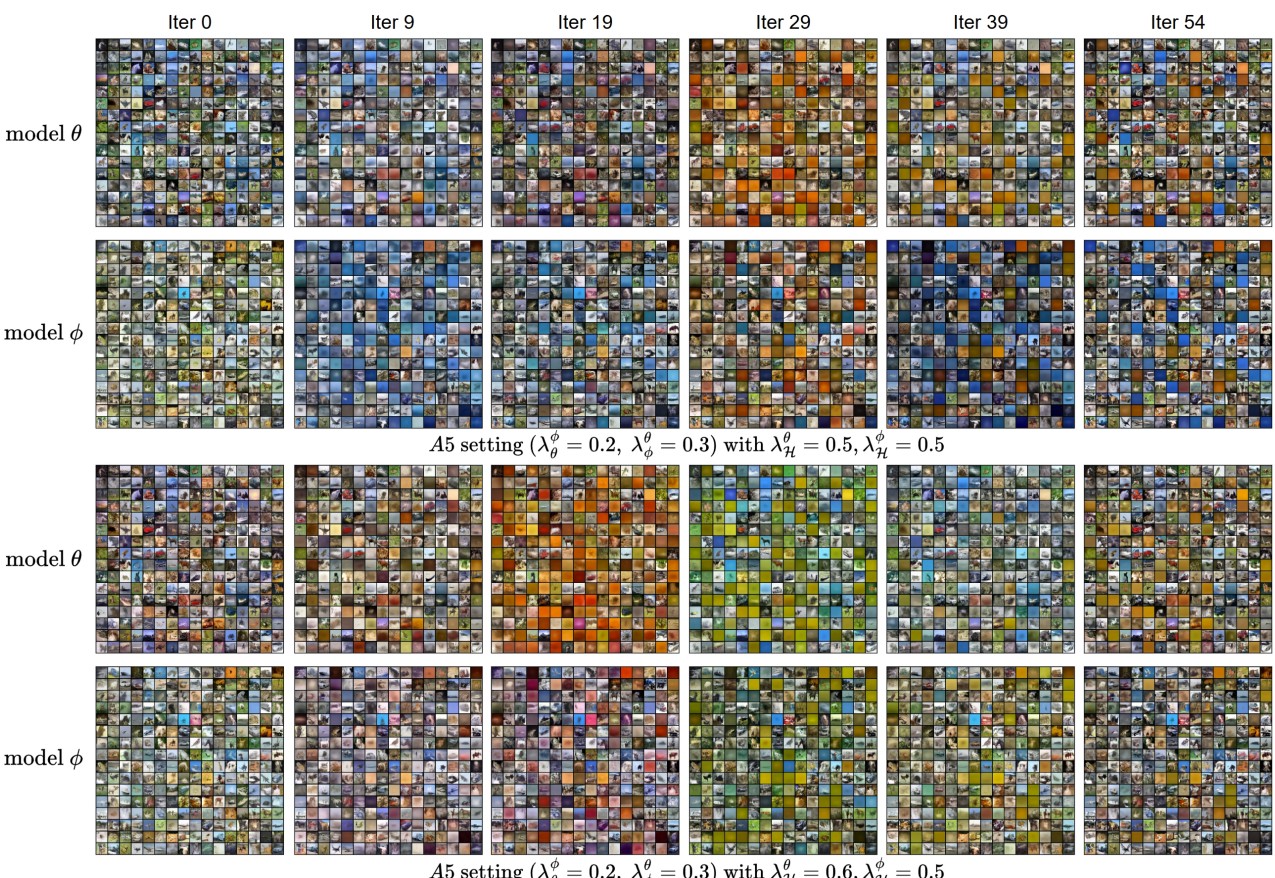

*Figure 8.* Samples generated by model $\theta$ and $\phi$ with different curation and cross-model data proportions. (1) Top 2 rows: The output results of model $\theta$ and $\phi$ after different rounds of iterative training using 50% curation data and 50% real data under setting $A5$ ($\lambda_\theta^\phi = 0.2$, $\lambda_\phi^\theta = 0.3$). (2) Bottom 2 rows: The output results of model $\theta$ and $\phi$ under setting $A5$. Model $\theta$ uses 60% curation data and 40% real data for iterative training, while model $\phi$ uses 50% curation data and 50% real data.

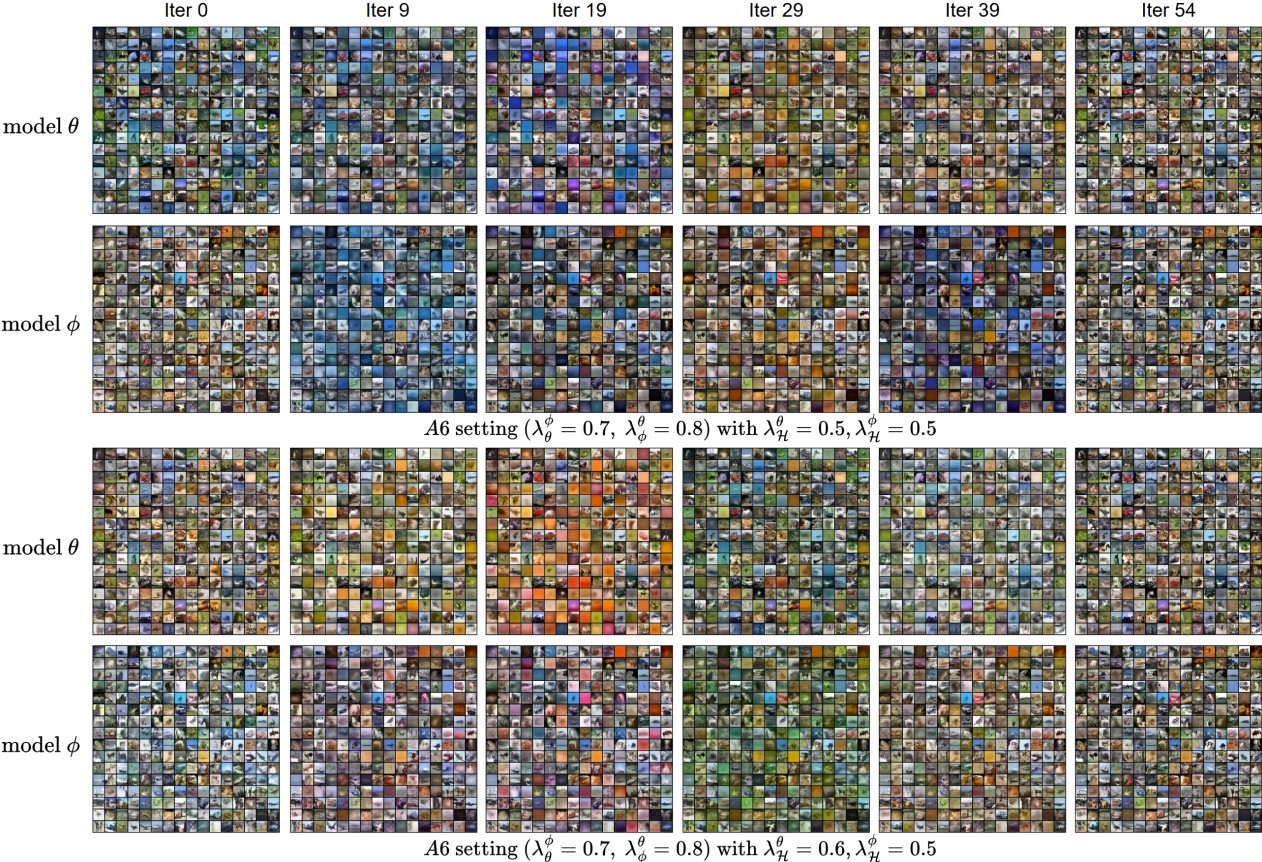

*Figure 9.* Samples generated by model $\theta$ and $\phi$ with different curation and cross-model data proportions. (1) Top 2 rows: The output results of model $\theta$ and $\phi$ after different rounds of iterative training using 50% curation data and 50% real data under setting $A6$ ($\lambda_\theta^\phi = 0.7, \lambda_\phi^\theta = 0.8$). (2) Bottom 2 rows: The output results of model $\theta$ and $\phi$ under setting $A6$. Model $\theta$ uses 60% curation data and 40% real data for iterative training, while model $\phi$ uses 50% curation data and 50% real data.

**D.2. Preference domain mismatch and Qwen2.5-0.5B experiment settings**

In this section, we conduct experiments with language models and observe that *preference domain mismatch* can effectively mask coupling effects.

**Preference domain mismatch.** In our earlier image-model experiments, models $\theta$ and $\phi$ are trained based on the same dataset CIFAR-10, and both models' preferences are tied to image tone. In that setting, conflicting preferences across models directly influence whether the images generated at evaluation stage aligned with human preferences. In practice, however, cross-model data reuse typically requires only that the *data modality* matches, without requiring that the *preference data distributions* to be perfectly identical. For example, in supervised learning, model $\theta$ may be trained to generate brightly colored images of flowers due to human preference and curation, and these generated images may enter a large-scale web corpus. If model $\phi$ instead is trained to generating cool-toned images, it may still scrape and reuse model $\theta$'s images for training, since they are of the correct modality. Although the underlying preferences (bright vs. cool tones) are strongly conflicting, this coupling does not necessarily influence model $\phi$'s rewards ($J_q(\phi_t)$) a lot if model $\phi$'s downstream use case does not require generating flower images. We refer to this phenomenon as *preference domain mismatch*, where strongly conflicting preferences can be present in the training data yet have negligible impact on the evaluated reward due to a mismatch between the preference domain and the evaluation/task domain. In summary, *preference domain mismatch* represents a situation where preference conflicts exist and are coupled, but are not visible to evaluation rewards $J_p, J_q$.

**The relationship between preference domain mismatch and our theory.** This observation is fully consistent with our theory. The intuitive definition of preference domain mismatch in multi-model self-consuming ecosystem is that the data variations driven by the preferences of model $\theta$ mainly occur within a certain "content domain/task domain", but the evaluation scenario of model $\phi$ does not depend on this domain. We consider the extreme case where all the curation data in the training set of model $\theta$ are generated by model $\phi$, which has a preference domain mismatch. Let $\delta\theta^* = \frac{\partial\theta^*}{\partial\lambda_{\mathcal{H}}^\theta}$ for simplicity. For model $\theta$, its curated samples ($\mathcal{H}_t^\theta$) concentrate on regions or feature dimensions that are poorly represented in the evaluation dataset $\mathcal{E}_\theta$, so their induced model parameter updates have limited local sensitivity on the evaluation log-likelihood geometry. Approximately, we assume that

$$\mathbb{E}_{y\sim\mathcal{E}_\theta, x\sim p_{\theta^*}(x|y)}\left[\nabla_\theta \log p_{\theta^*}(x|y)^T\right]\delta\theta^* = \mathbb{E}_{y\sim\mathcal{E}_\theta, x\sim p_{\theta^*}(x|y)}\left[\nabla_\theta \log p_{\theta^*}(x|y)^T\delta\theta^*\right] \approx 0. \tag{12}$$

Denote the Fisher matrix as

$$F_\theta \triangleq \mathbb{E}_{y\sim\mathcal{E}_\theta, x\sim p_{\theta^*}(x|y)}\left[\nabla_\theta \log p_{\theta^*}(x|y)^T\nabla_\theta \log p_{\theta^*}(x|y)\right].$$

By Eq. (12), $\delta\theta^{*T}F_\theta\delta\theta^* \approx 0$. Recall that in Section 4, $\frac{\partial J_p(\theta^*)}{\partial\lambda_{\mathcal{H}}^\theta} = \nabla_\theta J_p(\theta^*)\frac{\partial\theta^*}{\partial\lambda_{\mathcal{H}}^\theta}$, and $\nabla_\theta J_p(\theta^*) = \mathbb{E}_{y\sim\mathcal{E}_\theta, x\sim p_{\theta^*}(x|y)}[r_\theta(x,y)\nabla_\theta \log p_{\theta^*}(x|y)]$. Therefore, by Cauchy-Schwarz inequality,

$$\left|\frac{\partial J_p(\theta^*)}{\partial\lambda_{\mathcal{H}}^\theta}\right| \leq \sqrt{\mathbb{E}_{y\sim\mathcal{E}_\theta, x\sim p_{\theta^*}(x|y)}[r_\theta(x,y)^2]}\sqrt{\delta\theta^{*T}F_\theta\delta\theta^*} \approx 0, \tag{13}$$

which means $\frac{\partial J_p(\theta^*)}{\partial\lambda_{\mathcal{H}}^\theta} \approx 0$.

Eq. (13) formalizes how coupling can be masked with preference alignment mismatch: even if cross-model curation data induces a large parameter response ($\|\delta\theta^*\|$), the reward $J_p$ can be only weakly sensitive to $\lambda_{\mathcal{H}}^\theta$ whenever the "visible component" under the evaluation Fisher metric, $\delta\theta^{*T}F_\theta\delta\theta^*$, is small. With strong preference domain mismatch, the mean-squared projection of the parameter response direction onto the score function space, $\mathbb{E}_{y\sim\mathcal{E}_\theta, x\sim p_{\theta^*}(x|y)}\left[\left(\delta\theta^{*T}\nabla_\theta \log p_{\theta^*}(x|y)\right)^2\right] = \delta\theta^{*T}F_\theta\delta\theta^*$, is sufficiently small–equivalently, $\delta\theta^*$ lies mostly in (or near) the null space of the Fisher matrix.

**Experiment training setting.** We use the pretrained Qwen2.5-0.5B-Instruct (Team, 2024) as the shared base model for both model $\theta$ and $\phi$, with separate LoRA adapters. Our LoRA configuration follows the standard LoRA formulation (Hu et al., 2022) with rank $r = 16$, $\alpha = 32$, *dropout*=0.05.

For model $\theta$, we design it to tend towards summarizing the input text and adopt XSum (Narayan et al., 2018) which pairs long news articles with a short single-sentence summary as $\mathcal{R}^\theta$. Assume model $\phi$ prefers to paraphrase the input text, and $\mathcal{R}^\phi$ is the paraphrasing data from CoEdIT (Raheja et al., 2023), where the text data length is significantly shorter compared to the article data in XSum. The training data size for both models are 1024, and we finetune both models for 13 epochs

with training steps are set to be 512. Models generate synthetic data using temperature $T = 0.9$ and top-$p = 0.95$, and when calculating rewards on evaluation sets, $T = 0$ and top-$p = 1$. The evaluation datasets $\mathcal{E}_\theta, \mathcal{E}_\phi$ for calculating $J_p(\theta_t), J_q(\phi_t)$ are selected from XSum and CoEdIT, respectively, and they are independent of the real data $\mathcal{R}^\theta, \mathcal{R}^\phi$ used for training. Moreover, we keep fixed system prompts throughout training and evaluation:

- model $\theta$: *"You are a helpful English summarization assistant. Write a single-sentence, information-dense summary that is as short as possible. Do not add new facts or embellishments."*

- model $\phi$: *"You are an English paraphrasing assistant. Rewrite the text with different words while maintaining the core meaning. Do not add new facts."*

The iterative retraining process is the same as in Algorithm 1. For model $\theta$, the cross-model training curation inputs are sentence-level, while the evaluation inputs for summarization when calculating $J_p(\theta_t)$ are long articles, leading to a mismatch between the induced training distribution and the target evaluation domain. This mismatch is further reflected in the definitions of rewards below.

**Reward setting.** Assume $x$ is the model input and $y$ is the model output. For model $\theta$, its summarization reward $r_\theta$ is defined as a weighted combination of a length preference term, a keyword-coverage term, and a copying penalty. The length term encourages single-sentence, information-dense summaries by scoring the token length $L(y)$ through $s_{len} = \exp(-\frac{|L(y)-12|}{2})$, where the denominator controls the sharpness of the preference. The coverage term measures whether $y$ contains salient words from the input text, and we take the top-20 most frequent tokens in $x$ and compute the fraction that appear in $y$ as $s_{cov} \in [0, 1]$. To discourage copying, we compute a 4-gram overlap ratio between $x$ and $y$ and get $s_{copy}$. $r_\theta(x, y) = 0.55 s_{len} + 0.55 s_{cov} - 0.6 s_{copy}$.

For model $\phi$, to better align with its paraphrasing objective, we define $r_\phi$ as the weighted sum of a length-ratio constraint $\hat{s}_{len}$, a formality score $\hat{s}_{form}$ and a moderate copying term $\hat{s}_{mid}$ that penalizes both near-verbatim copying and excessive divergence. $\hat{s}_{len} = \exp(-\frac{|\frac{L(y)}{L(x)}-1|}{0.25})$. $\hat{s}_{form}$ is defined based on contraction usage, and we count common English contractions (e.g. "can't", "we're") and tokens ending with "n't". Fewer contractions correspond to more formal writing and let $s_{form} = 1 - \frac{\#contractions}{L(y)}$. For semantic preservation, we compute the same 4-gram copy ratio $s_{copy}$ as model $\theta$ and score it with a kernel centered at 0.2, $\hat{s}_{mid} = \exp(-\frac{|s_{copy}-0.2|}{0.1})$. The paraphrasing reward $r_\phi(x, y) = 0.45 s_{len} + 0.35 s_{form} + 0.3 s_{mid}$.

### D.3. Mechanism validation experiment setting

There are two factors making computing mechanism results in Section 4 on large models difficult: sampling error and statistical error from the approximation methods to estimate high-dimensional matrices. Since theory-related matrices are local and involve second-order information, this is closely related to the well-known difficulty of estimating inverse-Hessian-based quantities in large models. Prior works treat scalable and statistically stable estimation of such objects as a nontrivial problem (Li et al., 2025; Basu et al., 2021). Therefore, to validate the mechanism more detailed, we extend Example 4.6 in Section 5.

Recall that the mixture distributions $P = (1 - \lambda_\mathcal{H}^\theta)\mathcal{N}(\phi, \sigma^2 I) + \lambda_\mathcal{H}^\theta \mathcal{N}(\phi + a, \sigma^2 I)$ and $Q = \mathcal{N}(A(t)\theta, \sigma^2 I)$ where $A(t) \in \mathbb{R}^{24 \times 24}$ is a block-diagonal matrix with 12 $2 \times 2$ blocks. The $i$-th block $A_i(t) = t\beta_i R_i$ where $t \in [0.05, 1]$ is the coupling scale, $\beta_i \in [0.08, 0.95]$ controls the interaction magnitude of block $i$, and

$$R_i = \begin{pmatrix} \cos x_i, & -\sin x_i \\ \sin x_i, & \cos x_i \end{pmatrix}, \{x_i\} = \{-80°, -66°, -52°, -38°, -24°, -10°, 5°, 21°, 37°, 53°, 69°, 85°\}$$

is a 2-dimensional matrix introduces local geometric heterogeneity. $a = [a_1, ..., a_{12}] \in \mathbb{R}^{24}$ and $a_i = 0.9^i (1 + 0.2 \sin \frac{2\pi(i-1)}{11}) \begin{pmatrix} \cos(\frac{140(i-1)}{11} - 70)°, & -\sin(\frac{140(i-1)}{11} - 70)° \\ \sin(\frac{140(i-1)}{11} - 70)°, & \cos(\frac{140(i-1)}{11} - 70)° \end{pmatrix} (1, 2)^T$. We set the reward expectations $J_p(\theta) = g_p^T \theta - \frac{\eta_p \|\theta\|^2}{2}$ with $\eta_p = 0.18$, $J_q(\phi) = g_q^T \phi - \frac{\eta_q \|\phi\|^2}{2}$ with $\eta_q = 0.22$, $g_p = (g_{p1}, ..., g_{p12}) \in \mathbb{R}^{24}$ and $g_{pi} = 0.9^i (1 + 0.15 \cos \frac{1.5\pi(i-1)}{11}) \begin{pmatrix} \cos(\frac{80(i-1)}{11} - 55)°, & -\sin(\frac{80(i-1)}{11} - 55)° \\ \sin(\frac{80(i-1)}{11} - 55)°, & \cos(\frac{80(i-1)}{11} - 55)° \end{pmatrix} (1, 0)^T$, $g_q = (g_{q1}, ..., g_{q12}) \in \mathbb{R}^{24}$ and $g_{qi} =$

$0.9^i\left(1+0.18\sin(\frac{1.5\pi(i-1)}{11}+0.3)\right)\begin{pmatrix}\cos(\frac{100(i-1)}{11}-40)^\circ, -\sin(\frac{100(i-1)}{11}-40)^\circ \\ \sin(\frac{100(i-1)}{11}-40)^\circ, \cos(\frac{100(i-1)}{11}-40)^\circ\end{pmatrix}(1,4)^T$. For simplicity, since the total iteration number is 100, we use the mean value of model parameters after the 80th iteration as the approximation for $\theta^*$ and $\phi^*$. Moreover, we build the system including $a, g_p, g_q, R$ by repeating a common two dimensional coupling template across blocks. $R_i$ is set to be an orthogonal rotation, while $a_i, g_{pi}, g_{qi}$ are matched blockwise curation and reward directions. This yields a multi-mode coupled system that remains analytically tractable, cleanly separates coupling magnitude from geometric distortion.

Based on the blockwise Gaussian settings, it's easy to check that the self-influence $\langle\nabla_\theta J_p(\theta^*), S_p\mathbb{E}_{P_H}[-\nabla_\theta\ell_\theta(\theta^*)]\rangle = \sum_{i=1}^{12}\langle\nabla_{\theta_i}J_p(\theta^*), S_{p,i}\mathbb{E}_{P_H}[-\nabla_{\theta_i}\ell_\theta(\theta^*)]\rangle$ where $\theta_i$, $S_{p,i}$ are the components of $\theta$ and $S_p$ within specific dimension blocks, respectively, and $\theta = (\theta_1,...,\theta_{12})$, $S_p = diag(S_{p,1},...,S_{p,12})$, $\nabla_\theta J_p(\theta^*) = (\nabla_{\theta_1}J_p(\theta^*),...,\nabla_{\theta_{12}}J_p(\theta^*))$ and $\mathbb{E}_{P_H}[-\nabla_\theta\ell_\theta(\theta^*)] = (\mathbb{E}_{P_H}[-\nabla_{\theta_1}\ell_\theta(\theta^*)],...,\mathbb{E}_{P_H}[-\nabla_{\theta_{12}}\ell_\theta(\theta^*)])$. Similarly,

$$\langle\nabla_\theta J_p(\theta^*), \mathbb{E}_{P_H}[-\nabla_\theta\ell_\theta(\theta^*)]\rangle = \sum_{i=1}^{12}\langle\nabla_{\theta_i}J_p(\theta^*), \mathbb{E}_{P_H}[-\nabla_{\theta_i}\ell_\theta(\theta^*)]\rangle,$$

$$\langle\nabla_\phi J_q(\phi^*), C_q\mathbb{E}_{P_H}[-\nabla_\theta\ell_\theta(\theta^*)]\rangle = \sum_{i=1}^{12}\langle\nabla_{\phi_i}J_q(\phi^*), C_{q,i}\mathbb{E}_{P_H}[-\nabla_{\theta_i}\ell_\theta(\theta^*)]\rangle,$$

$$\langle\nabla_\phi J_q(\phi^*), S_q C_q\mathbb{E}_{P_H}[-\nabla_\theta\ell_\theta(\theta^*)]\rangle = \sum_{i=1}^{12}\langle\nabla_{\phi_i}J_q(\phi^*), S_{q,i}C_{q,i}\mathbb{E}_{P_H}[-\nabla_{\theta_i}\ell_\theta(\theta^*)]\rangle.$$

In Figure 3(E)(F), the value corresponding to the $i$-th dimension block on the horizontal axis corresponds to the $i$-th term in the above equations.

## E. Theory Extension And Generalization

First, in the main paper, our theory is based on the assumption that the outputs of models are each other's inputs, $x = p_\theta(x|y)$ and $y = q_\phi(y|x)$. It's easy to verify that our theory can be extended to simpler unsupervised learning multi-model systems or systems where the input and output of both models are the same, $x = p_\theta(x|y)$ and $x = q_\phi(x|y)$. Next, we discuss extensions of some theory results presented in the main paper.

### E.1. Alignment improvement in the single-model scenario

**Corollary E.1.** *For a single model $\theta$, under Assumptions 3.2-3.5, if $\rho_p \approx 1$, then $\frac{\partial J_p(\theta^*)}{\partial\lambda_\mathcal{H}^\theta} > 0$.*

### E.2. Sufficient conditions related to cross-model influence

**Corollary E.2.** *Denote $\tau_q = \gamma_\phi - L_\phi\varepsilon_\phi - \frac{L_\theta\varepsilon_\theta L_\phi\varepsilon_\phi}{\gamma_\theta - L_\theta\varepsilon_\theta}$ and let $m_q$ be the minimal eigenvalue of $\frac{S_q + S_q^T}{2}$, we have $\|S_q\| \leq \frac{1}{\tau_q}$ and if $|\rho_q| > \frac{1}{\sqrt{1+m_q^2\tau_q^2}}$, then $\text{sign}(\rho_q) \cdot \frac{\partial J_q(\beta^*)}{\partial\lambda_\mathcal{H}^\theta} < 0$.*

### E.3. The extension of Theorem 4.5

For the curation ratio $\lambda_\mathcal{H}^\theta$, changing the size of real dataset, synthetic dataset, or curated synthetic dataset will influence $\lambda_\mathcal{H}^\theta$. Different combinations of such modifications lead to different forms of equations in Theorem 4.5. For instance, varying the number of real data samples changes $\lambda_\mathcal{H}^\theta$, but the expression in Theorem 4.5 contains no term explicitly associated with real dataset $\mathcal{R}^\theta$. Mathematically, if we regard $J_p$ as a multivariate function of $\lambda_\mathcal{H}^\theta, \lambda_\mathcal{R}^\theta, \lambda_\mathcal{S}^\theta$ subject to the constraint $\lambda_\mathcal{H}^\theta + \lambda_\mathcal{R}^\theta + \lambda_\mathcal{S}^\theta = 1$, then each data-modification scheme corresponds to a particular definition of the total derivative $\frac{\partial J_p}{\partial\lambda_\mathcal{H}^\theta}$.

Generally, we suppose the variation in $\lambda_\mathcal{H}^\theta$ in practice is induced by adjusting the dataset sizes of $\mathcal{R}^\theta, \mathcal{S}^\theta, \mathcal{H}^\theta$ in fixed proportions $a_r, a_s, a_h$ and $a_r + a_s + a_h = 1$. For example, adding $n$ training samples means adding $a_r n$ real data samples, $a_s n$ synthetic data samples and $a_h n$ curated synthetic samples. If any of $a_r, a_s, a_h$ is negative, the corresponding operation represents removing samples rather than adding them. **Note that the setting $a_r = 0, a_s = 0, a_h = 1$ is equivalent to the scenario considered in the main paper.** By varying $(a_r, a_s, a_h)$, this parameterization covers all possible cases.

Under this parameterization, we can get the following theorem, which extends and generalized Theorem 4.5. Notably, the $S_p, S_q, C_q$ matrices remain unchanged, highlighting the generality of our methods.

**Theorem E.3.** [***Generalization of Theorem 4.5***] *Under the parameterization of $(a_r, a_s, a_h)$ and Assumptions 3.2-3.5 and condition in Theorem 3.6, the self-consuming multi-model system converges. Let $P_{\mathcal{H}}, P_{\mathcal{S}}$ be the distributions of human-curation data $\mathcal{H}_t^\theta$ and synthetic self-consuming data $\mathcal{S}_t^\theta$ after convergence, respectively, and we have*

$$\frac{\partial J_p(\theta^*)}{\partial \lambda_{\mathcal{H}}^\theta} = \frac{1}{a_h - \lambda_{\mathcal{H}}^\theta} \left\langle \nabla_\theta J_p(\theta^*), S_p \left[ a_s \mathbb{E}_{z \sim P_{\mathcal{S}}} \nabla_\theta \ell_\theta(\theta^*) + a_h \mathbb{E}_{z \sim P_{\mathcal{H}}} \nabla_\theta \ell_\theta(\theta^*) + a_r \mathbb{E}_{z \sim \mathcal{R}^\theta} \nabla_\theta \ell_\theta(\theta^*) \right] \right\rangle,$$

$$\frac{\partial J_q(\phi^*)}{\partial \lambda_{\mathcal{H}}^\theta} = \frac{1}{a_h - \lambda_{\mathcal{H}}^\theta} \left\langle \nabla_\phi J_q(\phi^*), S_q C_q \left[ a_s \mathbb{E}_{z \sim P_{\mathcal{S}}} \nabla_\theta \ell_\theta(\theta^*) + a_h \mathbb{E}_{z \sim P_{\mathcal{H}}} \nabla_\theta \ell_\theta(\theta^*) + a_r \mathbb{E}_{z \sim \mathcal{R}^\theta} \nabla_\theta \ell_\theta(\theta^*) \right] \right\rangle.$$

*Proof.* In this proof, we adopt the same notation as in the proof of Theorem 4.5 for simplicity. See Appendix F.4 for details. The overall proof strategy mirrors that of Theorem 4.5, with the sole difference is how to compute $\frac{df_\theta}{d\lambda_{\mathcal{H}}^\theta}$ where $f_\theta$ is the density function of model $\theta$'s mixture distribution $P$. Notice that the total derivative of $f_\theta$ is

$$df_\theta(\lambda_{\mathcal{S}}^\theta, \lambda_{\mathcal{H}}^\theta, \lambda_{\mathcal{R}}^\theta; z, \theta, \phi) = f_{\theta,s}(z; \theta, \phi) d\lambda_{\mathcal{S}}^\theta + f_{\theta,c}(z; \theta, \phi) d\lambda_{\mathcal{H}}^\theta + f_{\theta,r}(z) d\lambda_{\mathcal{R}}^\theta, \ d\lambda_{\mathcal{S}}^\theta + d\lambda_{\mathcal{H}}^\theta + d\lambda_{\mathcal{R}}^\theta = 0.$$

Under the parameterization of $(a_r, a_s, a_h)$, it's easy to check that $d\lambda_{\mathcal{R}}^\theta = \frac{a_r - \lambda_{\mathcal{R}}^\theta}{a_h - \lambda_{\mathcal{H}}^\theta} d\lambda_{\mathcal{H}}^\theta$ and $d\lambda_{\mathcal{S}}^\theta = \frac{a_s - \lambda_{\mathcal{S}}^\theta}{a_h - \lambda_{\mathcal{H}}^\theta} d\lambda_{\mathcal{H}}^\theta$. These two equations align with the results $d\lambda_{\mathcal{S}}^\theta = -\frac{\lambda_{\mathcal{S}}^\theta}{\lambda_{\mathcal{S}}^\theta + \lambda_{\mathcal{R}}^\theta} d\lambda_{\mathcal{H}}^\theta$, $d\lambda_{\mathcal{R}}^\theta = -\frac{\lambda_{\mathcal{R}}^\theta}{\lambda_{\mathcal{S}}^\theta + \lambda_{\mathcal{R}}^\theta} d\lambda_{\mathcal{H}}^\theta$ in the proof of Theorem 4.5 if $a_r = a_s = 0$ and $a_h = 1$. Therefore, similar to Eq. (31),

$$\frac{\partial F_p}{\partial \lambda_{\mathcal{H}}^\theta} = \int \nabla_\theta \ell_\theta(z; \theta^*) \frac{df_\theta(\lambda_{\mathcal{S}}^\theta, \lambda_{\mathcal{H}}^\theta, \lambda_{\mathcal{R}}^\theta; z, \theta^*, \phi^*)}{d\lambda_{\mathcal{H}}^\theta} dz$$

$$= \int \nabla_\theta \ell_\theta(z; \theta^*) \left( \frac{a_s - \lambda_{\mathcal{S}}^\theta}{a_h - \lambda_{\mathcal{H}}^\theta} f_{\theta,s}(z; \theta^*, \phi^*) + f_{\theta,c}(z; \theta^*, \phi^*) + \frac{a_r - \lambda_{\mathcal{R}}^\theta}{a_h - \lambda_{\mathcal{H}}^\theta} f_{\theta,r}(z) \right) dz$$

$$= \int \nabla_\theta \ell_\theta(z; \theta^*) \left( \frac{a_s}{a_h - \lambda_{\mathcal{H}}^\theta} f_{\theta,s}(z; \theta^*, \phi^*) + \frac{a_h}{a_h - \lambda_{\mathcal{H}}^\theta} f_{\theta,c}(z; \theta^*, \phi^*) + \frac{a_r}{a_h - \lambda_{\mathcal{H}}^\theta} f_{\theta,r}(z) \right.$$

$$\left. - \frac{\lambda_{\mathcal{S}}^\theta f_{\theta,s}(z; \theta^*, \phi^*) + \lambda_{\mathcal{H}}^\theta f_{\theta,c}(z; \theta^*, \phi^*) + \lambda_{\mathcal{R}}^\theta f_{\theta,r}(z)}{a_h - \lambda_{\mathcal{H}}^\theta} \right) dz$$

$$= \frac{1}{a_h - \lambda_{\mathcal{H}}^\theta} \left[ a_s \mathbb{E}_{z \sim P_{\mathcal{S}}} \nabla_\theta \ell_\theta(\theta^*) + a_h \mathbb{E}_{z \sim P_{\mathcal{H}}} \nabla_\theta \ell_\theta(\theta^*) + a_r \mathbb{E}_{z \sim \mathcal{R}^\theta} \nabla_\theta \ell_\theta(\theta^*) \right]. \tag{14}$$

Therefore, similar to the left parts in the proof of Theorem 4.5, we can prove this theorem. □

Symmetrically, if we vary only the curation strength of model $\phi$ ($\lambda_{\mathcal{H}}^\phi$), the following theorem characterizes both its local self-influence and cross-influence on the final rewards.

**Theorem E.4.** [***Symmetrical version of $\lambda_{\mathcal{H}}^\phi$***] *Under the parameterization of $(a_r, a_s, a_h)$ and Assumptions 3.2-3.5 and condition in Theorem 3.6, the self-consuming multi-model system converges. Let $Q_{\mathcal{H}}, Q_{\mathcal{S}}$ be the distributions of human-curation data $\mathcal{H}_t^\phi$ and synthetic self-consuming data $\mathcal{S}_t^\phi$ after convergence, respectively, and we have*

$$\frac{\partial J_q(\phi^*)}{\partial \lambda_{\mathcal{H}}^\phi} = \frac{1}{a_h - \lambda_{\mathcal{H}}^\phi} \left\langle \nabla_\phi J_q(\phi^*), S_q \left[ a_s \mathbb{E}_{z \sim Q_{\mathcal{S}}} \nabla_\phi \ell_\phi(\phi^*) + a_h \mathbb{E}_{z \sim Q_{\mathcal{H}}} \nabla_\phi \ell_\phi(\phi^*) + a_r \mathbb{E}_{z \sim \mathcal{R}^\phi} \nabla_\phi \ell_\phi(\phi^*) \right] \right\rangle,$$

$$\frac{\partial J_p(\theta^*)}{\partial \lambda_{\mathcal{H}}^\phi} = \frac{1}{a_h - \lambda_{\mathcal{H}}^\phi} \left\langle \nabla_\theta J_p(\theta^*), S_p C_p \left[ a_s \mathbb{E}_{z \sim Q_{\mathcal{S}}} \nabla_\phi \ell_\phi(\phi^*) + a_h \mathbb{E}_{z \sim Q_{\mathcal{H}}} \nabla_\phi \ell_\phi(\phi^*) + a_r \mathbb{E}_{z \sim \mathcal{R}^\phi} \nabla_\phi \ell_\phi(\phi^*) \right] \right\rangle.$$

In fact, the differences among these data-modification schemes in their impact on $\left\{ \frac{\partial J_i}{\partial \lambda_{\mathcal{H}}^j}, i \in \{\theta, \phi\}, \ j \in \{\theta, \phi\} \right\}$ are attributable to how they compose model parameter update directions induced by different datasets $\mathcal{R}^j, \mathcal{S}^j, \mathcal{H}^j, j \in \{\theta, \phi\}$. Under each scheme, the resulting update is a combination of the per-dataset update directions, with weights given exactly by the parameterization $(a_r, a_s, a_h)$.

**Extend to any mixing weight.** Prior results focus on the curation proportion $\lambda_{\mathcal{H}}^\phi$, as our analysis targets how human curation influences preference alignments. More generally, our framework can be extended to any mixture weight. Specifically, under the same parametrization $(a_r, a_s, a_h)$, we have the following theorem.

**Theorem E.5.** *[Extension to any mixture weight]* *Under the parameterization of $(a_r, a_s, a_h)$ and Assumptions 3.2-3.5 and condition in Theorem 3.6, the self-consuming multi-model system converges. Let $P_{\mathcal{H}}, P_{\mathcal{S}}$ be the distributions of human-curation data $\mathcal{H}_t^\theta$ and synthetic self-consuming data $\mathcal{S}_t^\theta$ after convergence, respectively. Let $Q_{\mathcal{H}}, Q_{\mathcal{S}}$ be the distributions of human-curation data $\mathcal{H}_t^\phi$ and synthetic self-consuming data $\mathcal{S}_t^\phi$ after convergence, respectively. If only changing the mixture weights of model $\theta$, then for any $(\lambda, a) \in \{(\lambda_{\mathcal{S}}^\theta, a_s), (\lambda_{\mathcal{R}}^\theta, a_r)\}$,*

$$\frac{\partial J_p(\theta^*)}{\partial \lambda} = \frac{1}{a - \lambda} \left\langle \nabla_\theta J_p(\theta^*), S_p \left[ a_s \mathbb{E}_{z \sim P_{\mathcal{S}}} \nabla_\theta \ell_\theta(\theta^*) + a_h \mathbb{E}_{z \sim P_{\mathcal{H}}} \nabla_\theta \ell_\theta(\theta^*) + a_r \mathbb{E}_{z \sim \mathcal{R}^\theta} \nabla_\theta \ell_\theta(\theta^*) \right] \right\rangle,$$

$$\frac{\partial J_q(\phi^*)}{\partial \lambda} = \frac{1}{a - \lambda} \left\langle \nabla_\phi J_q(\phi^*), S_q C_q \left[ a_s \mathbb{E}_{z \sim P_{\mathcal{S}}} \nabla_\theta \ell_\theta(\theta^*) + a_h \mathbb{E}_{z \sim P_{\mathcal{H}}} \nabla_\theta \ell_\theta(\theta^*) + a_r \mathbb{E}_{z \sim \mathcal{R}^\theta} \nabla_\theta \ell_\theta(\theta^*) \right] \right\rangle.$$

*If only changing the mixture weights of model $\phi$, then for any $(\lambda, a) \in \{(\lambda_{\mathcal{S}}^\phi, a_s), (\lambda_{\mathcal{R}}^\phi, a_r)\}$,*

$$\frac{\partial J_q(\phi^*)}{\partial \lambda} = \frac{1}{a - \lambda} \left\langle \nabla_\phi J_q(\phi^*), S_q \left[ a_s \mathbb{E}_{z \sim Q_{\mathcal{S}}} \nabla_\phi \ell_\phi(\phi^*) + a_h \mathbb{E}_{z \sim Q_{\mathcal{H}}} \nabla_\phi \ell_\phi(\phi^*) + a_r \mathbb{E}_{z \sim \mathcal{R}^\phi} \nabla_\phi \ell_\phi(\phi^*) \right] \right\rangle,$$

$$\frac{\partial J_p(\theta^*)}{\partial \lambda} = \frac{1}{a - \lambda} \left\langle \nabla_\theta J_p(\theta^*), S_p C_p \left[ a_s \mathbb{E}_{z \sim Q_{\mathcal{S}}} \nabla_\phi \ell_\phi(\phi^*) + a_h \mathbb{E}_{z \sim Q_{\mathcal{H}}} \nabla_\phi \ell_\phi(\phi^*) + a_r \mathbb{E}_{z \sim \mathcal{R}^\phi} \nabla_\phi \ell_\phi(\phi^*) \right] \right\rangle.$$

Based on Theorem E.3, E.4 and E.5, we can obtain the expressions of the partial derivatives of $J_p$ and $J_q$ with respect to different mixing weights. Therefore, we have the Jacobian of $J_p$ and $J_q$.

**Extend to cross-model-generated fractions.** Similarly, the above analysis can be extended to variations in cross-model proportions $\lambda_\theta^\phi$ and $\lambda_\phi^\theta$.

### E.4. The extension of Theorem 4.8

From the proof of Theorem 4.8 in in Appendix F.9, for any data proportion $\lambda^j \in \{\lambda_{\mathcal{S}}^j, \lambda_{\mathcal{H}}^j, \lambda_{\mathcal{R}}^j\}, j \in \{\theta, \phi\}$ and fixed constants $c, d \in [0, 1]$, we can estimate the upper bound of $\left| J_p\left(\theta^*(\lambda^\theta = c)\right) - J_p\left(\theta^*(\lambda^\theta = d)\right) \right|$ and $\left| J_q\left(\phi^*(\lambda^\phi = c)\right) - J_q\left(\phi^*(\lambda^\phi = d)\right) \right|$, as long as $J_p(\theta^*)$ and $J_q(\phi^*)$ are well-defined for $\lambda^\theta$ and $\lambda^\phi$, respectively, taking values at $c$ and $d$, which means that the system will converge to the stable point $(\theta^*, \phi^*)$ with $\lambda^\theta \in \{c, d\}$ and $\lambda^\phi \in \{c, d\}$. In the main paper, we choose $\lambda^\theta = \lambda_{\mathcal{H}}^\theta, \lambda^\phi = \lambda_{\mathcal{H}}^\phi$ as a special case. Theorem 4.8 and its extensions demonstrate that in multi-model self-consuming systems with model interactions, if the system can converge to a stable point after iterative retraining, the final evaluation rewards for each model $J_p(\theta^*), J_q(\phi^*)$ will not change significantly due to variations in the proportion of data sources.

Moreover, according to the theorem proof in Appendix F.9, the upper bound is related to models and rewards architecture, the loss functions, the real datasets, and model interaction strength. The upper bounds also decreases as the proportion of real data increases.

## F. Proofs

**Lemma F.1.** *(Kantorovich-Rubinstein)* *A distribution map $D(m)$, $m \in \mathcal{M}$ is $\varepsilon$-sensitive if and only if for any $m, m' \in \mathcal{M}$:*

$$\sup_{g:\mathbb{R}^n \to \mathbb{R}, g \in Lip^1} \left| \mathbb{E}_{Z \sim D(m)} g(Z) - \mathbb{E}_{Z \sim D(m')} g(Z) \right| \le \varepsilon \|m - m'\|.$$

### F.1. Proof of Theorem 3.6

**Theorem 3.6.** *Define $\kappa \triangleq \max \left( \frac{\gamma_\theta L_\phi \varepsilon_\phi + 2\gamma_\phi L_\theta \varepsilon_\theta}{\gamma_\phi (\gamma_\theta - L_\theta \varepsilon_\theta)}, \frac{\gamma_\phi L_\theta \varepsilon_\theta + 2\gamma_\theta L_\phi \varepsilon_\phi}{\gamma_\theta (\gamma_\phi - L_\phi \varepsilon_\phi)}, \frac{L_\theta \varepsilon_\theta}{\gamma_\theta} + \frac{L_\phi \varepsilon_\phi}{\gamma_\phi} \right)$. Under Assumptions 3.2, 3.3, and 3.5, if $\kappa < 1$, then the stable point $(\theta^*, \phi^*)$ exists, and iterative training loop (4) will drive $(\theta_t, \phi_t)$ to converge to $(\theta^*, \phi^*)$ at a linear rate:*

$$\|(\theta_t, \phi_t) - (\theta^*, \phi^*)\| \le \kappa^t \|(\theta_0, \phi_0) - (\theta^*, \phi^*)\|.$$

*Proof.* Since $P, Q$ are $\varepsilon_\theta, \varepsilon_\phi$-sensitive (Assump. 3.5), by its definition and Kantorovich-Rubinstein (Lemma F.1), we have

$$\begin{aligned} \left| \mathbb{E}_{z \sim P(\theta, \phi)} g(z) - \mathbb{E}_{z \sim P(\theta', \phi')} g(z) \right| &\le \varepsilon_\theta \|(\theta, \phi) - (\theta', \phi')\|, \\ \left| \mathbb{E}_{z \sim Q(\theta, \phi)} g(z) - \mathbb{E}_{z \sim Q(\theta', \phi')} g(z) \right| &\le \varepsilon_\phi \|(\theta, \phi) - (\theta', \phi')\|, \end{aligned} \tag{15}$$

for $\forall\, \theta, \phi, \theta', \phi'$ and $g \in Lip^1 : \mathbb{R}^n \to \mathbb{R}$ the unions of all 1-Lipchitz functions. Let $z = (x, y)$ be the data pair and notice that the loss gradients $\nabla_\theta \ell_\theta(z), \nabla_\phi \ell_\phi(z)$ are $L_\theta, L_\phi$-Lipschitz in $z$ for any $\theta, \phi$, respectively (Assump. 3.3), so for any vector $v$ fixed,

$$\left| \frac{\nabla_\theta \ell_\theta(z_1) v^T}{L_\theta} - \frac{\nabla_\theta \ell_\theta(z_2) v^T}{L_\theta} \right| \leq \|v\| \frac{\|\nabla_\theta \ell_\theta(z_1) - \nabla_\theta \ell_\theta(z_2)\|}{|L_\theta|} \leq \|v\| \|z_1 - z_2\|.$$

Therefore, $\frac{\nabla_\theta \ell_\theta(z) v^T}{L_\theta \|v\|}$ and $\frac{\nabla_\phi \ell_\phi(z) v^T}{L_\phi \|v\|}$ are 1-Lipschitz in $z$ for $\forall\, \theta, \phi$ and fixed $v$. Replace $g$ with these functions in Eq. (15) and for simplicity, let

$$R_p(\theta, P(\hat\theta, \hat\phi)) = \mathbb{E}_{z \sim P(\hat\theta, \hat\phi)} \ell_\theta(z), \ R_q(\phi, Q(\hat\theta, \hat\phi)) = \mathbb{E}_{z \sim Q(\hat\theta, \hat\phi)} \ell_\phi(z). \tag{16}$$

Then, for any fixed distributions $P(\theta_1, \phi_1), P(\theta_2, \phi_2), Q(\theta_1, \phi_1), Q(\theta_2, \phi_2)$, fixed vector $v$ and $\forall\, \theta, \phi$,

$$\begin{aligned}
\left\| \left( \nabla_\theta R_p(\theta, P(\theta_1, \phi_1)) - \nabla_\theta R_p(\theta, P(\theta_2, \phi_2)) \right) v^T \right\| &\leq L_\theta \varepsilon_\theta \|v\| \|(\theta_1, \phi_1) - (\theta_2, \phi_2)\|, \\
\left\| \left( \nabla_\phi R_q(\phi, Q(\theta_1, \phi_1)) - \nabla_\phi R_q(\phi, Q(\theta_2, \phi_2)) \right) v^T \right\| &\leq L_\phi \varepsilon_\phi \|v\| \|(\theta_1, \phi_1) - (\theta_2, \phi_2)\|,
\end{aligned} \tag{17}$$

First, we prove that for any fixed distributions $P(\theta_1, \phi_1), P(\theta_2, \phi_2)$,

$$\left\| \underset{\theta}{\mathrm{argmin}}\, R_p(\theta, P(\theta_1, \phi_1)) - \underset{\theta}{\mathrm{argmin}}\, R_p(\theta, P(\theta_2, \phi_2)) \right\| \leq \frac{L_\theta \varepsilon_\theta}{\gamma_\theta} \|(\theta_1, \phi_1) - (\theta_2, \phi_2)\|.$$

Notice that $R_p(\theta, P(\theta_1, \phi_1))$ is $\gamma_\theta$-strongly convex in $\theta$ for any fixed $\theta_1, \phi_1$, and this can be derived by taking expectations of both sides in Eq. (6) (Assump. 3.2). There exist minimal points $\varphi_1, \varphi_2$ so that $\nabla_\theta R_p(\theta, P(\theta_1, \phi_1))\big|_{\theta = \varphi_1} = \nabla_\theta R_p(\theta, P(\theta_2, \phi_2))\big|_{\theta = \varphi_2} = 0$. Therefore,

$$0 = \left( \nabla_\theta R_p(\varphi_1, P(\theta_1, \phi_1)) - \nabla_\theta R_p(\varphi_2, P(\theta_1, \phi_1)) \right) + \nabla_\theta R_p(\varphi_2, P(\theta_1, \phi_1)) - \nabla_\theta R_p(\varphi_2, P(\theta_2, \phi_2)). \tag{18}$$

Combined with strongly convexity, we have

$$\begin{aligned}
\nabla_\theta R_p(\varphi_1, P(\theta_1, \phi_1)) - \nabla_\theta R_p(\varphi_2, P(\theta_1, \phi_1)) &\geq \nabla_\theta R_p(\varphi_2, P(\theta_1, \phi_1))^T (\varphi_1 - \varphi_2) + \frac{\gamma_\theta}{2} \|\varphi_1 - \varphi_2\|^2 \\
\nabla_\theta R_p(\varphi_2, P(\theta_1, \phi_1)) - \nabla_\theta R_p(\varphi_1, P(\theta_1, \phi_1)) &\geq \frac{\gamma_\theta}{2} \|\varphi_1 - \varphi_2\|^2 = \frac{\gamma_\theta}{2} \|\varphi_1 - \varphi_2\|^2,
\end{aligned} \tag{19}$$

the second line of the formula is due to $\nabla_\theta R_p(\varphi_1, P(\theta_1, \phi_1))^T (\varphi_2 - \varphi_1) = 0$. Add the two formulas in Eq. (19) and get $0 \geq -\gamma_\theta \|\varphi_1 - \varphi_2\|^2 \geq (\varphi_1 - \varphi_2)^T \nabla_\theta R_p(\varphi_2, P(\theta_1, \phi_1)) = (\varphi_1 - \varphi_2)^T \left( \nabla_\theta R_p(\varphi_2, P(\theta_1, \phi_1)) - \nabla_\theta R_p(\varphi_1, P(\theta_1, \phi_1)) \right)$. Therefore, we have $\left\| (\varphi_1 - \varphi_2)^T \left( \nabla_\theta R_p(\varphi_2, P(\theta_1, \phi_1)) - \nabla_\theta R_p(\varphi_1, P(\theta_1, \phi_1)) \right) \right\| \geq \gamma_\theta \|\varphi_1 - \varphi_2\|^2$. Combined with Eq. (18), we have

$$\begin{aligned}
\left\| (\varphi_1 - \varphi_2)^T \left( \nabla_\theta R_p(\varphi_2, P(\theta_1, \phi_1)) - \nabla_\theta R_p(\varphi_2, P(\theta_2, \phi_2)) \right) \right\| &\geq \gamma_\theta \|\varphi_1 - \varphi_2\|^2 \\
\Rightarrow\ L_\theta \varepsilon_\theta \|(\theta_1, \phi_1) - (\theta_2, \phi_2)\| &\geq \gamma_\theta \|\varphi_1 - \varphi_2\| \ \text{by Eq. (17)} \\
\Rightarrow\ \left\| \underset{\theta}{\mathrm{argmin}}\, R_p(\theta, P(\theta_1, \phi_1)) - \underset{\theta}{\mathrm{argmin}}\, R_p(\theta, P(\theta_2, \phi_2)) \right\| &\leq \frac{L_\theta \varepsilon_\theta}{\gamma_\theta} \|(\theta_1, \phi_1) - (\theta_2, \phi_2)\|,
\end{aligned} \tag{20}$$

by the definitions of $\varphi_1, \varphi_2$. Similarly, for model $q$ it's can be proven that for $\forall\, \theta_1, \theta_2, \phi_1, \phi_2$, $\left\| \underset{\phi}{\mathrm{argmin}}\, R_q(\phi, Q(\theta_1, \phi_1)) - \underset{\phi}{\mathrm{argmin}}\, R_q(\phi, Q(\theta_2, \phi_2)) \right\| \leq \frac{L_\phi \varepsilon_\phi}{\gamma_\phi} \|(\theta_1, \phi_1) - (\theta_2, \phi_2)\|$.

Next, consider the iterating process (Eq. (4)) based on distribution parameter $\theta, \phi$. Let $G_p(\theta, \phi) = \underset{\theta'}{\mathrm{argmin}}\, R_p(\theta', P(\theta, \phi))$ and $G_q(\theta, \phi) = \underset{\phi'}{\mathrm{argmin}}\, R_q(\phi', Q(\theta, \phi))$. The iteration process can be regarded as three cases:

$$(\theta_{t+1}, \phi_{t+1}) = \hat{G}_1(\theta_t, \phi_t) \triangleq \left( G_p(\theta_t, \phi_t), G_q(G_p(\theta_t, \phi_t), \phi_t) \right), \tag{21}$$

$$(\theta_{t+1}, \phi_{t+1}) = \hat{G}_2(\theta_t, \phi_t) \triangleq \left( G_p(\theta_t, G_q(\theta_t, \phi_t)), G_q(\theta_t, \phi_t) \right), \tag{22}$$

$$(\theta_{t+1}, \phi_{t+1}) = \hat{G}_3(\theta_t, \phi_t) \triangleq \left( G_p(\theta_t, \phi_t), G_q(\theta_t, \phi_t) \right), \tag{23}$$

There are a total of 9 cases if we consider two adjacent iteration rounds. If both $t+1$th round and $t$th round follow the same iteration pattern. For case 1 (Eq. (21)), notice that

$$\left\|\hat{G}_1(\theta_{t+1},\phi_{t+1})-\hat{G}_1(\theta_t,\phi_t)\right\| \leq \underbrace{\left\|G_p(\theta_{t+1},\phi_{t+1})-G_p(\theta_t,\phi_t)\right\|}_{\text{(I)}} + \underbrace{\left\|G_q(G_p(\theta_{t+1},\phi_{t+1}),\phi_{t+1})-G_q(G_p(\theta_t,\phi_t),\phi_t)\right\|}_{\text{(II)}}.$$

For term (I), we can get (I) $\leq \frac{L_\theta \varepsilon_\theta}{\gamma_\theta}\left\|(\theta_{t+1},\phi_{t+1})-(\theta_t,\phi_t)\right\|$ by Eq. (20). For term (II),

$$\begin{aligned}
\text{(II)} &\leq \left\|G_q\big(G_p(\theta_{t+1},\phi_{t+1}),\phi_{t+1}\big)-G_q\big(G_p(\theta_t,\phi_t),\phi_{t+1}\big)\right\| + \left\|G_q\big(G_p(\theta_t,\phi_t),\phi_{t+1}\big)-G_q\big(G_p(\theta_t,\phi_t),\phi_t\big)\right\| \\
&\leq \frac{L_\phi \varepsilon_\phi}{\gamma_\phi}\left\|G_p(\theta_{t+1},\phi_{t+1})-G_p(\theta_t,\phi_t)\right\| + \frac{L_\phi \varepsilon_\phi}{\gamma_\phi}\left\|\phi_{t+1}-\phi_t\right\| \\
&\leq \frac{L_\phi \varepsilon_\phi}{\gamma_\phi}\frac{L_\theta \varepsilon_\theta}{\gamma_\theta}\left\|(\theta_{t+1},\phi_{t+1})-(\theta_t,\phi_t)\right\| + \frac{L_\phi \varepsilon_\phi}{\gamma_\phi}\left\|\phi_{t+1}-\phi_t\right\| \leq \Big(\frac{L_\theta \varepsilon_\theta L_\phi \varepsilon_\phi}{\gamma_\theta \gamma_\phi} + \frac{L_\phi \varepsilon_\phi}{\gamma_\phi}\Big)\left\|(\theta_{t+1},\phi_{t+1})-(\theta_t,\phi_t)\right\|.
\end{aligned}$$

The derivation of (II)'s upper bound is also based on Eq. (20). By combining the upper bounds of term (I) and (II), we can show that $\left\|(\theta_{t+1},\phi_{t+1})-(\theta_t,\phi_t)\right\| = \left\|\hat{G}_1(\theta_t,\phi_t)-\hat{G}_1(\theta_{t-1},\phi_{t-1})\right\| \leq \big(\frac{L_\theta \varepsilon_\theta}{\gamma_\theta} + \frac{L_\theta \varepsilon_\theta L_\phi \varepsilon_\phi}{\gamma_\theta \gamma_\phi} + \frac{L_\phi \varepsilon_\phi}{\gamma_\phi}\big)\left\|(\theta_t,\phi_t)-(\theta_{t-1},\phi_{t-1})\right\|$. For case 2 (Eq. (22)), similarly we can prove that $\left\|(\theta_{t+1},\phi_{t+1})-(\theta_t,\phi_t)\right\| = \left\|\hat{G}_2(\theta_t,\phi_t)-\hat{G}_2(\theta_{t-1},\phi_{t-1})\right\| \leq \big(\frac{L_\theta \varepsilon_\theta}{\gamma_\theta} + \frac{L_\theta \varepsilon_\theta L_\phi \varepsilon_\phi}{\gamma_\theta \gamma_\phi} + \frac{L_\phi \varepsilon_\phi}{\gamma_\phi}\big)\left\|(\theta_t,\phi_t)-(\theta_{t-1},\phi_{t-1})\right\|$. For case 3 (Eq. (23)), we can show that

$$\begin{aligned}
\left\|(\theta_{t+1},\phi_{t+1})-(\theta_t,\phi_t)\right\| &= \left\|\hat{G}_3(\theta_t,\phi_t)-\hat{G}_3(\theta_{t-1},\phi_{t-1})\right\| \\
&\leq \left\|G_p(\theta_t,\phi_t)-G_p(\theta_{t-1},\phi_{t-1})\right\| + \left\|G_q(\theta_t,\phi_t)-G_q(\theta_{t-1},\phi_{t-1})\right\| \\
&\leq \frac{L_\theta \varepsilon_\theta}{\gamma_\theta}\left\|(\theta_t,\phi_t)-(\theta_{t-1},\phi_{t-1})\right\| + \frac{L_\phi \varepsilon_\phi}{\gamma_\phi}\left\|(\theta_t,\phi_t)-(\theta_{t-1},\phi_{t-1})\right\| \\
&< \big(\frac{L_\theta \varepsilon_\theta}{\gamma_\theta} + \frac{L_\theta \varepsilon_\theta L_\phi \varepsilon_\phi}{\gamma_\theta \gamma_\phi} + \frac{L_\phi \varepsilon_\phi}{\gamma_\phi}\big)\left\|(\theta_t,\phi_t)-(\theta_{t-1},\phi_{t-1})\right\|.
\end{aligned}$$

Therefore, for all the 3 cases if 2 adjacent rounds follow the same iteration pattern, $\left\|(\theta_{t+1},\phi_{t+1})-(\theta_t,\phi_t)\right\| \leq \big(\frac{L_\theta \varepsilon_\theta}{\gamma_\theta} + \frac{L_\theta \varepsilon_\theta L_\phi \varepsilon_\phi}{\gamma_\theta \gamma_\phi} + \frac{L_\phi \varepsilon_\phi}{\gamma_\phi}\big)\left\|(\theta_t,\phi_t)-(\theta_{t-1},\phi_{t-1})\right\|$.

Next, if adjacent rounds' iteration case are different, for example $(\theta_{t+1},\phi_{t+1}) = \hat{G}_1(\theta_t,\phi_t)$ and $(\theta_t,\phi_t) = \hat{G}_2(\theta_t,\phi_t)$, we can get

$$\begin{aligned}
\left\|(\theta_{t+1},\phi_{t+1})-(\theta_t,\phi_t)\right\| &= \left\|(G_p(\theta_t,\phi_t),G_q(G_p(\theta_t,\phi_t),\phi_t))-(G_p(\theta_{t-1},G_q(\theta_{t-1},\phi_{t-1})),G_q(\theta_{t-1},\phi_{t-1}))\right\| \\
&\leq \left\|G_p(\theta_t,\phi_t)-G_p(\theta_{t-1},G_q(\theta_{t-1},\phi_{t-1}))\right\| + \left\|G_q(G_p(\theta_t,\phi_t),\phi_t))-G_q(\theta_{t-1},\phi_{t-1})\right\| \\
&\leq \frac{L_\theta \varepsilon_\theta}{\gamma_\theta}\big(\|\theta_t-\theta_{t-1}\| + \underbrace{\|\phi_t-G_q(\theta_{t-1},\phi_{t-1})\|}_{=0}\big) + \frac{L_\phi \varepsilon_\phi}{\gamma_\phi}\big(\underbrace{\|G_p(\theta_t,\phi_t)-\theta_{t-1}\|}_{=\theta_{t+1}-\theta_{t-1}} + \|\phi_t-\phi_{t-1}\|\big) \\
&\leq \big(\frac{L_\theta \varepsilon_\theta}{\gamma_\theta} + \frac{2L_\phi \varepsilon_\phi}{\gamma_\phi}\big)\left\|(\theta_t,\phi_t)-(\theta_{t-1},\phi_{t-1})\right\| + \frac{L_\phi \varepsilon_\phi}{\gamma_\phi}\left\|(\theta_{t+1},\phi_{t+1})-(\theta_t,\phi_t)\right\|.
\end{aligned}$$

(24)

If $\frac{L_\phi \varepsilon_\phi}{\gamma_\phi} < 1$, then

$$\left\|(\theta_{t+1},\phi_{t+1})-(\theta_t,\phi_t)\right\| \leq \frac{\gamma_\phi L_\theta \varepsilon_\theta + 2\gamma_\theta L_\phi \varepsilon_\phi}{\gamma_\theta(\gamma_\phi - L_\phi \varepsilon_\phi)}\left\|(\theta_t,\phi_t)-(\theta_{t-1},\phi_{t-1})\right\|.$$

Similarly, we can prove the compression parameter for all 9 cases shown in Table 1. It's easy to check that if $\frac{L_\phi \varepsilon_\phi}{\gamma_\phi} < 1$ and $\frac{L_\theta \varepsilon_\theta}{\gamma_\theta} < 1$, then $\max\big(\frac{\gamma_\theta L_\phi \varepsilon_\phi + 2\gamma_\phi L_\theta \varepsilon_\theta}{\gamma_\phi(\gamma_\theta - L_\theta \varepsilon_\theta)}, \frac{\gamma_\phi L_\theta \varepsilon_\theta + 2\gamma_\theta L_\phi \varepsilon_\phi}{\gamma_\theta(\gamma_\phi - L_\phi \varepsilon_\phi)}\big) \geq \frac{L_\theta \varepsilon_\theta}{\gamma_\theta} + \frac{L_\theta \varepsilon_\theta L_\phi \varepsilon_\phi}{\gamma_\theta \gamma_\phi} + \frac{L_\phi \varepsilon_\phi}{\gamma_\phi} > 0$.

If $\kappa = \max\big(\frac{\gamma_\theta L_\phi \varepsilon_\phi + 2\gamma_\phi L_\theta \varepsilon_\theta}{\gamma_\phi(\gamma_\theta - L_\theta \varepsilon_\theta)}, \frac{\gamma_\phi L_\theta \varepsilon_\theta + 2\gamma_\theta L_\phi \varepsilon_\phi}{\gamma_\theta(\gamma_\phi - L_\phi \varepsilon_\phi)}, \frac{L_\theta \varepsilon_\theta}{\gamma_\theta} + \frac{L_\phi \varepsilon_\phi}{\gamma_\phi}\big) < 1$, then by Banach's fixed point theorem, the stable point $(\theta^*,\phi^*)$ under our iteration setting exists and $(\theta^*,\phi^*) = \hat{G}_1(\theta^*,\phi^*)$. This also proves that the stable point satisfies Eq. (5). To prove the convergence rate, notice that

$$\begin{aligned}
\left\|(\theta_t,\phi_t)-(\theta^*,\phi^*)\right\| &= \left\|\hat{G}_1(\theta_{t-1},\phi_{t-1})-\hat{G}_1(\theta^*,\phi^*)\right\| \\
&\leq \kappa\left\|(\theta_{t-1},\phi_{t-1})-(\theta^*,\phi^*)\right\| \leq \ ... \ \leq \kappa^t\left\|(\theta_0,\phi_0)-(\theta^*,\phi^*)\right\|.
\end{aligned}$$

*Table 1.* Compression parameter for 9 possible iteration cases.

| $(\theta_{t+1},\phi_{t+1})$ | $(\theta_t,\phi_t)$ | Compression Parameter | $(\theta_{t+1},\phi_{t+1})$ | $(\theta_t,\phi_t)$ | Compression Parameter |
|---|---|---|---|---|---|
| $\hat{G}_i(\theta_t,\phi_t)$ | $\hat{G}_i(\theta_{t-1},\phi_{t-1})$ | $\frac{L_\theta\varepsilon_\theta}{\gamma_\theta}+\frac{L_\theta\varepsilon_\theta L_\phi\varepsilon_\phi}{\gamma_\theta\gamma_\phi}+\frac{L_\phi\varepsilon_\phi}{\gamma_\phi},i\neq3$ | $\hat{G}_1(\theta_t,\phi_t)$ | $\hat{G}_3(\theta_{t-1},\phi_{t-1})$ | $\frac{\gamma_\phi L_\theta\varepsilon_\theta+2\gamma_\theta L_\phi\varepsilon_\phi}{\gamma_\theta(\gamma_\phi-L_\phi\varepsilon_\phi)}$ |
| $\hat{G}_3(\theta_t,\phi_t)$ | $\hat{G}_3(\theta_{t-1},\phi_{t-1})$ | $\frac{L_\theta\varepsilon_\theta}{\gamma_\theta}+\frac{L_\phi\varepsilon_\phi}{\gamma_\phi}$ | $\hat{G}_3(\theta_t,\phi_t)$ | $\hat{G}_1(\theta_{t-1},\phi_{t-1})$ | $\frac{L_\theta\varepsilon_\theta}{\gamma_\theta}+\frac{L_\phi\varepsilon_\phi}{\gamma_\phi}$ |
| $\hat{G}_1(\theta_t,\phi_t)$ | $\hat{G}_2(\theta_{t-1},\phi_{t-1})$ | $\frac{\gamma_\phi L_\theta\varepsilon_\theta+2\gamma_\theta L_\phi\varepsilon_\phi}{\gamma_\theta(\gamma_\phi-L_\phi\varepsilon_\phi)}$ | $\hat{G}_2(\theta_t,\phi_t)$ | $\hat{G}_3(\theta_{t-1},\phi_{t-1})$ | $\frac{\gamma_\theta L_\phi\varepsilon_\phi+2\gamma_\phi L_\theta\varepsilon_\theta}{\gamma_\phi(\gamma_\theta-L_\theta\varepsilon_\theta)}$ |
| $\hat{G}_2(\theta_t,\phi_t)$ | $\hat{G}_1(\theta_{t-1},\phi_{t-1})$ | $\frac{\gamma_\theta L_\phi\varepsilon_\phi+2\gamma_\phi L_\theta\varepsilon_\theta}{\gamma_\phi(\gamma_\theta-L_\theta\varepsilon_\theta)}$ | $\hat{G}_3(\theta_t,\phi_t)$ | $\hat{G}_2(\theta_{t-1},\phi_{t-1})$ | $\frac{L_\theta\varepsilon_\theta}{\gamma_\theta}+\frac{L_\phi\varepsilon_\phi}{\gamma_\phi}$ |

$\square$

## F.2. Proof of Proposition 4.4

**Proposition 4.4.** *Under Assumptions 3.2-3.5 and condition in Theorem 3.6, $\nabla_\theta\mathbf{F}_p$, $\nabla_\phi\mathbf{F}_q$, $S_p$ and $S_q$ are all invertible, and $\nabla_\theta\mathbf{F}_p \succeq (\gamma_\theta-L_\theta\varepsilon_\theta)I$, $\nabla_\phi\mathbf{F}_q \succeq (\gamma_\phi-L_\phi\varepsilon_\phi)I$ and $S_p^{-1}=\nabla_\theta\mathbf{F}_p-\nabla_\phi\mathbf{F}_p\nabla_\phi\mathbf{F}_q^{-1}\nabla_\theta\mathbf{F}_q \succeq (\gamma_\theta-L_\theta\varepsilon_\theta-\frac{L_\theta\varepsilon_\theta L_\phi\varepsilon_\phi}{\gamma_\phi-L_\phi\varepsilon_\phi})I \succ 0$, $S_q^{-1}=\nabla_\phi\mathbf{F}_q-\nabla_\theta\mathbf{F}_q\nabla_\theta\mathbf{F}_p^{-1}\nabla_\phi\mathbf{F}_p \succeq (\gamma_\phi-L_\phi\varepsilon_\phi-\frac{L_\theta\varepsilon_\theta L_\phi\varepsilon_\phi}{\gamma_\theta-L_\theta\varepsilon_\theta})I \succ 0$ and $I$ is the identity matrix.*

*Proof.* From Thm. (3.6), since $\kappa=\max(\frac{\gamma_\phi L_\theta\varepsilon_\theta+2\gamma_\theta L_\phi\varepsilon_\phi}{\gamma_\phi(\gamma_\theta-L_\theta\varepsilon_\theta)}, \frac{\gamma_\phi L_\theta\varepsilon_\theta+2\gamma_\theta L_\phi\varepsilon_\phi}{\gamma_\theta(\gamma_\phi-L_\phi\varepsilon_\phi)}, \frac{L_\theta\varepsilon_\theta}{\gamma_\theta}+\frac{L_\phi\varepsilon_\phi}{\gamma_\phi})<1$, it's easy to check that $\gamma_\theta-L_\theta\varepsilon_\theta>0$, $\gamma_\phi-L_\phi\varepsilon_\phi>0$ and $\gamma_\theta-L_\theta\varepsilon_\theta-\frac{L_\theta\varepsilon_\theta L_\phi\varepsilon_\phi}{\gamma_\phi-L_\phi\varepsilon_\phi}>0$, $\gamma_\phi-L_\phi\varepsilon_\phi-\frac{L_\theta\varepsilon_\theta L_\phi\varepsilon_\phi}{\gamma_\theta-L_\theta\varepsilon_\theta}>0$.

First, we prove that $\nabla_\theta\mathbf{F}_p \succeq (\gamma_\theta-L_\theta\varepsilon_\theta)I$, $\nabla_\phi\mathbf{F}_q \succeq (\gamma_\phi-L_\phi\varepsilon_\phi)I$. For any vector $v\neq0$, by $\nabla_\theta\mathbf{F}_p$'s definition we decompose the matrix into different dimensions and can get

$$
\begin{aligned}
v\nabla_\theta\mathbf{F}_p v^T &= v\frac{\partial}{\partial\theta}\mathbb{E}_{z\sim P(\theta,\phi)}\nabla_\theta\ell_\theta(z;\theta)\big|_{(\theta,\phi)=(\theta^*,\phi^*)}v^T \\
&= \mathbb{E}_{z\sim P(\theta^*,\phi^*)}\big[v\nabla_\theta^2\ell_\theta(z;\theta^*)v^T\big] + \frac{\partial}{\partial t}\mathbb{E}_{z\sim P(\theta^*+tv,\phi^*)}\big[\nabla_\theta\ell_\theta(z;\theta^*)v^T\big]\big|_{t=0}.
\end{aligned}
\tag{25}
$$

For the first term in Eq. (25), since $\ell_\theta$ is $\gamma_\theta$-strong convex (Assump. (3.2)), it's easy to check that $\mathbb{E}_{z\sim P(\theta^*,\phi^*)}\nabla_\theta^2\ell_\theta(z;\theta^*) \succeq \gamma_\theta I$ where $I$ is the identity matrix. Therefore, $\mathbb{E}_{z\sim P(\theta^*,\phi^*)}\big[v\nabla_\theta^2\ell_\theta(z;\theta^*)v^T\big] \geq \gamma_\theta\|v\|^2$. For the second term in Eq. (25),

$$
\begin{aligned}
\left|\frac{\partial}{\partial t}\mathbb{E}_{z\sim P(\theta^*+tv,\phi^*)}\big[\nabla_\theta\ell_\theta(z;\theta^*)v^T\big]\big|_{t=0}\right| &= \left|\lim_{\delta\to0}\frac{\mathbb{E}_{z\sim P(\theta^*+\delta v,\phi^*)}\big[\nabla_\theta\ell_\theta(z;\theta^*)v^T\big]-\mathbb{E}_{z\sim P(\theta^*,\phi^*)}\big[\nabla_\theta\ell_\theta(z;\theta^*)v^T\big]}{\delta}\right| \\
&\leq \sup_{|\delta|\leq1}\frac{|\mathbb{E}_{z\sim P(\theta^*+\delta v,\phi^*)}\big[\nabla_\theta\ell_\theta(z;\theta^*)v^T\big]-\mathbb{E}_{z\sim P(\theta^*,\phi^*)}\big[\nabla_\theta\ell_\theta(z;\theta^*)v^T\big]|}{|\delta|},
\end{aligned}
$$

and notice that by Assump. (3.3) for any $z_1,z_2$,

$$
|\nabla_\theta\ell_\theta(z_1;\theta^*)v^T-\nabla_\theta\ell_\theta(z_2;\theta^*)v^T| \leq \|\nabla_\theta\ell_\theta(z_1;\theta^*)-\nabla_\theta\ell_\theta(z_2;\theta^*)\|\|v\| \leq L_\theta\|v\|\|z_1-z_2\|,
$$

which means $\nabla_\theta\ell_\theta(z;\theta^*)v^T:\mathbb{R}^n\to\mathbb{R}$ is a $L_\theta\|v\|$-Lipschitz function. Therefore, combined with Kantorovich-Rubinstein (Lemma F.1) and $\varepsilon_\theta$-sensitivity, we can get

$$
\left|\frac{\partial}{\partial t}\mathbb{E}_{z\sim P(\theta^*+tv,\phi^*)}\big[\nabla_\theta\ell_\theta(z;\theta^*)v^T\big]\big|_{t=0}\right| \leq \frac{L_\theta\|v\|\varepsilon_\theta\|\delta v\|}{|\delta|} = L_\theta\varepsilon_\theta\|v\|^2.
\tag{26}
$$

Back to Eq. (25), we have

$$
v\nabla_\theta\mathbf{F}_p v^T \geq (\gamma_\theta-L_\theta\varepsilon_\theta)\|v\|^2 > 0.
$$

Similarly, we can prove that $\nabla_\phi\mathbf{F}_q \succeq (\gamma_\phi-L_\phi\varepsilon_\phi)I$.

Next, we prove that $\hat{S}_p=\nabla_\theta\mathbf{F}_p-\nabla_\phi\mathbf{F}_p\nabla_\phi\mathbf{F}_q^{-1}\nabla_\theta\mathbf{F}_q \succeq (\gamma_\theta-L_\theta\varepsilon_\theta-\frac{L_\theta\varepsilon_\theta L_\phi\varepsilon_\phi}{\gamma_\phi-L_\phi\varepsilon_\phi})I$. The corresponding conclusion for $S_q$ can be proven symmetrically. Consider the upper bound of $\|\nabla_\theta\mathbf{F}_q\|$ and $\|\nabla_\phi\mathbf{F}_p\|$. Adding small perturbations $th$ to $\phi$ in

$F_p, t \in \mathbb{R}$ and similar to the proof of $\nabla_\theta \mathbf{F}_p$, calculate

$$
\begin{aligned}
\|\nabla_\theta \mathbf{F}_q\| &= \sup_{\|h\| \leq 1} \frac{\|h \nabla_\theta \mathbf{F}_q\|}{\|h\|} = \lim_{t \to 0} \sup_{\|h\| \leq 1} \frac{\left\| F_p(\theta, \phi + th) - F_p(\theta, \phi) \right\|}{\|th\|} \\
&= \lim_{t \to 0} \sup_{\|h\| \leq 1} \frac{\left\| \mathbb{E}_{z \sim P(\theta, \phi + th)} \nabla_\theta \ell_\theta(z; \theta) - \mathbb{E}_{z \sim P(\theta, \phi)} \nabla_\theta \ell_\theta(z; \theta) \right\|}{\|th\|} \\
&\leq \sup_{\|h\| \leq 1} \frac{L_\theta \left\| \mathbb{E}_{z \sim P(\theta, \phi + h)} \frac{\nabla_\theta \ell_\theta(z; \theta)}{L_\theta} - \mathbb{E}_{z \sim P(\theta, \phi)} \frac{\nabla_\theta \ell_\theta(z; \theta)}{L_\theta} \right\|}{\|h\|} \leq \frac{L_\theta \varepsilon_\theta \left\| (\theta, \phi + h) - (\theta, \phi) \right\|}{\|h\|} = L_\theta \varepsilon_\theta. \quad (27)
\end{aligned}
$$

The inequality in the derivation of Eq. 27 is due to the distribution $P$'s $\varepsilon_\theta$-sensitivity and Kantorovich-Rubinstein (Lemma F.1). Similarly, we have $\|\nabla_\theta \mathbf{F}_q\| \leq L_\phi \varepsilon_\phi$.

Recall that $\hat{S}_p = \nabla_\theta \mathbf{F}_p - \nabla_\phi \mathbf{F}_p \nabla_\phi \mathbf{F}_q^{-1} \nabla_\theta \mathbf{F}_q$. Therefore, for any vector $v \neq 0$,

$$
\begin{aligned}
v^T \hat{S}_p v &= v^T \nabla_\theta \mathbf{F}_p v - v^T \nabla_\phi \mathbf{F}_p \nabla_\phi \mathbf{F}_q^{-1} \nabla_\theta \mathbf{F}_q v \\
&\geq (\gamma_\theta - L_\theta \varepsilon_\theta) \|v\|^2 - \|v^T \nabla_\phi \mathbf{F}_p\| \|\nabla_\phi \mathbf{F}_q^{-1}\| \|\nabla_\theta \mathbf{F}_q v\| \geq (\gamma_\theta - L_\theta \varepsilon_\theta - \frac{L_\theta \varepsilon_\theta L_\phi \varepsilon_\phi}{\gamma_\phi - L_\phi \varepsilon_\phi}) \|v\|^2, \quad (28)
\end{aligned}
$$

since $L_\theta \varepsilon_\theta \geq \|\nabla_\phi \mathbf{F}_p\| \geq \frac{\|\nabla_\phi \mathbf{F}_p v\|}{\|v\|}$ by the matrix norm's definition and $\nabla_\phi \mathbf{F}_q \succeq (\gamma_\phi - L_\phi \varepsilon_\phi) I, \nabla_\theta \mathbf{F}_p \succeq (\gamma_\theta - L_\phi \varepsilon_\phi) I$. The proof of $\|\nabla_\phi \mathbf{F}_q^{-1}\| \leq (\gamma_\phi - L_\phi \varepsilon_\phi)^{-1}$ is similar to the proof in Corollary 4.7. Therefore, by Eq. (28), $\hat{S}_p \succeq (\gamma_\theta - L_\theta \varepsilon_\theta - \frac{L_\theta \varepsilon_\theta L_\phi \varepsilon_\phi}{\gamma_\phi - L_\phi \varepsilon_\phi}) I$ and it is invertible. Notice that $\hat{S}_p$ is not necessarily a symmetric matrix, and its positive definiteness differs from the traditional symmetric positive definiteness. Similarly, we can prove that $S_q$ is invertible. $\qquad\square$

### F.3. Proof of Proposition 4.1

[Characterization of $\frac{\partial \theta^*}{\partial \lambda_{\mathcal{H}}^\theta}$ and $\frac{\partial \phi^*}{\partial \lambda_{\mathcal{H}}^\theta}$] $\frac{\partial \theta^*}{\partial \lambda_{\mathcal{H}}^\theta} = -S_p \left( \frac{\partial F_p}{\partial \lambda_{\mathcal{H}}^\theta} + C_p \frac{\partial F_q}{\partial \lambda_{\mathcal{H}}^\theta} \right)$, $\frac{\partial \phi^*}{\partial \lambda_{\mathcal{H}}^\theta} = -S_q \left( C_q \frac{\partial F_p}{\partial \lambda_{\mathcal{H}}^\theta} + \frac{\partial F_q}{\partial \lambda_{\mathcal{H}}^\theta} \right)$. where $S_p, S_q$ are the sensitivity matrices and $C_p, C_q$ are the cross-model influence matrices.

*Proof.* In the main paper, we already have that

$$
\forall \lambda_{\mathcal{H}}^\theta, \quad F_p(\theta^*, \phi^*, \lambda_{\mathcal{H}}^\theta) = F_q(\theta^*, \phi^*, \lambda_{\mathcal{H}}^\theta) = 0.
$$

Differentiating the above equation with respect to $\lambda_{\mathcal{H}}^\theta$, we obtain:

$$
\begin{aligned}
\frac{\partial F_p(\theta^*, \phi^*, \lambda_{\mathcal{H}}^\theta)}{\partial \theta} \frac{\partial \theta^*(\lambda_{\mathcal{H}}^\theta)}{\partial \lambda_{\mathcal{H}}^\theta} + \frac{\partial F_p(\theta^*, \phi^*, \lambda_{\mathcal{H}}^\theta)}{\partial \phi} \frac{\partial \phi^*(\lambda_{\mathcal{H}}^\theta)}{\partial \lambda_{\mathcal{H}}^\theta} + \frac{\partial F_p(\theta^*, \phi^*, \lambda_{\mathcal{H}}^\theta)}{\partial \lambda_{\mathcal{H}}^\theta} = 0, \\
\frac{\partial F_q(\theta^*, \phi^*, \lambda_{\mathcal{H}}^\theta)}{\partial \phi} \frac{\partial \phi^*(\lambda_{\mathcal{H}}^\theta)}{\partial \lambda_{\mathcal{H}}^\theta} + \frac{\partial F_q(\theta^*, \phi^*, \lambda_{\mathcal{H}}^\theta)}{\partial \theta} \frac{\partial \theta^*(\lambda_{\mathcal{H}}^\theta)}{\partial \lambda_{\mathcal{H}}^\theta} + \frac{\partial F_q(\theta^*, \phi^*, \lambda_{\mathcal{H}}^\theta)}{\partial \lambda_{\mathcal{H}}^\theta} = 0.
\end{aligned} \quad (29)
$$

Combined with $\nabla_\theta \boldsymbol{F}_p, \nabla_\theta \boldsymbol{F}_q$ in Definition 4.2 and Proposition 4.4, solving Eq. (29) yields that

$$
\begin{aligned}
\frac{\partial \theta^*}{\partial \lambda_{\mathcal{H}}^\theta} &= -(\nabla_\theta \boldsymbol{F}_p - \nabla_\phi \boldsymbol{F}_p (\nabla_\phi \boldsymbol{F}_q)^{-1} \nabla_\theta \boldsymbol{F}_q)^{-1} \left( \frac{\partial F_p(\theta^*, \phi^*, \lambda_{\mathcal{H}}^\theta)}{\partial \lambda_{\mathcal{H}}^\theta} - \nabla_\phi \boldsymbol{F}_p (\nabla_\phi \boldsymbol{F}_q)^{-1} \frac{\partial F_q(\theta^*, \phi^*, \lambda_{\mathcal{H}}^\theta)}{\partial \lambda_{\mathcal{H}}^\theta} \right), \\
\frac{\partial \phi^*}{\partial \lambda_{\mathcal{H}}^\theta} &= -(\nabla_\phi \boldsymbol{F}_q - \nabla_\theta \boldsymbol{F}_q (\nabla_\theta \boldsymbol{F}_p)^{-1} \nabla_\phi \boldsymbol{F}_p)^{-1} \left( \frac{\partial F_q(\theta^*, \phi^*, \lambda_{\mathcal{H}}^\theta)}{\partial \lambda_{\mathcal{H}}^\theta} - \nabla_\theta \boldsymbol{F}_q (\nabla_\theta \boldsymbol{F}_p)^{-1} \frac{\partial F_p(\theta^*, \phi^*, \lambda_{\mathcal{H}}^\theta)}{\partial \lambda_{\mathcal{H}}^\theta} \right).
\end{aligned}
$$

$\qquad\square$

## F.4. Proof of Theorem 4.5

**Theorem 4.5.** *Under Assumptions 3.2-3.5 and condition in Theorem 3.6, the self-consuming multi-model system converges. Let $P_{\mathcal{H}}$ be the distribution of human-curated data $\mathcal{H}_t^\theta$ after convergence.*

$$\frac{\partial J_p(\theta^*)}{\partial \lambda_{\mathcal{H}}^\theta} = \frac{1}{1 - \lambda_{\mathcal{H}}^\theta} \langle \nabla_\theta J_p(\theta^*), \, S_p \, \mathbb{E}_{P_{\mathcal{H}}}[-\nabla_\theta \ell_\theta(\theta^*)] \rangle,$$

$$\frac{\partial J_q(\phi^*)}{\partial \lambda_{\mathcal{H}}^\theta} = \frac{1}{1 - \lambda_{\mathcal{H}}^\theta} \langle \nabla_\phi J_q(\phi^*), \, S_q C_q \, \mathbb{E}_{P_{\mathcal{H}}}[-\nabla_\theta \ell_\theta(\theta^*)] \rangle.$$

*where $S_p, S_q$ are the sensitivity matrices, $C_q$ is the cross-model influence matrix, and $\langle \cdot, \cdot \rangle$ denotes the inner product between two vectors.*

*Proof.* For model $\theta$, let $f_\theta(\theta, \phi)$ be $P(\theta, \phi)$'s density functions, and $f_{\theta,c}(\theta, \phi), f_{\theta,s}(\theta, \phi)$ be the density functions of $\mathcal{H}^\theta(\theta, \phi)$'s distribution, $\mathcal{S}^\theta(\theta, \phi)$'s distribution respectively. For model $\phi$, let $f_\phi(\theta, \phi)$ be $Q(\theta, \phi)$'s density functions, and $f_{\phi,c}(\theta, \phi), f_{\phi,s}(\theta, \phi)$ be the density functions of $\mathcal{H}^\phi(\theta, \phi)$'s distribution, $\mathcal{S}^\phi(\theta, \phi)$'s distribution respectively. For the fixed real data distribution, let $f_{\theta,r}, f_{\phi,r}$ be the corresponding density functions. Therefore, by the definition of the mixture distributions $P$ and $Q$, we have

$$f_\theta(\theta, \phi) = \lambda_{\mathcal{R}}^\theta f_{\theta,r} + \lambda_{\mathcal{S}}^\theta f_{\theta,s}(\theta, \phi) + \lambda_{\mathcal{H}}^\theta f_{\theta,c}(\theta, \phi), \; f_\phi(\theta, \phi) = \lambda_{\mathcal{R}}^\phi f_{\phi,r} + \lambda_{\mathcal{S}}^\phi f_{\phi,s}(\theta, \phi) + \lambda_{\mathcal{H}}^\phi f_{\phi,c}(\theta, \phi). \tag{30}$$

By the definition of $F_p$, its partial gradient of $\lambda_{\mathcal{H}}^\theta$ in Proposition 4.1 is

$$\frac{\partial F_p}{\partial \lambda_{\mathcal{H}}^\theta} = \frac{\partial}{\partial \lambda_{\mathcal{H}}^\theta} \mathbb{E}_{z=(x,y)\sim P(\theta^*, \phi^*)} \nabla_\theta \ell_\theta(z; \theta^*) = \frac{\partial}{\partial \lambda_{\mathcal{H}}^\theta} \int \nabla_\theta \ell_\theta(z; \theta^*) f_\theta(z; \theta^*, \phi^*) dz$$

$$= \frac{\partial}{\partial \lambda_{\mathcal{H}}^\theta} \int \nabla_\theta \ell_\theta(z; \theta^*) \left( \lambda_{\mathcal{S}}^\theta f_{\theta,s}(z; \theta^*, \phi^*) + \lambda_{\mathcal{H}}^\theta f_{\theta,c}(z; \theta^*, \phi^*) + \lambda_{\mathcal{R}}^\theta f_{\theta,r}(z) \right) dz. \tag{31}$$

We consider $f_\theta(z; \theta, \phi) = f_\theta(\lambda_{\mathcal{S}}^\theta, \lambda_{\mathcal{H}}^\theta, \lambda_{\mathcal{R}}^\theta; z, \theta, \phi)$ as a function of $(\lambda_{\mathcal{S}}^\theta, \lambda_{\mathcal{H}}^\theta, \lambda_{\mathcal{R}}^\theta)$ satisfying $\lambda_{\mathcal{R}}^\theta + \lambda_{\mathcal{S}}^\theta + \lambda_{\mathcal{H}}^\theta = 1$, and its total derivative is

$$df_\theta(\lambda_{\mathcal{S}}^\theta, \lambda_{\mathcal{H}}^\theta, \lambda_{\mathcal{R}}^\theta; z, \theta, \phi) = f_{\theta,s}(z; \theta, \phi) d\lambda_{\mathcal{S}}^\theta + f_{\theta,c}(z; \theta, \phi) d\lambda_{\mathcal{H}}^\theta + f_{\theta,r}(z) d\lambda_{\mathcal{R}}^\theta, \; d\lambda_{\mathcal{S}}^\theta + d\lambda_{\mathcal{H}}^\theta + d\lambda_{\mathcal{R}}^\theta = 0.$$

Since changing model $\theta$'s human curation strength is done by increasing or decreasing the size of $\mathcal{H}^\theta$, it's easy to check that $d\lambda_{\mathcal{S}}^\theta = -\frac{\lambda_{\mathcal{S}}^\theta}{\lambda_{\mathcal{S}}^\theta + \lambda_{\mathcal{R}}^\theta} d\lambda_{\mathcal{H}}^\theta$ and $d\lambda_{\mathcal{R}}^\theta = -\frac{\lambda_{\mathcal{R}}^\theta}{\lambda_{\mathcal{S}}^\theta + \lambda_{\mathcal{R}}^\theta} d\lambda_{\mathcal{H}}^\theta$.

Therefore, by Eq. (31),

$$\frac{\partial F_p}{\partial \lambda_{\mathcal{H}}^\theta} = \int \nabla_\theta \ell_\theta(z; \theta^*) \frac{df_\theta(\lambda_{\mathcal{S}}^\theta, \lambda_{\mathcal{H}}^\theta, \lambda_{\mathcal{R}}^\theta; z, \theta^*, \phi^*)}{d\lambda_{\mathcal{H}}^\theta} dz$$

$$= \int \nabla_\theta \ell_\theta(z; \theta^*) \left( -\frac{\lambda_{\mathcal{S}}^\theta}{\lambda_{\mathcal{S}}^\theta + \lambda_{\mathcal{R}}^\theta} f_{\theta,s}(z; \theta^*, \phi^*) + f_{\theta,c}(z; \theta^*, \phi^*) - \frac{\lambda_{\mathcal{R}}^\theta}{\lambda_{\mathcal{S}}^\theta + \lambda_{\mathcal{R}}^\theta} f_{\theta,r}(z) \right) dz$$

$$= \int \nabla_\theta \ell_\theta(z; \theta^*) \left( \frac{f_{\theta,c}(z; \theta^*, \phi^*)}{\lambda_{\mathcal{S}}^\theta + \lambda_{\mathcal{R}}^\theta} - \frac{\lambda_{\mathcal{S}}^\theta f_{\theta,s}(z; \theta^*, \phi^*) + \lambda_{\mathcal{H}}^\theta f_{\theta,c}(z; \theta^*, \phi^*) + \lambda_{\mathcal{R}}^\theta f_{\theta,r}(z)}{\lambda_{\mathcal{S}}^\theta + \lambda_{\mathcal{R}}^\theta} \right) dz$$

$$= \frac{1}{1 - \lambda_{\mathcal{H}}^\theta} \left( \mathbb{E}_{z\sim P_{\mathcal{H}}} \nabla_\theta \ell_\theta(z; \theta^*) - \mathbb{E}_{z\sim P(\theta^*, \phi^*)} \nabla_\theta \ell_\theta(z; \theta^*) \right) = \frac{1}{1 - \lambda_{\mathcal{H}}^\theta} \mathbb{E}_{z\sim P_{\mathcal{H}}} \nabla_\theta \ell_\theta(\theta^*) \tag{32}$$

since $\mathbb{E}_{z\sim P(\theta^*, \phi^*)} \nabla_\theta \ell_\theta(\theta^*) = 0$ by the definition of $\theta^*, \phi^*$. Similarly, since we only change the human curation ratio $\lambda_{\mathcal{H}}^\theta$ for model $p$, it's easy to check that $\frac{\partial F_q}{\partial \lambda_{\mathcal{H}}^\theta} = 0$.

Therefore, by the chain rule and Proposition 4.1, we can get

$$\frac{\partial J_p(\theta^*)}{\partial \lambda_{\mathcal{H}}^\theta} = \frac{\partial J_p(\theta^*)}{\partial \theta} \frac{\partial \theta^*}{\partial \lambda_{\mathcal{H}}^\theta} = \langle \nabla_\theta J_p(\theta^*), -S_p \left( \frac{\partial F_p}{\partial \lambda_{\mathcal{H}}^\theta} + C_p \frac{\partial F_q}{\partial \lambda_{\mathcal{H}}^\theta} \right) \rangle$$

$$= \frac{1}{1 - \lambda_{\mathcal{H}}^\theta} \langle \nabla_\theta J_p(\theta^*), S_p \mathbb{E}_{P_{\mathcal{H}}}[-\nabla_\theta \ell_\theta(\theta^*)] \rangle.$$

Similarly, by Proposition 4.1 and the chain rule, we have

$$\frac{\partial J_q(\phi^*)}{\partial \lambda_{\mathcal{H}}^\theta} = \frac{\partial J_q(\phi^*)}{\partial \phi^*}\frac{\partial \phi^*}{\partial \lambda_{\mathcal{H}}^\theta} = \langle \nabla_\phi J_q(\phi^*), -S_q\big(\frac{\partial F_q}{\partial \lambda_{\mathcal{H}}^\theta} + C_q\frac{\partial F_p}{\partial \lambda_{\mathcal{H}}^\theta}\big)\rangle$$

$$= \frac{1}{1-\lambda_{\mathcal{H}}^\theta}\langle \nabla_\phi J_q(\phi^*), S_q C_q \, \mathbb{E}_{P_{\mathcal{H}}}[-\nabla_\theta \ell_\theta(\theta^*)]\rangle.$$

$\square$

*Remark* F.2 (Why $\frac{1}{1-\lambda_{\mathcal{H}}^\theta}$ is well-defined in Theorem 4.5?). In this theorem, $\lambda_{\mathcal{H}}^\theta$ cannot be equal to 1. Since if $\lambda_{\mathcal{H}}^\theta = 1$, the curation data ratio $\lambda_{\mathcal{H}}^\theta$ will still be 1 regardless of whether the number of curated synthetic samples is increased or decreased. According to the definition of total derivative, $\frac{df_\theta(\lambda_{\mathcal{S}}^\theta, \lambda_{\mathcal{H}}^\theta, \lambda_{\mathcal{R}}^\theta; z, \theta, \phi)}{\lambda_{\mathcal{H}}^\theta}$ is undefined at $\lambda_{\mathcal{H}}^\theta = 1$ unless it is specified how $\mathcal{R}^\theta$ and $\mathcal{S}^\theta$ change at that point.

## F.5. Proof of Corollary E.1

**Corollary E.1** *For a single model $\theta$, under Assumptions 3.2-3.5, if $\rho_p \approx 1$, then $\frac{\partial J_p(\theta^*)}{\partial \lambda_{\mathcal{H}^\theta}} > 0$.*

*Proof.* Since there is only one model, by the definitions of $S_p$ and $C_p$, $S_p = (\nabla_\theta \mathbf{F}_p)^{-1}$ and $C_p = 0$ without model interactions. Notice that, $\nabla_\theta \mathbf{F}_p \succeq (\gamma_\theta - L_\theta \varepsilon_\theta)I \succ 0$. Therefore, $\nabla_\theta \mathbf{F}_p$ is invertible and for any $x \neq 0$, let $y = S_p x \neq 0$, we have

$$x^T S_p x = x^T S_p^T \frac{\nabla_\theta \mathbf{F}_p + \nabla_\theta \mathbf{F}_p^T}{2} S_p x = y^T \frac{\nabla_\theta \mathbf{F}_p + \nabla_\theta \mathbf{F}_p^T}{2} y = y^T \nabla_\theta \mathbf{F}_p y > 0,$$

which means $S_p = (\nabla_\theta \mathbf{F}_p)^{-1}$ is also positive definite. Moreover, since $\rho_p \approx 1$, there exists a constant $c > 0$ such that $\mathbb{E}_{P_{\mathcal{H}}}[-\nabla_\theta \ell_\theta(\theta^*)] \approx c\nabla_\theta J_p(\theta^*)$ which means these two vectors have the same direction. By Theorem 4.5,

$$\frac{\partial J_p(\theta^*)}{\partial \lambda_{\mathcal{H}}^\theta} = \frac{1}{1-\lambda_{\mathcal{H}}^\theta}\langle \nabla_\theta J_p(\theta^*), S_p \mathbb{E}_{P_{\mathcal{H}}}[-\nabla_\theta \ell_\theta(\theta^*)]\rangle = \frac{c}{1-\lambda_{\mathcal{H}}^\theta}\nabla_\theta J_p(\theta^*)^T S_p \nabla_\theta J_p(\theta^*) > 0,$$

since $S_p$ is positive definite. $\square$

## F.6. Proof of Corollary 4.7

**Corollary 4.7.** *Briefly note $\tau_p = \gamma_\theta - L_\theta \varepsilon_\theta - \frac{L_\theta \varepsilon_\theta L_\phi \varepsilon_\phi}{\gamma_\phi - L_\phi \varepsilon_\phi} \in (0, 1)$ and let $m_p$ be the minimal eigenvalue of $\frac{S_p + S_p^T}{2}$, we have $\|S_p\| \leq \frac{1}{\tau_p}$ and*

$$\text{if } |\rho_p| > \frac{1}{\sqrt{1 + m_p^2 \tau_p^2}}, \text{ then } \text{sign}(\rho_p) \cdot \frac{\partial J_p(\theta^*)}{\partial \lambda_{\mathcal{H}}^\theta} > 0.$$

*Proof.* Let $ST_p = \frac{S_p + S_p^T}{2}$, and $m_p$ be its minimal eigenvalues, respectively. First, we prove that $0 < m_p \leq \|S_p\| \leq \frac{1}{\tau_p}$ and $S_p$ is an asymmetric positive definite matrix. By Proposition 4.4, $S_p^{-1} \succeq \tau_p I$ means that for $\forall v \neq 0$, $vS_p^{-1}v^T \geq \tau_p\|v\|^2 > 0$. Notice that $vS_p \neq 0$, otherwise 0 is an eigenvalue of $S_p$ which contradicts it being invertible. Therefore, for any $v \neq 0$ we have

$$(vS_p)S_p^{-1}(vS_p)^T = (vS_p)S_p^{-T}(vS_p)^T = vS_p v^T = v\frac{S_p + S_p^T}{2}v^T = vST_p v^T \geq \tau_p\|vS_p\|^2 > 0$$

indicates that $ST_p$ is a symmetric positive definite matrix. Since $m_p$ is $ST_p$'s minimal eigenvalue, $m_p > 0$. Moreover, for any unit vector $x$,

$$m_p \leq \max_{\|x\|=1} xST_p x^T = \max_{\|x\|=1} xS_p x^T \leq \|x\|\|S_p x^T\| \leq \|S_p\|.$$

$\max_{\|x\|=1} x^T ST_p x$ equals the maximum eigenvalue of $ST_p$, so $m_p \leq \max_{\|x\|=1} x^T ST_p x$. Furthermore, for any vector $v \neq 0$,

$$\tau_p\|v\|^2 \leq vS_p^{-1}v^T \leq \|v\|\|S_p^{-1}v^T\| \Rightarrow \min_{\|v\|=1} \|S_p^{-1}v^T\| \geq \tau_p,$$

and for any vector $v \neq 0, \exists \hat{v}$ s.t. $v = \hat{v} S_p^{-1}$ and $v \in \mathbb{R}^n/\{0\} \rightarrow \hat{v} \in \mathbb{R}^n/\{0\}$ is a bijection for some $n$ since $S_p$ is invertible,

$$\|S_p\| = \max_{v \neq 0} \frac{\|vS_p\|}{\|v\|} = \max_{\hat{v} \neq 0} \frac{\|\hat{v}S_p^{-1}S_p\|}{\|\hat{v}S_p^{-1}\|} = \max_{\hat{v} \neq 0} \frac{\|\hat{v}\|}{\|\hat{v}S_p^{-1}\|} = \frac{1}{\min\limits_{\hat{v} \neq 0} \frac{\|\hat{v}S_p^{-1}\|}{\|\hat{v}\|}} = \frac{1}{\min\limits_{\|\hat{v}\|=1} \|\hat{v}S_p^{-1}\|} \leq \frac{1}{\tau_p}.$$

Next, we prove that for any vector $v \neq 0, vS_p v^T \geq m_p\|v\|^2$. Since $ST_p$ is real symmetric matrix, denote its orthogonal diagonalization as $ST_p = Q\Lambda Q^T$ and $QQ^T = Q^TQ = I$, $\Lambda = \text{diag}(\lambda_1, ..., \lambda_n)$ is the diagonal matrix composed of $ST_p$'s eigenvalues. Therefore, for any vector $v \neq 0$, let $y = vQ$ and we have

$$vS_p v^T = vST_p v^T = vQ\Lambda Q^T v^T = y\Lambda y^T = \sum_{i=1}^n \lambda_i y_i^2.$$

Notice that $\lambda_i \geq m_p$, so $vS_p v^T = \sum_{i=1}^n \lambda_i y_i^2 \geq m_p\|y\|^2 = m_p\|v\|^2$ since $Q$ is a orthogonal matrix.

Finally, we prove the main conclusion. By Theorem 4.5, we decompose vector $\mathbb{E}_{P_\mathcal{H}}[-\nabla_\theta \ell_\theta]$ into the direction of vector $\nabla_\theta J_p$ and the direction perpendicular to it, and we can get

$$\begin{aligned}
\frac{\partial J_p(\theta^*)}{\partial \lambda_\mathcal{H}^\theta} &= \frac{1}{1-\lambda_\mathcal{H}^\theta} \langle \nabla_\theta J_p(\theta^*), S_p \, \mathbb{E}_{P_\mathcal{H}}[-\nabla_\theta \ell_\theta(\theta^*)] \rangle \\
&= \frac{1}{1-\lambda_\mathcal{H}^\theta} \Big( \underbrace{\frac{\rho_p\|\mathbb{E}_{P_\mathcal{H}}[-\nabla_\theta \ell_\theta]\|}{\|\nabla_\theta J_p\|} \nabla_\theta J_p \cdot S_p \cdot \nabla_\theta J_p^T}_{(A)} + \underbrace{\sqrt{1-\rho_p^2}\|\mathbb{E}_{P_\mathcal{H}}[-\nabla_\theta \ell_\theta]\|\nabla_\theta J_p \cdot S_p \cdot w^T}_{(B)} \Big)
\end{aligned} \tag{33}$$

where $w \perp \nabla_\theta J_p$, $\|w\| = 1$. For the term $(A)$, notice that $\|S_p\| \leq \frac{1}{\tau_p}$ and for any $v \neq 0$, $vS_p v^T \geq m_p\|v\|^2$, we have

$$\begin{aligned}
\frac{\rho_p\|\mathbb{E}_{P_\mathcal{H}}[-\nabla_\theta \ell_\theta]\|}{\|\nabla_\theta J_p\|}\|\nabla_\theta J_p\|^2 m_p &\leq (A) \leq \frac{\rho_p\|\mathbb{E}_{P_\mathcal{H}}[-\nabla_\theta \ell_\theta]\|}{\|\nabla_\theta J_p\|}\|\nabla_\theta J_p\|^2 \frac{1}{\tau_p}, \text{ if } \rho_p > 0 \\
\frac{\rho_p\|\mathbb{E}_{P_\mathcal{H}}[-\nabla_\theta \ell_\theta]\|}{\|\nabla_\theta J_p\|}\|\nabla_\theta J_p\|^2 \frac{1}{\tau_p} &\leq (A) \leq \frac{\rho_p\|\mathbb{E}_{P_\mathcal{H}}[-\nabla_\theta \ell_\theta]\|}{\|\nabla_\theta J_p\|}\|\nabla_\theta J_p\|^2 m_p, \text{ if } \rho_p \leq 0
\end{aligned} \tag{34}$$

For the term $(B)$, using $\|S_p\| \leq \frac{1}{\tau_p}$, it's easy to verify that

$$-\frac{\sqrt{1-\rho_p^2}}{\tau_p}\|\mathbb{E}_{P_\mathcal{H}}[-\nabla_\theta \ell_\theta]\|\|\nabla_\theta J_p\| \leq (B) \leq \frac{\sqrt{1-\rho_p^2}}{\tau_p}\|\mathbb{E}_{P_\mathcal{H}}[-\nabla_\theta \ell_\theta]\|\|\nabla_\theta J_p\|. \tag{35}$$

Combining Eq. (34) and Eq. (35) into Eq. (33), we can get

$$\begin{aligned}
\frac{1}{1-\lambda_\mathcal{H}^\theta}\|\mathbb{E}_{P_\mathcal{H}}[\nabla_\theta \ell_\theta]\|\|\nabla_\theta J_p\|\big(\rho_p m_p - \frac{\sqrt{1-\rho_p^2}}{\tau_p}\big) &\leq \frac{\partial J_p(\theta^*)}{\partial \lambda_\mathcal{H}^\theta} \\
&\leq \frac{1}{1-\lambda_\mathcal{H}^\theta}\|\mathbb{E}_{P_\mathcal{H}}[\nabla_\theta \ell_\theta]\|\|\nabla_\theta J_p\|\big(\frac{\rho_p + \sqrt{1-\rho_p^2}}{\tau_p}\big), \text{ if } \rho_p > 0 \\
\frac{1}{1-\lambda_\mathcal{H}^\theta}\|\mathbb{E}_{P_\mathcal{H}}[\nabla_\theta \ell_\theta]\|\|\nabla_\theta J_p\|\big(\frac{\rho_p - \sqrt{1-\rho_p^2}}{\tau_p}\big) &\leq \frac{\partial J_p(\theta^*)}{\partial \lambda_\mathcal{H}^\theta} \\
&\leq \frac{1}{1-\lambda_\mathcal{H}^\theta}\|\mathbb{E}_{P_\mathcal{H}}[\nabla_\theta \ell_\theta]\|\|\nabla_\theta J_p\|\big(\rho_p m_p + \frac{\sqrt{1-\rho_p^2}}{\tau_p}\big), \text{ if } \rho_p \leq 0
\end{aligned} \tag{36}$$

Notice that $\frac{1}{1-\lambda_\mathcal{H}^\theta}\|\mathbb{E}_{P_\mathcal{H}}[\nabla_\theta \ell_\theta]\|\|\nabla_\theta J_p\| > 0$, so based on Eq. (36) when $\rho_p > 0$, if $\rho_p > \frac{1}{\sqrt{1+m_p^2\tau_p^2}}$ then $\frac{\partial J_p(\theta^*)}{\partial \lambda_\mathcal{H}^\theta} > 0$; when $\rho_p \leq 0$, if $-\rho_p > \frac{1}{\sqrt{1+m_p^2\tau_p^2}}$ then $\frac{\partial J_p(\theta^*)}{\partial \lambda_\mathcal{H}^\theta} < 0$ i.e. $\text{sign}(\rho_p)\frac{\partial J_p(\theta^*)}{\partial \lambda_\mathcal{H}^\theta} > 0$. $\qquad \square$

### F.7. Proof of Corollary E.2

**Corollary E.2.** *Briefly note $\tau_q = \gamma_\phi - L_\phi \varepsilon_\phi - \frac{L_\theta \varepsilon_\theta L_\phi \varepsilon_\phi}{\gamma_\theta - L_\theta \varepsilon_\theta} \in (0, 1)$ and let $m_q$ be the minimal eigenvalue of $\frac{S_q + S_q^T}{2}$, we have $\|S_q\| \leq \frac{1}{\tau_q}$ and if $|\rho_q| > \frac{1}{\sqrt{1 + m_p^2 \tau_p^2}}$, then $\operatorname{sign}(\rho_q) \cdot \frac{\partial J_q(\beta^*)}{\partial \lambda_{\mathcal{H}}^\theta} < 0$.*

*Remark* F.3. The proof of Corollary E.2 is the same as the proof of Corollary 4.7.

**Lemma F.4.** *(Two iteration paths' distribution distance bounds between adjacent rounds) Suppose Assumptions 3.2 and 3.3 hold, and data spaces $\mathcal{X}, \mathcal{Y}$ are bounded. There exists $K_c > 0$ is a positive such that for any $\forall (\theta_1, \phi_1), (\theta_2, \phi_2)$, any mixing weights $\lambda_{\mathcal{R}}^j, \lambda_{\mathcal{H}}^j, \lambda_{\mathcal{S}}^j, j \in \{\theta, \phi\}$, and any cross model data fractions $\lambda_\theta^\phi, \lambda_\phi^\theta$,*

$$
W(P(\theta_1, \phi_1), P(\theta_2, \phi_2)) \leq (1 - \lambda_{\mathcal{R}}^\theta)\Big[(1 - \lambda_\theta^\phi)K_c\|\theta_1 - \theta_2\| + \lambda_\theta^\phi K_c\|\phi_1 - \phi_2\|
$$
$$
+ (1 - \lambda_\theta^\phi)(K_c + 1)W(P(\theta_1^-, \phi_1^-), P(\theta_2^-, \phi_2^-)) + \lambda_\theta^\phi(K_c + 1)W(Q(\theta_1^-, \phi_1^-), Q(\theta_2^-, \phi_2^-))\Big], \quad (37)
$$

$$
W(Q(\theta_1, \phi_1), Q(\theta_2, \phi_2)) \leq (1 - \lambda_{\mathcal{R}}^\phi)\Big[(1 - \lambda_\phi^\theta)K_c\|\phi_1 - \phi_2\| + \lambda_\phi^\theta K_c\|\theta_1 - \theta_2\|
$$
$$
+ (1 - \lambda_\phi^\theta)(K_c + 1)W(Q(\theta_1^-, \phi_1^-), Q(\theta_2^-, \phi_2^-)) + \lambda_\phi^\theta(K_c + 1)W(P(\theta_1^-, \phi_1^-), P(\theta_2^-, \phi_2^-))\Big], \quad (38)
$$

*where $W$ is the Wasserstein distance and $(\theta_i^-, \phi_i^-), i \in \{1, 2\}$ are the previous round's model parameters and $(\theta_i, \phi_i)$ are obtained after updating the model parameters $(\theta_i^-, \phi_i^-)$ based on Eq. (4) for one whole round no matter the updates in this round are synchronous or asynchronous.*

*Remark* F.5. This lemma describes the sensitivity of the mixture distributions to parameters when iterating in a multi-model interaction framework. The sensitivity corresponding to the parameters obtained in subsequent iterations is determined by the differences in the parameters themselves and the distance between the mixture distributions from the previous iteration. Also, the strength of the interaction corresponds to the proportion of synthetic data, including curation data. The weaker the interaction, the smaller the upper bound of the distribution distances due to parameter differences.

*Proof.* Let $\mathcal{S}_p(\hat{\theta}, \hat{\phi}), \mathcal{S}_q(\hat{\theta}, \hat{\phi})$ be the distributions of model $\theta$'s and model $\phi$'s synthetic model-generated data with model parameters $\hat{\theta}$ and $\hat{\phi}$, respectively. Similarly we denote $\mathcal{H}_p(\hat{\theta}, \hat{\phi}), \mathcal{H}_q(\hat{\theta}, \hat{\phi})$ as the distributions of model $\theta$'s and model $\phi$'s curated synthetic data with model parameters $\hat{\theta}$ and $\hat{\phi}$, respectively. Let $\mathcal{R}_p$ and $\mathcal{R}_q$ be the fixed real data distribution of model $\theta$ and model $\phi$, respectively. By the definition of the mixture distributions, we have

$$
P(\hat{\theta}, \hat{\phi}) = \lambda_{\mathcal{S}}^\theta \mathcal{S}_p(\hat{\theta}, \hat{\phi}) + \lambda_{\mathcal{H}}^\theta \mathcal{H}_p(\hat{\theta}, \hat{\phi}) + \lambda_{\mathcal{R}}^\theta \mathcal{R}_p, \; Q(\hat{\theta}, \hat{\phi}) = \lambda_{\mathcal{S}}^\phi \mathcal{S}_q(\hat{\theta}, \hat{\phi}) + \lambda_{\mathcal{H}}^\phi \mathcal{H}_q(\hat{\theta}, \hat{\phi}) + \lambda_{\mathcal{R}}^\phi \mathcal{R}_q. \quad (39)
$$

First, we prove that for synthetic data distribution, $\forall (\theta_1, \phi_1), (\theta_2, \phi_2)$,

$$
W(\mathcal{S}_p(\theta_1, \phi_1), \mathcal{S}_p(\theta_2, \phi_2)) \leq (1 - \lambda_\theta^\phi)L\|\theta_1 - \theta_2\| + \lambda_\theta^\phi L\|\phi_1 - \phi_2\|
$$
$$
+ (1 - \lambda_\theta^\phi)(L + 1)W(P(\theta_1^-, \phi_1^-), P(\theta_2^-, \phi_2^-)) + \lambda_\theta^\phi(L + 1)W(Q(\theta_1^-, \phi_1^-), Q(\theta_2^-, \phi_2^-)), \quad (40)
$$

where $(\theta_i^-, \phi_i^-)$ are the parameters at the previous round and $L$ is the Lipschitz constant for models and reward functions. By the definition of $\mathcal{S}_p$ and Eq. (1)-(2), for any random variables $z_1 \sim \mathcal{S}_p(\theta_1, \phi_1), z_2 \sim \mathcal{S}_p(\theta_2, \phi_2)$ we can get

$$
z_1 = (1 - \lambda_\theta^\phi)(x_p^1, y_p^1) + \lambda_\theta^\phi(x_q^1, y_q^1), z_2 = (1 - \lambda_\theta^\phi)(x_p^2, y_p^2) + \lambda_\theta^\phi(x_q^2, y_q^2). \quad (41)
$$

where $x_p^1 = p(y_p^1, \theta_1), y_p^1 \sim P^y(\theta_1^-, \phi_1^-), y_q^1 = q(x_q^1, \phi_1), x_q^1 \sim Q^x(\theta_1^-, \phi_1^-)$ and $x_p^2 \sim p(y_p^2, \theta_2), y_p^2 \sim P^y(\theta_2^-, \phi_2^-), y_q^2 \sim q(x_q^2, \phi_1), x_q^2 \sim Q^x(\theta_2^-, \phi_2^-)$. Therefore,

$$
W(\mathcal{S}_p(\theta_1, \phi_1), \mathcal{S}_p(\theta_2, \phi_2)) = \inf_{\pi \in \Pi(\mathcal{S}_p(\theta_1, \phi_1), \mathcal{S}_p(\theta_2, \phi_2))} \mathbb{E}_{(\omega_1, \omega_2) \sim \pi}\|\omega_1 - \omega_2\| \leq \mathbb{E}\|z_1 - z_2\|
$$
$$
= \mathbb{E}\big\|(1 - \lambda_\theta^\phi)\big((p(y_p^1, \theta_1), y_p^1) - (p(y_p^2, \theta_2), y_p^2)\big) + \lambda_\theta^\phi\big((x_q^1, q(x_q^1, \phi_1)) - (x_q^2, q(x_q^2, \phi_2))\big)\big\|
$$
$$
\leq (1 - \lambda_\theta^\phi)\big(\mathbb{E}\|p(y_p^1, \theta_1) - p(y_p^2, \theta_2)\| + \mathbb{E}\|y_p^1 - y_p^2\|\big) + \lambda_\theta^\phi\big(\mathbb{E}\|q(x_q^1, \phi_1) - q(x_q^2, \phi_2)\| + \mathbb{E}\|x_q^1 - x_q^2\|\big)
$$
$$
\leq (1 - \lambda_\theta^\phi)\big(L\|\theta_1 - \theta_2\| + (L + 1)\mathbb{E}\|y_p^1 - y_p^2\|\big) + \lambda_\theta^\phi\big(L\|\phi_1 - \phi_2\| + (L + 1)\mathbb{E}\|x_q^1 - x_q^2\|\big). \quad (42)
$$

The last step is derived from the Lipschitz property of models $\theta, \phi$, and under $\|\|$ norm, marginal distribution's $W$ distance is less than the $W$ distance of $P$. Since Eq. (42) is satisfied for any $z_1, z_2$, taking inf on both sides and we can get Eq. (40). Similarly, we can prove the same conclusion for $\mathcal{S}_q$.

Next, we prove similar results for synthetic curation data distributions. By Eq. (3), for the model $\theta$, $(x_1, .., x_K)$ sampled independently from $p(y, \theta)$, then user picks one sample $\hat{x}$ based on the following probabilities:

$$\mathbb{P}(\hat{x} = x_k | x_1, ..., x_K, y) = \frac{e^{r(y, x_k)}}{\sum_{i=1}^{K} e^{r(y, x_i)}}, \quad \text{denote as } \hat{x} \sim \mathcal{BT}(x_1, .., x_K). \tag{43}$$

Let $K_c = LK(1 + \frac{LB}{2}) > 0$ be a constant where $L$ is the Lipschitz constant for model functions and reward, $B$ is the data norm's upper bound ($\mathcal{X}, \mathcal{Y}$ are bounded by assumption) and $K$ is the user curation sampling number. We prove that for $\forall (\theta_1, \phi_1), (\theta_2, \phi_2)$

$$\begin{aligned} W(\mathcal{H}_p(\theta_1, \phi_1), \mathcal{H}_p(\theta_2, \phi_2)) &\leq (1 - \lambda_\theta^\phi) K_c \|\theta_1 - \theta_2\| + \lambda_\theta^\phi K_c \|\phi_1 - \phi_2\| \\ &+ (1 - \lambda_\theta^\phi)(K_c + 1) W(P(\theta_1^-, \phi_1^-), P(\theta_2^-, \phi_2^-)) + \lambda_\theta^\phi (K_c + 1) W(Q(\theta_1^-, \phi_1^-), Q(\theta_2^-, \phi_2^-)), \end{aligned} \tag{44}$$

where $(\theta_i^-, \phi_i^-)$ are the parameters for previous mixture distributions. By the definition of $\mathcal{H}_p$ and combined with notations in Eq. (41), for any random variables $z_1^c \sim \mathcal{H}_p(\theta_1, \phi_1), z_2^c \sim \mathcal{H}_p(\theta_2, \phi_2)$,

$$z_1^c = (1 - \lambda_\theta^\phi)(\hat{x}_p^1, y_p^1) + \lambda_\theta^\phi (x_q^1, \hat{y}_q^1), z_2^c = (1 - \lambda_\theta^\phi)(\hat{x}_p^2, y_p^2) + \lambda_\theta^\phi (x_q^2, \hat{y}_q^2), \tag{45}$$

where $\hat{x}_p^1 \sim \mathcal{BT}(x_p^1(1), ..., x_p^1(K)), x_p^1(i) \overset{i.i.d.}{\sim} p(y_p^1, \theta_1), y_p^1 \sim P^y(\theta_1^-, \phi_1^-), \hat{y}_q^1 \sim \mathcal{BT}(y_q^1(1), ...y_q^1(K)), y_q^1(i) \overset{i.i.d.}{\sim} q(x_q^1, \phi_1), x_q^1 \sim Q^x(\theta_1^-, \phi_1^-)$ and $\hat{x}_p^2 \sim \mathcal{BT}(x_p^2(1), ..., x_p^2(K)), x_p^2(i) \overset{i.i.d.}{\sim} p(y_p^2, \theta_2), y_p^2 \sim P^y(\theta_2^-, \phi_2^-), \hat{y}_q^2 \sim \mathcal{BT}(y_q^2(1), ...y_q^2(K)), y_q^2(i) \overset{i.i.d.}{\sim} q(x_q^2, \phi_2), x_q^2 \sim Q^x(\theta_2^-, \phi_2^-)$. Therefore, similar to Eq. (42), we have

$$W(\mathcal{H}_p(\theta_1, \phi_1), \mathcal{H}_p(\theta_2, \phi_2)) \leq (1 - \lambda_\theta^\phi)(\mathbb{E}\|\hat{x}_p^1 - \hat{x}_p^2\| + \mathbb{E}\|y_p^1 - y_p^2\|) + \lambda_\theta^\phi (\mathbb{E}\|\hat{y}_q^1 - \hat{y}_q^2\| + \mathbb{E}\|x_q^1 - x_q^2\|). \tag{46}$$

For the term $\mathbb{E}\|\hat{x}_p^1 - \hat{x}_p^2\|$, by its definition,

$$\begin{aligned} \mathbb{E}\|\hat{x}_p^1 - \hat{x}_p^2\| &= \mathbb{E}\|\sum_{i=1}^{K} p_i^1 x_p^1(i) - \sum_{i=1}^{K} p_i^2 x_p^2(i)\|, \; p_i^k = \frac{e^{r(x_p^k(i))}}{\sum_{j=1}^{K} e^{r(x_p^k(j))}}, \; i \in \{1, .., K\}, k \in \{1, 2\} \\ &= \mathbb{E}\|\sum_{i=1}^{K} p_i^1(x_p^1(i) - x_p^2(i)) - \sum_{i=1}^{K} (p_i^1 - p_i^2) x_p^2(i)\| \leq \mathbb{E} \sum_{i=1}^{K} p_i^1 \|x_p^1(i) - x_p^2(i)\| + B \sum_{i=1}^{K} \mathbb{E}|p_i^1 - p_i^2| \\ &= \mathbb{E} \sum_{i=1}^{K} p_i^1 \|p(y_p^1, \theta_1) - p(y_p^2, \theta_2)\| + B \sum_{i=1}^{K} \mathbb{E}|p_i^1 - p_i^2| \leq LK(\mathbb{E}\|y_p^1 - y_p^2\| + \|\theta_1 - \theta_2\|) + B \sum_{i=1}^{K} \mathbb{E}|p_i^1 - p_i^2|. \end{aligned} \tag{47}$$

For the softmax function, it's easy to check that its 0.5-Lipschitz under the $\ell_1$ norm, and since $r$ is also Lipschitz,

$$\begin{aligned} B \sum_{i=1}^{K} \mathbb{E}|p_i^1 - p_i^2| &\leq \frac{B}{2} \sum_{i=1}^{K} \mathbb{E}|r(x_p^1(i)) - r(x_p^2(i))| \leq \frac{LB}{2} \sum_{i=1}^{K} \mathbb{E}\|x_p^1(i) - x_p^2(i)\| \\ &\leq \frac{L^2 BK}{2} (\mathbb{E}\|y_p^1 - y_p^2\| + \|\theta_1 - \theta_2\|). \end{aligned} \tag{48}$$

Combine Eq. (47) and Eq. (48), we have

$$\mathbb{E}\|\hat{x}_p^1 - \hat{x}_p^2\| \leq LK(1 + \frac{LB}{2})(\mathbb{E}\|y_p^1 - y_p^2\| + \|\theta_1 - \theta_2\|). \tag{49}$$

Similarly, we can get the upper bound of $\mathbb{E}\|\hat{y}_q^1 - \hat{y}_q^2\|$ as following

$$\mathbb{E}\|\hat{y}_q^1 - \hat{y}_q^2\| \leq LK(1 + \frac{LB}{2})(\mathbb{E}\|x_q^1 - x_q^2\| + \|\phi_1 - \phi_2\|). \tag{50}$$

Substitute Eq. (49) and Eq. (50) into Eq. (46) and notice the inequality between marginal distribution $W$ distance and $P$'s $W$ distance, we can get Eq. (44). Similarly, we can prove the same conclusion for $\mathcal{H}_q$.

Finally, we prove the lemma conclusion. Recall the definition of the mixture distribution $P$ in Eq. (39), and similarly we have for $\forall (\theta_1, \phi_1), (\theta_2, \phi_2)$,

$$
\begin{aligned}
W(P(\theta_1, \phi_1), P(\theta_2, \phi_2)) &\leq \lambda_{\mathcal{S}}^{\theta} W(\mathcal{S}_p(\theta_1, \phi_1), \mathcal{S}_p(\theta_2, \phi_2)) + \lambda_{\mathcal{H}}^{\theta} W(\mathcal{H}_p(\theta_1, \phi_1), \mathcal{H}_p(\theta_2, \phi_2)) + \lambda_{\mathcal{R}}^{\theta} W(\mathcal{R}_p, \mathcal{R}_p) \\
&= \lambda_{\mathcal{S}}^{\theta} W(\mathcal{S}_p(\theta_1, \phi_1), \mathcal{S}_p(\theta_2, \phi_2)) + \lambda_{\mathcal{H}}^{\theta} W(\mathcal{H}_p(\theta_1, \phi_1), \mathcal{H}_p(\theta_2, \phi_2)).
\end{aligned}
$$

Substitute Eq. (40) and Eq. (44) into the above equation, and we can prove Eq. (37). Similarly, Eq. (38) is also satisfied. $\quad\square$

**Corollary F.6.** *(Two iteration paths' distribution distance bounds for single model update in one round) Suppose Assumptions 3.2 and 3.3 hold, and data spaces $\mathcal{X}, \mathcal{Y}$ are bounded. $K_c$ is the positive constant in Lemma F.4. For any mixing weights $\lambda_{\mathcal{R}}^j, \lambda_{\mathcal{H}}^j, \lambda_{\mathcal{S}}^j, j \in \{\theta, \phi\}$, and any cross model data fractions $\lambda_{\theta}^{\phi}, \lambda_{\phi}^{\theta}$, and for $\forall (\theta_1, \phi_1), (\theta_2, \phi_2)$, if $(\theta_i^-, \phi_i^-), i \in \{1, 2\}$ are the previous round's model parameters before $(\theta_i, \phi_i), i \in \{1, 2\}$, then during the iteration process to get $(\theta_i, \phi_i)$, regardless of the iteration order (Eq. (21)-(23)) in the current iteration, the distribution distance in this round's process ($\hat{\theta}_i \in \{\theta_i, \theta_i^-\}, \hat{\phi}_i \in \{\phi_i, \phi_i^-\}, i \in \{1, 2\}$) has the following inequality:*

$$
\begin{aligned}
W(P(\hat{\theta}_1, \hat{\phi}_1), P(\hat{\theta}_2, \hat{\phi}_2)) &\leq (1 - \lambda_{\mathcal{R}}^{\theta}) \Big[ (1 - \lambda_{\theta}^{\phi}) K_c \|\hat{\theta}_1 - \hat{\theta}_2\| + \lambda_{\theta}^{\phi} K_c \|\hat{\phi}_1 - \hat{\phi}_2\| \\
&\quad + (1 - \lambda_{\theta}^{\phi})(K_c + 1) W(P(\theta_1^-, \phi_1^-), P(\theta_2^-, \phi_2^-)) + \lambda_{\theta}^{\phi}(K_c + 1) W(Q(\theta_1^-, \phi_1^-), Q(\theta_2^-, \phi_2^-)) \Big], \\
W(Q(\hat{\theta}_1, \hat{\phi}_1), Q(\hat{\theta}_2, \hat{\phi}_2)) &\leq (1 - \lambda_{\mathcal{R}}^{\phi}) \Big[ (1 - \lambda_{\phi}^{\theta}) K_c \|\hat{\phi}_1 - \hat{\phi}_2\| + \lambda_{\phi}^{\theta} K_c \|\hat{\theta}_1 - \hat{\theta}_2\| \\
&\quad + (1 - \lambda_{\phi}^{\theta})(K_c + 1) W(Q(\theta_1^-, \phi_1^-), Q(\theta_2^-, \phi_2^-)) + \lambda_{\phi}^{\theta}(K_c + 1) W(P(\theta_1^-, \phi_1^-), P(\theta_2^-, \phi_2^-)) \Big].
\end{aligned}
\tag{51}
$$

*Remark* F.7. The proof of Corollary F.6 is the same as the proof of Lemma F.4. The main difference is whether the model parameters $(\theta, \phi)$ in Eq. (41) and Eq. (45) are currently updated parameters $\theta_1, \theta_2, \phi_1, \phi_2$ or parameters from the previous round $\theta_1^-, \theta_2^-, \phi_1^-, \phi_2^-$.

For example, $(\theta_1^-, \phi_1^-)$ is updated to $(\theta_1, \phi_1)$ based on Eq. (21), meaning that in this round model $\theta$ is updated before model $\phi$. $(\theta_2^-, \phi_2^-)$ is updated to $(\theta_2, \phi_2)$ based on Eq. (22), meaning that in this iteration path, model $\phi$ is updated before model $\theta$ in this round. Therefore, after both iterative paths updating only one model, the mixture distributions for model $\theta$ are $P(\theta_1, \phi_1^-), P(\theta_2^-, \phi_2)$ respectively. Eq. (51) actually describes how the distribution distance $W$ of the intermediate process in each round is constrained by the previous round's distribution distances and the current parameters. We will use this corollary and Lemma F.4 to prove Proposition 3.4.

**Lemma F.8.** *If the real data is bounded and data spaces $\mathcal{X}, \mathcal{Y}$ are bounded, then for $\forall t > 0$, the mixture distributions $P(\theta_t, \phi_t)$ and $Q(\theta_t, \phi_t)$ have finite first moments.*

*Proof.* If real data is bounded which means for $Z \sim R_p$ or $Z \sim R_q$, then $\|Z\| \leq M$ for some $M > 0$ then $\mathbb{E}[\|Z\|] \leq M$ is finite, real data distributions have finite first moment. We use the induction method to prove this lemma. Assume that $P(\theta_{t-1}, \phi_{t-1})$ and $Q(\theta_{t-1}, \phi_{t-1})$ are distributions with finite first moment, then for $Z^- \sim P(\theta_{t-1}, \phi_{t-1})$, there exists $M^- > 0$ such that

$$
\mathbb{E}[\|Z^-\|] \leq M^-.
$$

Next, we prove that $P(\theta_t, \phi_t)$ and $Q(\theta_t, \phi_t)$ still have finite first moment. Take $Z \sim P(\theta_t, \phi_t)$ with synchronous updates as an example. Since $P(\theta_t, \phi_t)$ is a mixture distribution, for $y_s, y_c \overset{i.i.d.}{\sim} P^y(\theta_{t-1}, \phi_{t-1})$ and $\hat{x}_c \sim \mathcal{BT}(x_c(1), ..., x_c(K))$, $x_c(i) \overset{i.i.d.}{\sim} p(y_c, \theta_t)$, we can get

$$
\begin{aligned}
Z &= \lambda_{\mathcal{S}}^{\theta}(p(y_s, \theta_t), y_s) + \lambda_{\mathcal{H}}^{\theta}(\hat{x}_c, y_c) + \lambda_{\mathcal{R}}^{\theta} Z_r, \\
\mathbb{E}[\|Z\|] &\leq \lambda_{\mathcal{S}}^{\theta}(\mathbb{E}[\|p(y_s, \theta_t)\|] + \mathbb{E}[\|y_s\|]) + \lambda_{\mathcal{H}}^{\theta}(\mathbb{E}[\|\hat{x}_c\|] + \mathbb{E}[\|y_c\|]) + \lambda_{\mathcal{R}}^{\theta} M \\
&\leq (1 - \lambda_{\mathcal{R}}^{\theta})(B + M^-) + \lambda_{\mathcal{R}}^{\theta} M < \infty,
\end{aligned}
\tag{52}
$$

where $B$ is the upper bound of data space. Therefore, $P(\theta_t, \beta_t)$ has finite first moment for any $t > 0$. Similarly, we can prove that $Q(\theta_t, \beta_t)$ has finite first moment for any $t > 0$. If training updates are asynchronous, the proof is similar. $\quad\square$

*Remark* F.9. Our analysis of Proposition 3.4 is carried out on a compact set of the input–parameter space. The models and rewards Lipschitz property in Proposition 3.4 is natural in modern ML systems: real data is finite and evaluation is performed on a finite benchmark (hence compact), images are typically normalized to a bounded range, and we take model outputs to be Euclidean objects such as logits tensors (for LLMs) or image tensors (for vision models). In addition, common training practices (e.g., weight decay, gradient/activation clipping, spectral constraints, and other regularizers) keep the effective parameter trajectory within a bounded region. On such compact sets, neural networks and reward models built from standard primitives (affine/conv operators, residual connections, activation functions, and stabilized normalization/softmax modules) are locally Lipschitz and therefore admit uniform Lipschitz constants when restricted to the compact domain. This type of Lipschitz regularity—either assumed directly or enforced/estimated explicitly—is pervasive in the theory literature on optimization stability/generalization and on robustness/certification for deep networks (Meunier et al., 2022; Kim et al., 2021).

### F.8. Proof of Proposition 3.4

**Proposition 3.4.** *Suppose Assumptions 3.2 and 3.3 hold, and data spaces $\mathcal{X}, \mathcal{Y}$ are bounded. If both models are Lipschitz in their inputs and $\theta, \phi$ and reward functions are Lipschitz in the inputs, then there exists $\tau \in (0, 1)$ such that if the fraction of real data in each round of model training is sufficiently large, i.e., $\min\left(\lambda_{\mathcal{R}}^{\theta}, \lambda_{\mathcal{R}}^{\phi}\right) > \tau$, then a unique stable point $(\theta^*, \phi^*)$ exists. Moreover, the iterative training loop (4) will drive the multi-model ecosystem $(\theta_t, \phi_t)$ to converge to $(\theta^*, \phi^*)$, and the training data distributions will also converge, i.e., $\lim_{t\to\infty} P_t^{x,y} = P(\theta^*, \phi^*)$, $\lim_{t\to\infty} Q_t^{x,y} = Q(\theta^*, \phi^*)$.*

*Proof.* We only need to prove that there exists data ratios such that the iteration processes in Eq. (21)-(23) are always compression mappings, thus combining the Banach convergence theorem to prove this proposition.

Recall the notations $R_p, R_q$ in Eq. (16). First, we prove that for any $(\theta_1, \phi_1), (\theta_2, \phi_2)$,

$$\left\|\operatorname*{argmin}_{\theta} R_p(\theta, P(\theta_1, \phi_1)) - \operatorname*{argmin}_{\theta} R_p(\theta, P(\theta_2, \phi_2))\right\| \leq \frac{L_\theta}{\gamma_\theta} W(P(\theta_1, \phi_1), P(\theta_2, \phi_2)). \tag{53}$$

Similar to the proof in Thm. 3.6, $R(\theta, P(\theta_1, \phi_1))$ is $\gamma_\theta$-strongly convex in $\theta$ for any fixed $(\theta_1, \phi_1)$ and there always exists unique minimal points $\varphi_1, \varphi_2$ so that $\nabla_\theta R_p\left(\theta, P(\theta_1, \phi_1)\right)\big|_{\theta=\varphi_1} = \nabla_\theta R_p\left(\theta, P(\theta_2, \phi_2)\right)\big|_{\theta=\varphi_2} = 0$. Therefore, we can get the same result in Thm. 3.6,

$$\left\|(\varphi_1 - \varphi_2)^T \left(\nabla_\theta R_p\left(\varphi_1, P(\theta_1, \phi_1)\right) - \nabla_\theta R_p\left(\varphi_2, P(\theta_1, \phi_1)\right)\right)\right\| \geq \gamma_\theta \|\varphi_1 - \varphi_2\|^2.$$

Notice that $\nabla_\theta R_p\left(\theta, P(\theta_1, \phi_1)\right)\big|_{\theta=\varphi_1} = 0$, so

$$\left\|\nabla_\theta R_p\left(\varphi_2, P(\theta_1, \phi_1)\right)\right\| \geq \gamma_\theta \|\varphi_1 - \varphi_2\|. \tag{54}$$

Moreover, let $\Pi(P(\theta_1, \phi_1), P(\theta_2, \phi_2))$ be set consisting all the distributions whose marginal distributions are $P(\theta_1, \phi_1)$ and $P(\theta_2, \phi_2)$ respectively. For $\forall (Z_1, Z_2) \sim \pi \in \Pi(P(\theta_1, \phi_1), P(\theta_2, \phi_2))$,

$$\begin{aligned}
\left\|\nabla_\theta R_p\left(\varphi_2, P(\theta_1, \phi_1)\right) - \nabla_\theta R_p\left(\varphi_2, P(\theta_2, \phi_2)\right)\right\| &= \left\|\mathbb{E}_{P(\theta_1, \phi_1)} \nabla_\theta \ell_\theta(z; \varphi_2) - \mathbb{E}_{P(\theta_2, \phi_2)} \nabla_\theta \ell_\theta(z; \varphi_2)\right\| \\
&= \left\|\mathbb{E}_{(Z_1, Z_2) \sim \pi}[\nabla_\theta \ell_\theta(Z_1; \varphi_2) - \nabla_\theta \ell_\theta(Z_2; \varphi_2)]\right\| \\
&\leq L_\theta \mathbb{E}_{(Z_1, Z_2) \sim \pi} \|Z_1 - Z_2\|,
\end{aligned} \tag{55}$$

and since $\pi$ is an arbitrary element in $\Pi$, taking inf on both sides of Eq. (55) and combining Eq. (54) we have

$$\begin{aligned}
\gamma_\theta \|\varphi_1 - \varphi_2\| &\leq \left\|\nabla_\theta R_p\left(\varphi_2, P(\theta_1, \phi_1)\right)\right\| = \left\|\nabla_\theta R_p\left(\varphi_2, P(\theta_1, \phi_1)\right) - \nabla_\theta R_p\left(\varphi_2, P(\theta_2, \phi_2)\right)\right\| \\
&\leq L_\theta \inf_{\pi \in \Pi} \mathbb{E}_{(Z_1, Z_2) \sim \pi} \|Z_1 - Z_2\| = L_\theta W(P(\theta_1, \phi_1), P(\theta_2, \phi_2)).
\end{aligned} \tag{56}$$

Recall the definition of $\varphi_1, \varphi_2$, so Eq. (53) is proved. Similarly, we can prove the same thing for model $q$, that is

$$\left\|\operatorname*{argmin}_{\phi} R_q(\phi, Q(\theta_1, \phi_1)) - \operatorname*{argmin}_{\phi} R_q(\phi, Q(\theta_2, \phi_2))\right\| \leq \frac{L_\phi}{\gamma_\phi} W(Q(\theta_1, \phi_1), Q(\theta_2, \phi_2)). \tag{57}$$

Next, assume that the parameter $(\theta, \phi) \in \mathbb{R}^n$ and let $\mathcal{P}(\mathbb{R}^n)$ be the probability distribution space consisting of all the distributions with finite first moment on $\mathbb{R}^n$. All the mixture distributions $P$ and $Q$ satisfy $P, Q \in \mathcal{P}(\mathbb{R}^n)$ by

lemma F.8. It's easy to check that $(\mathcal{P}(\mathbb{R}^n), W)$ is a complete metric space with metric $W$, the Wasserstein(-1) distance. For $\forall (\theta_1, \phi_1), (\theta_2, \phi_2)$, let

$$r_i = \big((\theta_i, \phi_i), P_i, Q_i\big) \in \mathbb{R}^n \times \mathcal{P}(\mathbb{R}^n) \times \mathcal{P}(\mathbb{R}^n), i \in \{1, 2\} \tag{58}$$

be any two model parameter and data distribution pairs, and $P_i = P(\theta_i^-, \phi_i^-)$, $Q_i = Q(\theta_i^-, \phi_i^-)$. $\theta_i, \phi_i$ are updated based on the previous round parameters $\theta_i^-, \phi_i^-$ and distributions $P(\theta_i^-, \phi_i^-), Q(\theta_i^-, \phi_i^-)$. If we summarize the iterative process as $r_i^+ = \Phi(r_i)$ and $r_i^+ = ((\theta_i^+, \phi_i^+), P_i^+, Q_i^+), P_i^+ = P(\theta_i, \phi_i), Q_i^+ = Q(\theta_i, \phi_i)$ for $i \in \{1, 2\}$, then we only need to prove that $\Phi$ can be a compression mapping. Let

$$\begin{aligned}
v &= (\|(\theta_1, \phi_1) - (\theta_2, \phi_2)\|, W(P_1, P_2), W(Q_1, Q_2))^T \in \mathbb{R}^3_{\geq 0}, \\
v^+ &= (\|(\theta_1^+, \phi_1^+) - (\theta_2^+, \phi_2^+)\|, W(P_1^+, P_2^+), W(Q_1^+, Q_2^+))^T \in \mathbb{R}^3_{\geq 0}.
\end{aligned} \tag{59}$$

For convenience, we denote that **vector $a \leq b$ is equivalent to every element in $a$ is less than or equal to that in $b$.** We next prove that $\exists M \in \mathbb{R}^{3 \times 3}$ such that $v^+ \leq Mv$.

Considering three cases (Eq. (21)-(23)) in each iteration, there are total of nine cases for the two iteration paths. For simplicity, denote

$$\sigma_\theta = \lambda^\theta_{\mathcal{H}} + \lambda^\theta_{\mathcal{S}} = 1 - \lambda^\theta_{\mathcal{R}}, \; \sigma_\phi = \lambda^\phi_{\mathcal{H}} + \lambda^\phi_{\mathcal{S}} = 1 - \lambda^\phi_{\mathcal{R}}$$

as the ratios of synthetic data including synthetic curation data. Table 2 shows the nine $M$ matrices corresponding to the nine cases. Let's start with the simplest case where $(\theta_i^+, \phi_i^+) = \hat{G}_3(\theta_i, \phi_i) = (G_p(\theta_i, \phi_i), G_q(\theta_i, \phi_i))$ (Eq. (23)) for $i \in \{1, 2\}$. Notice that

$$
v^+ = \begin{bmatrix} \|(\theta_1^+, \phi_1^+) - (\theta_2^+, \phi_2^+)\| \\ W(P_1^+, P_2^+) \\ W(Q_1^+, Q_2^+) \end{bmatrix} \leq \begin{bmatrix} \frac{L_\theta}{\gamma_\theta} W(P_1^+, P_2^+) + \frac{L_\phi}{\gamma_\phi} W(Q_1^+, Q_2^+) \\ W(P_1^+, P_2^+) \\ W(Q_1^+, Q_2^+) \end{bmatrix} \quad by\ Eq.\ (53)(57)
$$

$$
\leq \begin{bmatrix} \frac{L_\theta}{\gamma_\theta} W(P_1^+, P_2^+) + \frac{L_\phi}{\gamma_\phi} W(Q_1^+, Q_2^+) \\ \sigma_\theta [K_c \|(\theta_1, \phi_1) - (\theta_2, \phi_2)\| + (1 - \lambda^\phi_\theta)(K_c + 1) W(P_1, P_2) + \lambda^\phi_\theta (K_c + 1) W(Q_1, Q_2)] \\ \sigma_\phi [K_c \|(\theta_1, \phi_1) - (\theta_2, \phi_2)\| + \lambda^\theta_\phi (K_c + 1) W(P_1, P_2) + (1 - \lambda^\theta_\phi)(K_c + 1) W(Q_1, Q_2)] \end{bmatrix} \quad by\ Lemma\ (F.4)
$$

$$
\leq M_3 v, \; M_3 = \begin{pmatrix} \frac{L_\theta}{\gamma_\theta} \sigma_\theta K_c + \frac{L_\phi}{\gamma_\phi} \sigma_\phi K_c & \frac{L_\theta \sigma_\theta (1 - \lambda^\phi_\theta)(K_c+1)}{\gamma_\theta} + \frac{L_\phi \sigma_\phi \lambda^\theta_\phi (K_c+1)}{\gamma_\phi} & \frac{L_\theta \sigma_\theta \lambda^\phi_\theta (K_c+1)}{\gamma_\theta} + \frac{L_\phi \sigma_\phi (1 - \lambda^\theta_\phi)(K_c+1)}{\gamma_\phi} \\ \sigma_\theta K_c & \sigma_\theta (1 - \lambda^\phi_\theta)(K_c + 1) & \sigma_\theta \lambda^\phi_\theta (K_c + 1) \\ \sigma_\phi K_c & \sigma_\phi \lambda^\theta_\phi (K_c + 1) & \sigma_\phi (1 - \lambda^\theta_\phi)(K_c + 1) \end{pmatrix},
\tag{60}
$$

where $K_c = LK(1 + \frac{LB}{2})$ as in Lemma F.4 and $L$ is the Lipschitz constant for models and rewards, $B$ is the data space norm upper bound and $K$ is the curation sample number.

Moreover, we prove the slightly more complex case, where both iterative paths update the same model first in this round. Take $(\theta_i^+, \phi_i^+) = \hat{G}_1(\theta_i, \phi_i) = (G_p(\theta_i, \phi_i), G_q(G_p(\theta_i, \phi_i), \phi_i))$ (Eq. (21)) for $i \in \{1, 2\}$ as the example. Similar to Eq. (60), we can get

$$
v^+ = \begin{bmatrix} \|(\theta_1^+, \phi_1^+) - (\theta_2^+, \phi_2^+)\| \\ W(P_1^+, P_2^+) \\ W(Q_1^+, Q_2^+) \end{bmatrix} \leq \begin{bmatrix} \frac{L_\theta}{\gamma_\theta} W(P_1^+, P_2^+) + \frac{L_\phi}{\gamma_\phi} W(Q(\theta_1^+, \phi_1), Q(\theta_2^+, \phi_2)) \\ W(P_1^+, P_2^+) \\ W(Q_1^+, Q_2^+) \end{bmatrix}.
\tag{61}
$$

The only difference between this vector and the vector in Eq. (60) is the first element in the RHS of Eq. (61). By the Corollary F.6,

$$
\begin{aligned}
W(Q(\theta_1^+, \phi_1), Q(\theta_2^+, \phi_2)) &\leq \sigma_\phi [K_c \|(\theta_1^+, \phi_1) - (\theta_2^+, \phi_2)\| + \lambda^\theta_\phi (K_c + 1) W(P_1, P_2) + (1 - \lambda^\theta_\phi)(K_c + 1) W(Q_1, Q_2)] \\
&\leq \sigma_\phi [K_c \|\theta_1^+ - \theta_2^+\| + K_c \|\phi_1 - \phi_2\| + \lambda^\theta_\phi (K_c + 1) W(P_1, P_2) + (1 - \lambda^\theta_\phi)(K_c + 1) W(Q_1, Q_2)] \\
&\leq \sigma_\phi [K_c \|(\theta_1^+, \phi_1^+) - (\theta_2^+, \phi_2^+)\| + K_c \|(\theta_1, \phi_1) - (\theta_2, \phi_2)\| + \lambda^\theta_\phi (K_c + 1) W(P_1, P_2) + (1 - \lambda^\theta_\phi)(K_c + 1) W(Q_1, Q_2)],
\end{aligned}
$$

*Table 2.* Compression matrices for 9 possible iteration cases. $M_3[i]$ : $i$ th row of $M_3$, $i \in \{1,2,3\}$.

| $(\theta_1^+, \phi_1^+)$ | $(\theta_2^+, \phi_2^+)$ | Compression Matrix | $(\theta_1^+, \phi_1^+)$ | $(\theta_2^+, \phi_2^+)$ | Compression Matrix |
|---|---|---|---|---|---|
| $\hat{G}_3(\theta_1,\phi_1)$ | $\hat{G}_3(\theta_2,\phi_2)$ | $M_3$ in Eq. (60) | $\hat{G}_2(\theta_1,\phi_1)$ | $\hat{G}_1(\theta_2,\phi_2)$ | $M_{21} = \begin{bmatrix} \frac{\gamma_\theta \gamma_\phi}{\gamma_\theta\gamma_\phi - \gamma_\phi\sigma_\theta L_\theta K_c - \gamma_\theta\sigma_\phi L_\phi K_c} M_3[1] \\ M_3[2] \\ M_3[3] \end{bmatrix}$ |
| $\hat{G}_2(\theta_1,\phi_1)$ | $\hat{G}_2(\theta_2,\phi_2)$ | $M_2 = \begin{bmatrix} \frac{\gamma_\theta}{\gamma_\theta - \sigma_\theta L_\theta K_c} M_3[1] \\ M_3[2] \\ M_3[3] \end{bmatrix}$ | $\hat{G}_2(\theta_1,\phi_1)$ | $\hat{G}_3(\theta_2,\phi_2)$ | $M_{23} = \begin{bmatrix} \frac{\gamma_\theta}{\gamma_\theta - \sigma_\theta L_\theta K_c} M_3[1] \\ M_3[2] \\ M_3[3] \end{bmatrix}$ |
| $\hat{G}_1(\theta_1,\phi_1)$ | $\hat{G}_1(\theta_2,\phi_2)$ | $M_1 = \begin{bmatrix} \frac{\gamma_\phi}{\gamma_\phi - \sigma_\phi L_\phi K_c} M_3[1] \\ M_3[2] \\ M_3[3] \end{bmatrix}$ | $\hat{G}_1(\theta_1,\phi_1)$ | $\hat{G}_2(\theta_2,\phi_2)$ | $M_{12} = \begin{bmatrix} \frac{\gamma_\theta \gamma_\phi}{\gamma_\theta\gamma_\phi - \gamma_\phi\sigma_\theta L_\theta K_c - \gamma_\theta\sigma_\phi L_\phi K_c} M_3[1] \\ M_3[2] \\ M_3[3] \end{bmatrix}$ |
| $\hat{G}_1(\theta_1,\phi_1)$ | $\hat{G}_3(\theta_2,\phi_2)$ | $M_{13} = \begin{bmatrix} \frac{\gamma_\phi}{\gamma_\phi - \sigma_\phi L_\phi K_c} M_3[1] \\ M_3[2] \\ M_3[3] \end{bmatrix}$ | $\hat{G}_3(\theta_1,\phi_1)$ | $\hat{G}_1(\theta_2,\phi_2)$ | $M_{31} = \begin{bmatrix} \frac{\gamma_\phi}{\gamma_\phi - \sigma_\phi L_\phi K_c} M_3[1] \\ M_3[2] \\ M_3[3] \end{bmatrix}$ |
| $\hat{G}_3(\theta_1,\phi_1)$ | $\hat{G}_2(\theta_2,\phi_2)$ | $M_{32} = \begin{bmatrix} \frac{\gamma_\theta}{\gamma_\theta - \sigma_\theta L_\theta K_c} M_3[1] \\ M_3[2] \\ M_3[3] \end{bmatrix}$ | | | |

Therefore, substitute the inequality into the first row of Eq. (61) and combine with Lemma F.4, we have

$$\|(\theta_1^+, \phi_1^+) - (\theta_2^+, \phi_2^+)\| \leq \frac{L_\theta}{\gamma_\theta} W(P_1^+, P_2^+) + \frac{L_\phi}{\gamma_\phi} W(Q(\theta_1^+, \phi_1), Q(\theta_2^+, \phi_2))$$

$$\leq \frac{L_\theta}{\gamma_\theta}\sigma_\theta\big[K_c\|(\theta_1,\phi_1) - (\theta_2,\phi_2)\| + (1-\lambda_\theta^\phi)(K_c+1)W(P_1,P_2) + \lambda_\theta^\phi(K_c+1)W(Q_1,Q_2)\big] + \frac{L_\phi}{\gamma_\phi}\sigma_\phi\big[K_c \quad (62)$$

$$\|(\theta_1^+, \phi_1^+) - (\theta_2^+, \phi_2^+)\| + K_c\|(\theta_1,\phi_1) - (\theta_2,\phi_2)\| + \lambda_\phi^\theta(K_c+1)W(P_1,P_2) + (1-\lambda_\phi^\theta)(K_c+1)W(Q_1,Q_2)\big].$$

Combine Eq. (62) with Eq. (60) and notice that if $\frac{L_\phi \sigma_\phi K_c}{\gamma_\phi} < 1$, we can get

$$v^+ \leq M_1 v, \; M_1 = \begin{bmatrix} \frac{\gamma_\phi}{\gamma_\phi - \sigma_\phi L_\phi K_c} M_3[1] \\ M_3[2] \\ M_3[3] \end{bmatrix}, \; M_3[i] : i \text{ th row of } M_3, \; i \in \{1,2,3\}. \tag{63}$$

Similarly, if $(\theta_i^+, \phi_i^+) = \hat{G}_2(\theta_i, \phi_i) = (G_p(\theta_i, G_q(\theta_i, \phi_i)), G_q(\theta_i, \phi_i))$, we can get the $M_2$ matrix that $v^+ \leq M_2 v$ in Table 2. Now we show the proof of the most complex cases, where two iterative paths $(\theta_1, \phi_1), (\theta_2, \phi_2)$ have different update order in this round. Take $(\theta_1^+, \phi_1^+) = \hat{G}_1(\theta_1, \phi_1) = (G_p(\theta_1, \phi_1), G_q(G_p(\theta_1, \phi_1), \phi_1))$ (Eq. (21)) and $(\theta_2^+, \phi_2^+) = \hat{G}_2(\theta_2, \phi_2) = (G_p(\theta_2, G_q(\theta_2, \phi_2)), G_q(\theta_2, \phi_2))$ (Eq. (22)) as the example. Similarly, in this case we can get

$$v^+ = \begin{bmatrix} \|(\theta_1^+, \phi_1^+) - (\theta_2^+, \phi_2^+)\| \\ W(P_1^+, P_2^+) \\ W(Q_1^+, Q_2^+) \end{bmatrix} \leq \begin{bmatrix} \frac{L_\theta}{\gamma_\theta} W(P_1^+, P(\theta_2, \phi_2^+)) + \frac{L_\phi}{\gamma_\phi} W(Q(\theta_1^+, \phi_1), Q_2^+) \\ W(P_1^+, P_2^+) \\ W(Q_1^+, Q_2^+) \end{bmatrix}. \tag{64}$$

Only the first row is different from the previous case proof, and by Corollary F.6 we have

$$\|(\theta_1^+,\phi_1^+) - (\theta_2^+,\phi_2^+)\| \leq \frac{L_\theta}{\gamma_\theta}W(P_1^+, P(\theta_2,\phi_2^+)) + \frac{L_\phi}{\gamma_\phi}W(Q(\theta_1^+,\phi_1), Q_2^+)$$

$$\leq \frac{L_\theta}{\gamma_\theta}\sigma_\theta\big[K_c\|(\theta_1,\phi_1) - (\theta_2,\phi_2^+)\| + (1-\lambda_\theta^\phi)(K_c+1)W(P_1,P_2) + \lambda_\theta^\phi(K_c+1)W(Q_1,Q_2)\big]$$

$$+ \frac{L_\phi}{\gamma_\phi}\sigma_\phi\big[K_c\|(\theta_1^+,\phi_1) - (\theta_2,\phi_2)\| + \lambda_\phi^\theta(K_c+1)W(P_1,P_2) + (1-\lambda_\phi^\theta)(K_c+1)W(Q_1,Q_2)\big]$$

$$\leq (\frac{\sigma_\theta L_\theta K_c}{\gamma_\theta} + \frac{\sigma_\phi L_\phi K_c}{\gamma_\phi})\|(\theta_1^+,\phi_1^+) - (\theta_2^+,\phi_2^+)\| + M_3[1]v,$$

(65)

therefore, if $\frac{\sigma_\theta L_\theta K_c}{\gamma_\theta} + \frac{\sigma_\phi L_\phi K_c}{\gamma_\phi} < 1$, then

$$v^+ \leq M_{12}v, \quad M_{12} = \begin{bmatrix} \frac{\gamma_\theta\gamma_\phi}{\gamma_\theta\gamma_\phi - \gamma_\phi\sigma_\theta L_\theta K_c - \gamma_\theta\sigma_\phi L_\phi K_c} \times M_3[1] \\ M_3[2] \\ M_3[3] \end{bmatrix}, \quad M_3[i]: i \text{ th row of } M_3, \ i \in \{1,2,3\}. \quad (66)$$

Similarly, we can estimate the compression matrices of rest cases and the results are shown in Table 2.

Notice that for compression matrix $M_3$ in Eq. (60), if $\sigma_\theta, \sigma_\phi \to 0$ which means improving the real data ratios $\lambda_\mathcal{H}^\theta, \lambda_\mathcal{H}^\phi$, then every element in $M_3$ converges to 0. Since all elements in $M_3$ are positive and are linear combinations of $\sigma_\theta, \sigma_\phi$ and constants, let

$$\tau_1 = \frac{\gamma_\theta\gamma_\phi}{(K_c+1)(L_\theta\gamma_\phi + L_\phi\gamma_\theta + 2\gamma_\theta\gamma_\phi)} \in (0,1), \quad (67)$$

and it's easy to check that if $\sigma_\theta, \sigma_\phi < \tau_1$, then

$$\|M_3\|_1 = \max_{1\leq j\leq 3}\sum_{i=1}^3 |M_3[i,j]| < 1,$$

where $\|\|_1$ is the 1-norm. For other compression matrices, from their expressions in Table 2, we can see that the coefficients before $M_3[1]$ all monotonically converge to 1 as $\sigma_\theta, \sigma_\phi \to 0$. Therefore, $\exists \tau_2 > 0$ such that if $\sigma_\theta, \sigma_\phi < \tau_2$, then

$$\max_{k\neq 3}\|M_k\|_1 < 1.$$

It's easy to check that

$$\tau_2 = \min\Big(\frac{(\gamma_\theta L_\phi + \gamma_\phi L_\theta)\frac{2K_c+1}{K_c+1} + 2\gamma_\theta\gamma_\phi}{4K_c(\gamma_\theta L_\phi + \gamma_\phi L_\theta)}, \frac{\gamma_\theta\gamma_\phi}{K_c(\gamma_\theta L_\phi + \gamma_\phi L_\theta)}, \frac{\gamma_\phi L_\theta\frac{2K_c+1}{K_c+1} + 2\gamma_\theta\gamma_\phi}{4K_c\gamma_\phi L_\theta},$$

$$\frac{\gamma_\theta}{K_cL_\theta}, \frac{\gamma_\theta L_\phi\frac{2K_c+1}{K_c+1} + 2\gamma_\theta\gamma_\phi}{4K_c\gamma_\theta L_\phi}, \frac{\gamma_\phi}{K_cL_\phi}\Big) \quad (68)$$

satisfies this condition. Let $\tau = \min(\tau_1, \tau_2) \in (0,1)$ in Eq. (67) and (68), then for any cases we have

$$\|v^+\|_1 \leq \max_k \|M_k\|_1\|v\|_1 < \|v\|_1. \quad (69)$$

Define $d^*(r_1, r_2) = \|(\theta_1,\phi_1) - (\theta_2,\phi_2)\| + W(P_1,P_2) + W(Q_1,Q_2)$ for any $r_1 = ((\theta_1,\phi_1),P_1,Q_1)$, $r_2 = ((\theta_2,\phi_2),P_2,Q_2) \in \mathbb{R}^n \times \mathcal{P}(\mathbb{R}^n) \times \mathcal{P}(\mathbb{R}^n)$. It's easy to check this is a metric defined on $\mathbb{R}^n \times \mathcal{P}(\mathbb{R}^n) \times \mathcal{P}(\mathbb{R}^n)$. By Eq. (69) and Eq. (58), notice that

$$\|v^+\|_1 = d^*(\Phi(r_1), \Phi(r_2)) = d^*(r_1^+, r_2^+) \leq \max_k \|M_k\|_1\|v\|_1 = \max_k \|M_k\|_1 d^*(r_1,r_2) < d^*(r_1,r_2).$$

We only need to prove that $\big(\mathbb{R}^n \times \mathcal{P}(\mathbb{R}^n) \times \mathcal{P}(\mathbb{R}^n), d^*\big)$ is a complete metric space, then by Banach fixed point theorem, we can prove our conclusion.

Let $\{r_n\}_{n=1}^{\infty} = \{((\theta_n, \phi_n), P_n, Q_n)\}$ be any Cauchy sequence in $(\mathbb{R}^n \times \mathcal{P}(\mathbb{R}^n) \times \mathcal{P}(\mathbb{R}^n), d^*)$, that is

$$\forall\, \varepsilon > 0,\; \exists\, N \in \mathbb{Z}^+,\; s.t.\; \forall\, n, m \geq N, d^*(r_n, r_m) < \varepsilon.$$

Thus, for any $n, m \geq N$,

$$\|(\theta_1, \phi_1) - (\theta_2, \phi_2)\| < \varepsilon,\; W(P_1, P_2) < \varepsilon,\; W(Q_1, Q_2) < \varepsilon$$

which means $\{(\theta_n, \phi_n)\}$, $\{P_n\}$ and $\{Q_n\}$ are all Cauchy sequences in their spaces. Since $(\mathbb{R}^n, \|\|)$ and $(\mathcal{P}(\mathbb{R}^n), W)$ are complete metric spaces, there exists $(\theta_*, \phi_*) \in \mathbb{R}^n$, $P_* \in \mathcal{P}(\mathbb{R}^n)$ and $Q_* \in \mathcal{P}(\mathbb{R}^n)$ such that $(\theta_n, \phi_n) \to (\theta_*, \phi_*)$, $W(P_n, P_*) \to 0$ and $W(Q_n, Q_*) \to 0$. Define $r_* = ((\theta_*, \phi_*), P_*, Q_*) \in \mathbb{R}^n \times \mathcal{P}(\mathbb{R}^n) \times \mathcal{P}(\mathbb{R}^n)$, then $d^*(r_n, r_*) \to 0$. Therefore, $(\mathbb{R}^n \times \mathcal{P}(\mathbb{R}^n) \times \mathcal{P}(\mathbb{R}^n), d^*)$ is a complete metric space.

In summary, if the real data ratios greater than $1 - \tau \in (0, 1)$, then the iterating stable point $(\theta^*, \phi^*)$ always exists and mixture distribution converges to stable distributions. By the definitions of the mixture distributions $P$ and $Q$, it's easy to check that $P_t^{x,y} \to P(\theta^*, \phi^*)$ and $Q_t^{x,y} \to Q(\theta^*, \phi^*)$ as $t \to \infty$. By Eq. (67) and (68), $\tau \in (0, 1)$ is a constant **related to $K$ value in the curation process, network architectures, datasets, loss functions' choice**. Formally,

$$\tau = \min(\tau_1, \tau_2) = \min\left(\frac{\gamma_\theta \gamma_\phi}{(K_c+1)(L_\theta \gamma_\phi + L_\phi \gamma_\theta + 2\gamma_\theta \gamma_\phi)}, \frac{\gamma_\phi}{K_c L_\phi}, \frac{\gamma_\theta}{K_c L_\theta}, \frac{\gamma_\theta \gamma_\phi}{K_c(\gamma_\theta L_\phi + \gamma_\phi L_\theta)},\right.$$
$$\left.\frac{(\gamma_\theta L_\phi + \gamma_\phi L_\theta)\frac{2K_c+1}{K_c+1} + 2\gamma_\theta \gamma_\phi}{4K_c(\gamma_\theta L_\phi + \gamma_\phi L_\theta)}, \frac{\gamma_\phi L_\theta \frac{2K_c+1}{K_c+1} + 2\gamma_\theta \gamma_\phi}{4K_c \gamma_\phi L_\theta}, \frac{\gamma_\theta L_\phi \frac{2K_c+1}{K_c+1} + 2\gamma_\theta \gamma_\phi}{4K_c \gamma_\theta L_\phi}\right).$$

$\square$

## F.9. Proof of Theorem 4.8

**Theorem 4.8.** *Under the conditions of Proposition 3.4 and Assumption 3.5, for any $\widehat{\lambda}_{\mathcal{R}}^\theta, \widehat{\lambda}_{\mathcal{R}}^\phi > \tau$ as in Proposition 3.4, we have*

$$\left| J_p\left(\theta^*\left(\widehat{\lambda}_{\mathcal{R}}^\theta\right)\right) - J_p\left(\theta^*(1)\right) \right| = \mathcal{O}\left(1 - \widehat{\lambda}_{\mathcal{R}}^\theta\right),$$
$$\left| J_q\left(\phi^*\left(\widehat{\lambda}_{\mathcal{R}}^\phi\right)\right) - J_q\left(\phi^*(1)\right) \right| = \mathcal{O}\left(1 - \widehat{\lambda}_{\mathcal{R}}^\phi\right).$$

*Proof.* First, we prove that if the real data ratios $\lambda_{\mathcal{R}}^\theta, \lambda_{\mathcal{R}}^\phi > \tau$ as in Proposition 3.4, then $J_p(\theta), J_q(\phi)$ are Lipschitz in $\theta, \phi$, respectively. Since $\lambda_{\mathcal{R}}^\theta, \lambda_{\mathcal{R}}^\phi > \tau$, the stable points $\theta^*, \phi^*$ exist. For any $\theta_1, \theta_2 \in \Theta$, let $y$ be a random variable that follows $\mathcal{D}_\theta$ distribution and $x_1 = p_{\theta_1}(x|y), x_2 = p_{\theta_2}(x|y)$ and by $J_p$'s definition in Eq.(8) and rewards' Lipschitz property, we can get

$$|J_p(\theta_1) - J_p(\theta_2)| = |\mathbb{E}[r_\theta(x_1, y) - r_\theta(x_2, y)]| \leq L\mathbb{E}[|x_1 - x_2|] = L\mathbb{E}_{y \sim \mathcal{D}_\theta}[|p_{\theta_1}(y) - p_{\theta_2}(y)|] \leq L^2\|\theta_1 - \theta_2\|,$$

where $L$ is the Lipschitz parameter for models. Therefore, $J_p$ is Lipschitz in $\theta$, and $\|\nabla_\theta J_p(\theta^*)\| \leq \sup_{\theta \in \Theta}\|\nabla_\theta J_p\| \leq L^2$. Next, similar to the proof of Proposition 4.1, we have

$$\frac{\partial \theta^*}{\partial \lambda_{\mathcal{R}}^\theta} = -S_p\left(\frac{\partial F_p}{\partial \lambda_{\mathcal{R}}^\theta} + C_p\frac{\partial F_q}{\partial \lambda_{\mathcal{R}}^\theta}\right), \frac{\partial \phi^*}{\partial \lambda_{\mathcal{R}}^\theta} = -S_q\left(C_q\frac{\partial F_p}{\partial \lambda_{\mathcal{R}}^\theta} + \frac{\partial F_q}{\partial \lambda_{\mathcal{R}}^\theta}\right).$$
$$\frac{\partial \theta^*}{\partial \lambda_{\mathcal{S}}^\theta} = -S_p\left(\frac{\partial F_p}{\partial \lambda_{\mathcal{S}}^\theta} + C_p\frac{\partial F_q}{\partial \lambda_{\mathcal{S}}^\theta}\right), \frac{\partial \phi^*}{\partial \lambda_{\mathcal{S}}^\theta} = -S_q\left(C_q\frac{\partial F_p}{\partial \lambda_{\mathcal{S}}^\theta} + \frac{\partial F_q}{\partial \lambda_{\mathcal{S}}^\theta}\right), \quad (70)$$
$$\frac{\partial \theta^*}{\partial \lambda_{\mathcal{H}}^\theta} = -S_p\left(\frac{\partial F_p}{\partial \lambda_{\mathcal{H}}^\theta} + C_p\frac{\partial F_q}{\partial \lambda_{\mathcal{H}}^\theta}\right), \frac{\partial \phi^*}{\partial \lambda_{\mathcal{H}}^\theta} = -S_q\left(C_q\frac{\partial F_p}{\partial \lambda_{\mathcal{H}}^\theta} + \frac{\partial F_q}{\partial \lambda_{\mathcal{H}}^\theta}\right).$$

Moreover, similar to the proof in Theorem 4.5 and the extension proof in Theorem E.3 (Eq. (31),(14)), if $\lambda_{\mathcal{R}}^\theta$ is changed by increasing or decreasing the sample size of real data, it's easy to check that $d\lambda_{\mathcal{S}}^\theta = -\frac{\lambda_{\mathcal{S}}^\theta}{\lambda_{\mathcal{H}}^\theta + \lambda_{\mathcal{S}}^\theta}d\lambda_{\mathcal{R}}^\theta, d\lambda_{\mathcal{H}}^\theta = -\frac{\lambda_{\mathcal{H}}^\theta}{\lambda_{\mathcal{H}}^\theta + \lambda_{\mathcal{S}}^\theta}d\lambda_{\mathcal{R}}^\theta,$

and

$$\frac{\partial F_p}{\partial \lambda_{\mathcal{R}}^{\theta}} = \frac{1}{1-\lambda_{\mathcal{R}}^{\theta}}\mathbb{E}_{\mathcal{R}^{\theta}}[\nabla_{\theta}\ell_{\theta}(\theta^*)] = \mathbb{E}_{\mathcal{R}^{\theta}}[\nabla_{\theta}\ell_{\theta}(\theta^*)] - \mathbb{E}_{P(\theta^*,\phi^*)/\mathcal{R}^{\theta}}[\nabla_{\theta}\ell_{\theta}(\theta^*)], \quad \frac{\partial F_q}{\partial \lambda_{\mathcal{R}}^{\theta}} = 0,$$

$$\frac{\partial F_p}{\partial \lambda_{\mathcal{S}}^{\theta}} = -\frac{1}{\lambda_{\mathcal{S}}^{\theta}}\mathbb{E}_{\mathcal{R}^{\theta}}[\nabla_{\theta}\ell_{\theta}(\theta^*)] = -\frac{1-\lambda_{\mathcal{R}}^{\theta}}{\lambda_{\mathcal{S}}^{\theta}}\frac{\partial F_p}{\partial \lambda_{\mathcal{R}}^{\theta}}, \quad \frac{\partial F_q}{\partial \lambda_{\mathcal{S}}^{\theta}} = 0, \tag{71}$$

$$\frac{\partial F_p}{\partial \lambda_{\mathcal{H}}^{\theta}} = -\frac{1}{\lambda_{\mathcal{H}}^{\theta}}\mathbb{E}_{\mathcal{R}^{\theta}}[\nabla_{\theta}\ell_{\theta}(\theta^*)] = -\frac{1-\lambda_{\mathcal{R}}^{\theta}}{\lambda_{\mathcal{H}}^{\theta}}\frac{\partial F_p}{\partial \lambda_{\mathcal{R}}^{\theta}}, \quad \frac{\partial F_q}{\partial \lambda_{\mathcal{H}}^{\theta}} = 0,$$

where $P(\theta^*, \phi^*)/\mathcal{R}^{\theta}$ represents the distribution of synthetic and curated synthetic data with model parameters $\theta^*$ and $\phi^*$. Notice that if $\lambda_{\mathcal{R}}^{\theta}$ is changed by other reasons, for example reducing the size of non-real datasets $\mathcal{H}^{\theta}$ and $\mathcal{S}^{\theta}$, we can prove this theorem similar to the methods in Appendix E.3.

For any $\widehat{\lambda}_{\mathcal{R}}^{\theta} > \tau$ fixed, denote the other two mixing weights as $\widehat{\lambda}_{\mathcal{S}}^{\theta}, \widehat{\lambda}_{\mathcal{H}}^{\theta}$ and $\widehat{\lambda}_{\mathcal{R}}^{\theta} + \widehat{\lambda}_{\mathcal{S}}^{\theta} + \widehat{\lambda}_{\mathcal{H}}^{\theta} = 1$. For simplicity, since different $(\lambda_{\mathcal{R}}^{\theta}, \lambda_{\mathcal{S}}^{\theta}, \lambda_{\mathcal{H}}^{\theta})$ decides different $\theta^*$, we define

$$\mathcal{J}(t) = J_p\left(\theta^*\left(1 + t(\widehat{\lambda}_{\mathcal{R}}^{\theta} - 1), t\widehat{\lambda}_{\mathcal{S}}^{\theta}, t\widehat{\lambda}_{\mathcal{H}}^{\theta}\right)\right) : [0,1] \to \mathbb{R}, \; \mathcal{J}(1) = J_p\left(\theta^*\left(\widehat{\lambda}_{\mathcal{R}}^{\theta}\right)\right), \; \mathcal{J}(0) = J_p\left(\theta^*(1)\right).$$

We ignore the dataset $D_{\theta}$ in this notation since it's fixed. $\mathcal{J}(t)$ is differentiable with respect to $t$ guaranteed by $J_p$ and its definition. By combining Eq. (70) and Eq. (71), we have that for any $\widehat{\lambda}_{\mathcal{R}}^{\theta} > \tau$ regardless of the values of $\widehat{\lambda}_{\mathcal{S}}^{\theta}, \widehat{\lambda}_{\mathcal{H}}^{\theta}$,

$$\left| J_p\left(\theta^*\left(\widehat{\lambda}_{\mathcal{R}}^{\theta}\right)\right) - J_p\left(\theta^*(1)\right) \right| = |\mathcal{J}(1) - \mathcal{J}(0)| = \left| \int_{t=0}^{1} \frac{d}{dt}\mathcal{J}(t)dt \right|$$

$$= \left| \int_{t=0}^{1} \left\langle \nabla_{\theta}J_p(\theta^*), \frac{d}{dt}\theta^*\left(1 + t(\widehat{\lambda}_{\mathcal{R}}^{\theta} - 1), t\widehat{\lambda}_{\mathcal{S}}^{\theta}, t\widehat{\lambda}_{\mathcal{H}}^{\theta}\right) \right\rangle dt \right|$$

$$= \left| \int_{t=0}^{1} \left\langle \nabla_{\theta}J_p(\theta^*), \left(\frac{\partial\theta^*}{\partial\lambda_{\mathcal{R}}^{\theta}}\left(1 + t(\widehat{\lambda}_{\mathcal{R}}^{\theta} - 1)\right), \frac{\partial\theta^*}{\partial\lambda_{\mathcal{S}}^{\theta}}\left(t\widehat{\lambda}_{\mathcal{S}}^{\theta}\right), \frac{\partial\theta^*}{\partial\lambda_{\mathcal{H}}^{\theta}}\left(t\widehat{\lambda}_{\mathcal{H}}^{\theta}\right)\right)\left(\widehat{\lambda}_{\mathcal{R}}^{\theta} - 1, \widehat{\lambda}_{\mathcal{S}}^{\theta}, \widehat{\lambda}_{\mathcal{H}}^{\theta}\right)^{T} \right\rangle dt \right|$$

$$= \left| \int_{t=0}^{1} \left\langle \nabla_{\theta}J_p(\theta^*), \left(-S_p\mathcal{F}, \frac{t(\widehat{\lambda}_{\mathcal{R}}^{\theta} - 1)}{t\widehat{\lambda}_{\mathcal{S}}^{\theta}}S_p\mathcal{F}, \frac{t(\widehat{\lambda}_{\mathcal{R}}^{\theta} - 1)}{t\widehat{\lambda}_{\mathcal{H}}^{\theta}}S_p\mathcal{F}\right)\left(\widehat{\lambda}_{\mathcal{R}}^{\theta} - 1, \widehat{\lambda}_{\mathcal{S}}^{\theta}, \widehat{\lambda}_{\mathcal{H}}^{\theta}\right)^{T} \right\rangle dt \right| \quad \text{by Eq. (70), (71)}$$

$$\left(\mathcal{F} := \frac{\partial F_p}{\partial\lambda_{\mathcal{R}}^{\theta}}\left(\lambda_{\mathcal{R}}^{\theta} = 1 + t(\widehat{\lambda}_{\mathcal{R}}^{\theta} - 1)\right) = \mathbb{E}_{\mathcal{R}^{\theta}}[\nabla_{\theta}\ell_{\theta}(\theta^*)] - \mathbb{E}_{P(\theta^*,\phi^*)/\mathcal{R}^{\theta}}[\nabla_{\theta}\ell_{\theta}(\theta^*)] \quad \text{by Eq. (71)}\right)$$

$$= 3(1 - \widehat{\lambda}_{\mathcal{R}}^{\theta})\left| \int_{t=0}^{1} \langle \nabla_{\theta}J_p(\theta^*), S_p\mathcal{F}\rangle dt \right| \le 3(1 - \widehat{\lambda}_{\mathcal{R}}^{\theta})\sup_{t\in[0,1]} |\langle \nabla_{\theta}J_p(\theta^*), S_p\mathcal{F}\rangle|$$

$$= 3(1 - \widehat{\lambda}_{\mathcal{R}}^{\theta})\sup_{t\in[0,1]} \left|\langle \nabla_{\theta}J_p(\theta^*), S_p\mathbb{E}_{\mathcal{R}^{\theta}}[\nabla_{\theta}\ell_{\theta}(\theta^*)] - \mathbb{E}_{P(\theta^*,\phi^*)/\mathcal{R}^{\theta}}[\nabla_{\theta}\ell_{\theta}(\theta^*)]\rangle\right|$$

$$\le 3(1 - \widehat{\lambda}_{\mathcal{R}}^{\theta})\sup_{t\in[0,1]} \|\nabla_{\theta}J_p(\theta^*)\|\,\|S_p\|\,\left\|\mathbb{E}_{\mathcal{R}^{\theta}}[\nabla_{\theta}\ell_{\theta}(\theta^*)] - \mathbb{E}_{P(\theta^*,\phi^*)/\mathcal{R}^{\theta}}[\nabla_{\theta}\ell_{\theta}(\theta^*)]\right\|$$

$$\le 3(1 - \widehat{\lambda}_{\mathcal{R}}^{\theta})\sup_{t\in[0,1]} L^2\,\|S_p\|\,\left\|\mathbb{E}_{\mathcal{R}^{\theta}}[\nabla_{\theta}\ell_{\theta}(\theta^*)] - \mathbb{E}_{P(\theta^*,\phi^*)/\mathcal{R}^{\theta}}[\nabla_{\theta}\ell_{\theta}(\theta^*)]\right\| \tag{72}$$

where $\langle\rangle$ denotes the inner product between two vectors. By the proof of Proposition 4.4 in Appendix F.2 and Assumptions 3.2-3.5, $S_p^{-1} \succeq (\gamma_{\theta} - L_{\theta}\varepsilon_{\theta} - \frac{L_{\theta}\varepsilon_{\theta}L_{\phi}\varepsilon_{\phi}}{\gamma_{\phi} - L_{\phi}\varepsilon_{\phi}})I \succ 0$, so $\|S_p\| \le (\gamma_{\theta} - L_{\theta}\varepsilon_{\theta} - \frac{L_{\theta}\varepsilon_{\theta}L_{\phi}\varepsilon_{\phi}}{\gamma_{\phi} - L_{\phi}\varepsilon_{\phi}})^{-1}$ (the proof of Corollary 4.7 shows the details in Appendix F.6). Also, by Assumption 3.3, $\ell_{\theta}$ is $L_{\theta}$-Lipschitz in $\theta$, so

$$\|\mathbb{E}_{\mathcal{R}^{\theta}}[\nabla_{\theta}\ell_{\theta}(\theta^*)] - \mathbb{E}_{P(\theta^*,\phi^*)/\mathcal{R}^{\theta}}[\nabla_{\theta}\ell_{\theta}(\theta^*)]\| \le \mathbb{E}_{\mathcal{R}^{\theta}}\|\nabla_{\theta}\ell_{\theta}(\theta^*)\| + \mathbb{E}_{P(\theta^*,\phi^*)/\mathcal{R}^{\theta}}\|\nabla_{\theta}\ell_{\theta}(\theta^*)\| \le 2L_{\theta}.$$

Therefore, for Eq. (72), we can get

$$\left| J_p\left(\theta^*\left(\widehat{\lambda}_{\mathcal{R}}^{\theta}\right)\right) - J_p\left(\theta^*(1)\right) \right| \le (1 - \widehat{\lambda}_{\mathcal{R}}^{\theta})\frac{6L_{\theta}L^2}{\gamma_{\theta} - L_{\theta}\varepsilon_{\theta} - \frac{L_{\theta}\varepsilon_{\theta}L_{\phi}\varepsilon_{\phi}}{\gamma_{\phi} - L_{\phi}\varepsilon_{\phi}}} = \mathcal{O}(1 - \widehat{\lambda}_{\mathcal{R}}^{\theta}). \tag{73}$$

Similarly we can prove the same result for $J_q$. □

