# OpenReview forum: "When and How Human Curation Backfires: Preference Alignment under Multi-Model Self-Consuming Loop"
_ICML.cc/2026/Conference — ICML 2026 regular_

### Official Review · Reviewer_maEX · 2026-02-27

**Soundness:** 4
**Presentation:** 3
**Significance:** 4
**Originality:** 4
**Overall Recommendation:** 6
**Confidence:** 3

**Summary:**

The paper considers the problem of what happens when multiple models interact with each other when they train on the data generated from one model to another, and specifically investigate the alignment with respect to human alignment within this framework. Thus far, only the single model setting has been considered conditioned on human alignment, and the authors build on this previous theoretical work to create a framework to analyse the multi-model setting. For the theoretical derivations, the paper considers two models that are trained on real, synthetic, and human-curated data, and importantly, iteratively train on the data from the other model. From this, the authors derive parameter-convergence properties and study precisely how changes in human-curated data influence alignment objectives, and show that in the multi-model setting, alignment is not guaranteed and might even have the opposite from the intended effect. Experiments varying the ratio of human-curated data then show the interactions both on the model itself and on the other model.

**Compliance With Llm Reviewing Policy:**

Affirmed.

**Final Justification:**

Score was originally high.

**Key Questions For Authors:**

How do you think preference domain mismatch should be tackled, and what recommendations can you give to evaluate models on this dimension?
Have you done experiments trying to quantify/observe the (effects of the) sensitivity matrix or the cross-model influence matrix in the larger experiments, besides the example provided?

**Limitations:**

Yes

**Strengths And Weaknesses:**

*Soundness (4)*
Strengths:
The technical soundness of this paper is quite astounding, I find the minimal-working two-model setup very natural and interpretable, and its easy to intuitively see how this generalizes. The design of the CIFAR10 and Gwen2.5-0.5B experiments are very well done, and in light of the preference domain mismatch, this works really well and is interpretable.
Without a doubt are the mathematical derivations well supported

Weaknesses:
There seems to be a slight mismatch between the crux of the theoretical results and what the experimental results seem to show. The results of Theorem 3.6 is reflected in the experiments (convergence fig 3), the results of Theorem 4.8 are also reflected in the experiments (fig 3), however the sensitivity matrices and the cross-model influence matrix are not mentioned in the experiments.


*Presentation (3)*
Strengths
The paper is excellently placed inside the previous and recent literature
The theoretical contributions are introduced clearly and in a structured way, the narrative and the choices and the order of assumptions and theorems is good and supported

Weaknesses
The text and math is very clear in my opinion until you ask the important question of identifying the conditions in terms of rho_p under which the fraction of human-curated data is guaranteed to improve alignment. As its not clear how corollary 4.7 answers this question by either looking at the statement or reading the text below, I think the question is important enough to warrant some more intuitive explanation, as currently its difficult to interpret
The space constraints that the manuscript suffers from, for example the whole discussion is delegated to the supplementary. Since the paper has a lot of content, the eight-page layout is quite constraining. A couple of points to improve this (besides being able to use a ninth-page in the camera ready version): (1) I think the text is already crystal clear and that Figure 1 does not add anything substantial, and maybe even adds unnecessary clutter, as the figure is difficult to read and contains a lot of elements and arrows (2) I think you can condense the assumptions to a single sentence, as strong convexity and smoothness are common assumptions (“We assume the loss is gamma-strongly convex and L-smooth, which are standard assumptions …”) as the space obtained by removing the figure and condensing the assumptions can be used to write your discussion (as the manuscript ought to be self-contained) and clarify corollary 4.7

*Significance (4)*
Strengths
I think the problem is very relevant and having a theoretical framework for this question that is so contemporary is important
The breath of the question is wide, even bordering on a philosophical discussion regarding the interactions of this nested multi-model training in the future where these problems might be further and further exacerbated
Weaknesses
I think the discussion of the paper should be in the manuscript, even if it's a condensed version, to make it clear how future research can be shaped (i.e. can the PDM be overcome? how?)


*Originality (4)*
Strengths
The work provides the insight that interactions between models which are aligned differently and train on each other's output are not as trivial to interpret as the alignment of a single model and deepens our understanding by theoretically showing how model interaction can be quantified and by counterintuitive situations might arise.
The work builds upon previous techniques for investigating the single-model scenario and the extension to the multi-model case is very natural.

---

> ### Author Rebuttal · Authors · 2026-03-31
>
> We sincerely thank the reviewer for the thorough and positive review, and for recognizing the technical strength, originality, and relevance of the paper. We are glad that the reviewer finds the two-model setup natural and interpretable, and that the CIFAR-10 and Qwen2.5-0.5B experiments are well aligned with the story of multi-model interaction and preference domain mismatch. We also appreciate the constructive suggestions on improving the presentation and better connecting the theory to the experiments, and will address these in the revision. We now address the reviewer's concerns:
>
> > Clarify Corollary 4.7.
>
> At the end of Section 4.1, we state that model $\theta$'s curation data $\mathcal{H}\_{t}^{\theta}$ includes cross-model data with different preference, which may result in $\rho\_p<0$. The reason is that $\theta$ model update direction based on curation data can be decomposed as $(1-\lambda\_{\theta}^{\phi})\mathbb{E}\_{P\_{\mathcal{H}^{\theta}}}[-\nabla\_{\theta}\ell\_{\theta}] + \lambda\_{\theta}^{\phi}\mathbb{E}\_{P\_{\mathcal{H}^{\phi}}}[-\nabla\_{\theta}\ell\_{\theta}]$. If cross-model data ratio $\lambda_{\theta}^{\phi}$ is large and $\phi$'s preference is misaligned with $\theta$'s, the curation update direction will deviate from the reward improving direction, and by Eq. (10), $\rho_p$ can be negative. This is how cross-model interaction influences $\rho_p$.
>
> Based on this, increasing curation ratio is beneficial when:
> - curation update direction is sufficiently aligned with the reward-improving direction (large $\rho_p$>0); and
> - the coupled dynamics do not distort that direction too strongly (captured by model/coupling related quantities $m_p,\tau_p$).
>
> Corollary 4.7 combines these two factors and gives sufficient conditions: if $\rho_p>0$ is large enough to overcome the distortion caused by coupling iterations ($\rho_p>\frac{1}{\sqrt{1+m^2_p \tau^2_p}}$), then sign($\rho_p$)=1 and $\frac{\partial J_p}{\partial \lambda_{\mathcal{H}}^{\theta}}>0$, meaning that improving curation data ratio will improve alignment $J_p$. Conversely, if $\rho_p<0$ and sufficiently negative ($\rho_p<-\frac{1}{\sqrt{1+m^2_p \tau^2_p}}$), increasing the curation ratio decreases $J_p$ (sign($\rho_p$)=-1 and $\frac{\partial J_p}{\partial \lambda_{\mathcal{H}}^{\theta}}<0$). We will revise it to make this interpretation explicit.
>
> > Preference domain mismatch
>
> In large-scale continuously updated systems, PDM may be difficult to fully avoid in practice: the preference of web-scale training data is often only partially observable, and some degree of cross-source mismatch is likely inevitable. The more important question is therefore not whether mismatch exists, but whether the resulting cross-model coupling is strong enough to have meaningful consequences. When it is, the most reliable mitigation is to prioritize data sources with well-understood provenance and interpretable preference properties, and to avoid poorly characterized cross-source coupling where possible. Crucially, practitioners should not interpret a stable target metric as evidence that coupling is absent: as our analysis shows, PDM can mask substantial behavioral or parametric shifts that only become visible under a different evaluation domain.
>
> For evaluation, we recommend going beyond a single alignment score and measuring coupled systems under both target-domain and shifted or cross-domain metrics, as we do in the Qwen experiment. This is a useful way to make PDM-induced coupling more detectable. Complementing this with local sensitivity and cross-influence diagnostics can further help practitioners assess whether any hidden coupling is practically significant.
>
> > Quantify/observe the sensitivity/cross-model influence matrices.
>
> These quantities are local and high-order, and estimating them in large models is closely related to the known difficulty of inverse-Hessian-based diagnostics [1]. We therefore view theory results in Section 4 primarily as a structural characterization of the coupled mechanism, while developing efficient and robust diagnostics in large models is itself an important future direction. To make this more concrete, we added a 24-dim. Gaussian experiment [2], where the RHS of Theorem 4.5 matches the closed-forem derivatives of $J_p, J_q$ ([2]B) and a dimension-wise decomposition shows that the matrices can amplify, attenuate, or even reverse curation effects ([2]E,F). These experiments provide direct mechanism-level support for the self-/cross-influence interpretation and show that the effect is empirically observable and nontrivial.
>
> We thank the reviewer again for the strong support and for the helpful suggestions.
>
> [1] Basu S., Pope P., Feizi S. Influence Functions in Deep Learning Are Fragile, 2021.
>
> [2] https://anonymous.4open.science/r/Anonymize-C240/fig.pdf

---

> > ### Author Rebuttal · Reviewer_maEX · 2026-04-02
> >
> > Makes sense. I will keep the score.

---

> > > ### Author Response · Authors · 2026-04-07
> > >
> > > Thank you for the follow-up and for the positive acknowledgment. We sincerely appreciate your thoughtful feedback, and we are glad that our clarifications addressed your concerns.

---

### Official Review · Reviewer_xzA8 · 2026-03-11

**Soundness:** 3
**Presentation:** 2
**Significance:** 3
**Originality:** 3
**Overall Recommendation:** 4
**Confidence:** 2

**Summary:**

The paper’s main contribution is a coupled-system view of curation. Instead of treating human curation as a per-model improvement knob, it studies a multi-model self-consuming loop and analyzes curation as a perturbation of a shared fixed point, where one model’s curated data can help, do nothing, or backfire once feedback through the other model is taken into account. Within that lens, the key technical idea is the decomposition of long-run curation effects into self-influence and cross-influence, which gives a concrete account of non-monotonic alignment behavior in terms of coupled sensitivities rather than isolated failure cases. A second real contribution is a sufficient stability story under enough anchoring from real data, so the curation analysis is framed around an equilibrium rather than an arbitrary training snapshot. The paper also introduces preference domain mismatch as a useful cautionary concept: strong cross-model coupling can move parameters substantially while standard in-domain alignment metrics barely change.

**Compliance With Llm Reviewing Policy:**

Affirmed.

**Final Justification:**

After reading the rebuttal, my assessment is somewhat more positive. This is reflected in my current recommendation.

**Key Questions For Authors:**

Proposition 3.4: As written, I do not think the result follows from the assumptions stated in the proposition itself. The conclusion may still be correct under stronger conditions, but the current proof appears to use stronger regularity and has at least one problematic step in how mixture distributions are handled. Please elevate those conditions to explicit assumptions or restate Proposition 3.4 as conditional on a proved sensitivity bound. This matters because it changes how much of the stability story should be read as established versus conditional.

**Limitations:**

yes

**Strengths And Weaknesses:**

The paper tackles an important question and has a clear conceptual point. The strongest part is the move from viewing curation as a one-model improvement knob to viewing it as a coupled-system perturbation with feedback. That is an interesting and moderately original extension of curated self-consuming loops to the multi-model setting.

My main concern is whether the current theory and experiments establish as much as the paper wants them to establish. I do not think the formal support currently matches the strength of the conceptual claims.

* The central lens is useful. The self-influence versus cross-influence framing gives a concrete way to think about why curation need not be monotone once models interact through recycled data. Even setting aside some of the stronger claims, that perspective is valuable.

* At a high level, the formal support is not fully there for the paper’s strongest conceptual claims. The high-level ideas are plausible, but at least two of the main theory claims appear to rely on assumptions or proof steps that are not fully stated. The paper is honest that the assumptions are strong, but that does not fix the current proof gaps.
One concrete example is Proposition 3.4. I do not think it is established from the assumptions stated in the proposition itself. The conclusion may still be plausible, but the current proof appears to rely on stronger regularity than the proposition states, and it also seems to contain at least one problematic step in how mixture distributions are handled.
* I do think the experiments support the paper’s most defensible qualitative claim (that monotonic improvement from more curation should not be assumed in a coupled setting). That said, the empirical section is more convincing as evidence that the phenomenon can occur than as evidence that the paper has validated the specific mechanism it proposes. The results are broadly consistent with the theory, but mechanism validation requires more than directional agreement. It requires showing that the central theoretical drivers are actually present, that they explain the observed behavior better than simpler alternatives, and that the conclusions are robust beyond a small set of tightly controlled settings. Here, the experiments are framed largely as stress tests or existence proofs, which is reasonable for illustrating plausibility, but it also limits how strongly they support the causal mechanism itself. So the empirical evidence supports the qualitative intuition, but it remains too limited.

* Preference domain mismatch is a useful idea and it strengthens the paper’s overall story. Still, it feels less formally established than the core convergence and local sensitivity results.

---

> ### Author Rebuttal · Authors · 2026-03-31
>
> We thank the reviewer for the constructive and detailed feedbacks. We address the reviewer’s concerns below:
>
> > Proposition 3.4.
>
> Prop. 3.4 indeed relies on a Lipschitz condition that is currently implicit: it appears in Lemma F.4 (line 1730), specifically that models $\theta, \phi$ are Lipschitz in inputs and parameters, and rewards $r_\theta, r_\phi$ are Lipschitz in inputs. The strength and implications of these conditions are discussed in Remark F.5 (lines 1734–1743). We agree this should be made explicit, and will move it from appendix to Prop. 3.4 in the revised paper.
>
> We note that all other theoretical results have their assumptions stated explicitly. The Lipschitz condition is specific to Prop. 3.4 and does not affect the other results. Specifically, Theorem 3.6, which proves convergence of the multi-model system, is independent of Prop. 3.4 and relies only on Assumptions 3.2, 3.3, and 3.5. The results in Section 4 (including Theorem 4.8) are derived under the condition that the system converges--either via Theorem 3.6 or Prop. 3.4--so the analysis of human curation impact remains fully intact.
>
> > Handling of mixture distributions.
>
> We agree that this part is currently not stated clearly enough and we will state this explicitly in Section 2 and F in the revision. If the reviewer had a particular step in mind, we would be happy to clarify it further. Our intended protocol is that synthetic samples used for training in round $t+1$ to get $(\theta_{t+1}, \phi_{t+1})$ are conditioned on the marginals of the previous-round mixture distributions. Asynchronous updating only affects which model parameters are used in the generation step: if $\theta$ is updated before $\phi$, then the training data to get $\phi_{t+1}$ uses the already updated $\theta_{t+1}$ to generate samples, but the conditioning inputs are still from the previous-round marginals. The same proof strategy extends to multiple previous rounds, which we leave for future work.
>
> > Preference domain mismatch (PDM).
>
> The main objective in this paper is to isolate the effects of cross-model interactions and preference heterogeneity on curation dynamics; PDM is introduced to illustrate how such effects can become less visible under a given evaluation domain, even when the underlying coupled dynamics remain strong. Given space constraints, we treat it as a conceptual observation rather than a formal result, and discuss it briefly in the main paper and Appendix D.2. We agree PDM deserves more systematic study. A fuller treatment of the tradeoff between synthetic-data usage and detectable coupling under mismatch is an important future direction.
>
> > Mechanism validaton.
>
> Two factors make computing mechanism results on large models difficult: sampling error and statistical error from the approximation methods to estimate high-dim. matrices. Since related matrices are local and high-order, involving secend-order information, this is closely related to the well-known difficulty of estimating inverse-Hessian-based quantities in large models. Prior work treats scalable and statistically stable estimation of such objects as a nontrivial problem [1]. Therefore, we view theory results in Section 4 primarily as a structural characterization of the coupled mechanism, while developing efficient and robust diagnostics in large models is itself an important future work. Instead, to validate the mechanism more detailed, we added a 24-dim. Gaussian experiment and results [2] show that:
>
> - RHS of Thm. 4.5 matches the the derivative obtained from the closed-form expressions of $J_p, J_q$, confirming the theorem in this analyzable regime ([2]B).
>
> - By comparing the closed-form $J_p,J_q$ ([2]G,H) and their derivatives of curation ratio ([2]C,D) with finite-sample estimates, we find that the gaps grow rapidly as sample size decreases. This shows that even in a 24-dim. setting, finite-sample noise can cause substantial estimation error; this helps explain why direct theorem-level diagnosis in large models needs future work.
>
> - When the theory predicts a positive derivative, the $J_{p/q}$ indeed increases under small curation perturbations ([2]G,H).
>
> - By computing dimension contributions of the key inner products in Thm. 4.5, we show that $S_p$ and $S_qC_q$ can amplify, attenuate, or even reverse influences in different dimensions, and the dominant dimensions before and after projection can differ substantially ([2]E,F). This provides a mechanism-level explanation for why $\rho_p>0$ does not guarantee a positive curation effect.
>
> Overall, these experiments do not claim full large-scale theorem-level validation, but they do provide direct evidence that the self-/cross-influence mechanism is mathematically correct, empirically observable, and genuinely nontrivial. We will include the above experiment in the revision.
>
> [1] Basu S., Pope P., Feizi S. Influence Functions in Deep Learning Are Fragile, 2021.
>
> [2] https://anonymous.4open.science/r/Anonymize-C240/fig.pdf

---

> > ### Author Rebuttal · Reviewer_xzA8 · 2026-03-31
> >
> > Thank you for the detailed rebuttal. The clarification on the missing assumptions behind Proposition 3.4 substantially addresses my main technical concern. I also appreciate the other clarifications. I still view some of these points as only partially resolved in the current submission, but the rebuttal improves my overall assessment

---

> > > ### Author Response · Authors · 2026-04-07
> > >
> > > Thank you for the follow-up and for the updated assessment. We are glad that the clarification on the assumptions behind Proposition 3.4 addressed the main technical concern.
> > >
> > > In the revision we will further improve the presentation of the mixture-distribution modeling by moving the formulations currently in the appendix (like Eq. 41) into Section 2, and refer to Algorithm 1 (line 770-800) in the parts related to the sampling and retraining processes. We hope these will make the modeling protocol and the logical dependency of the later results substantially clearer.

---

### Official Review · Reviewer_jaJH · 2026-03-12

**Soundness:** 3
**Presentation:** 3
**Significance:** 2
**Originality:** 3
**Overall Recommendation:** 4
**Confidence:** 4

**Summary:**

Prior work on model collapse examine models trained iteratively on a single prior model's output, and how human preference optimization affects this process. This oversimplifies the real-world setting, where multiple models may participate in the iteration and multiple preference curations may be applied. This paper provides theoretical understanding in this direction, establishing conditions under which self-influence and cross-influence can hurt alignment.

**Compliance With Llm Reviewing Policy:**

Affirmed.

**Final Justification:**

The rebuttal addressed most of my concerns. I continue to recommend acceptance.

**Key Questions For Authors:**

Can any of the key conditions identified in the paper be measured or estimated in practice?

**Limitations:**

Yes

**Strengths And Weaknesses:**

## Strength:

1. The paper provides new theoretical results on an important and practically relevant setup that has been oversimplified by prior work.

2. The theoretical framework is clean.

3. The paper provide surprising findings to the community (eg Example 4.6).

## Weakness:

1. The CIFAR-10 experiment design feels overly artificial, particularly the use of hue-conflicting rewards. Can the authors provide a more naturally occurring setup, even one involving a mild but realistic preference conflict?

2. The analysis focuses primarily on stable fixed points and local behavior at equilibrium. However, Figure 3 suggests that the transient dynamics can be quite diverse. How might alignment evolve along the trajectory?

3. The theory relies on a strong convexity assumption in Assumption 3.2. The reviewer appreciates the discussion in Section B acknowledging that relaxing this assumption is challenging in theory, but would like to see further discussion or experiments exploring what happens when this assumption is violated.

---

> ### Author Rebuttal · Authors · 2026-03-31
>
> We thank the reviewer for the detailed feedback and positive assessment of our work. We appreciate the reviewer’s recognition that the paper studies an important multi-model setting that is largely oversimplified in prior work, and that the theoretical framework is clean. We address the reviewer's concerns below:
>
> > The CIFAR-10 setup and a more natural preference conflict.
>
> As discussed in Appendix D.1.2, we adopt hue-based rewards which cleanly exposes preference conflict between coupled models; using a hue-derived reward is an instantiation of the color-aware reward shaping paradigm established in prior works [1,2]. Our intent is not to claim that hue conflict is itself the canonical real-world preference, but to use a controlled test that cleanly isolates the mechanism predicted by the theory: in a coupled self-consuming system, increasing curation need not monotonically improve long-run alignment.
>
> We note that our theoretical analysis does not depend on a specific preference type; what matters is the induced self-/cross-influence structure. We agree that naturally occurring and milder preference conflicts are important, which is why we also included the Qwen summarization/paraphrasing experiment, a more task-grounded and realistic setting. Broader large-scale experiments with more realistic preference conflicts would be valuable future work; however, in our multi-round, multi-model setup, such experiments are computationally expensive as each round requires large-scale synthetic data generation followed by model retraining, and multiple rounds are needed until convergence.
>
> > How alignment evolves along the trajectory.
>
> This is an interesting point. As shown in Figure 3 and Section D.1.3 (line 906-925), the trajectory dynamics can be rich and non-monotonic, with hue shifting and oscillating across rounds. We agree that studying alignment evolution along the trajectory is important; in fact, we already included a discussion in Section B (**Preference alignment along the training trajectory**), where we describe how our theory framework can be extended to trajectory-level alignment, with metrics $J_p$ and $J_q$ depending jointly on the evolving model parameters. In this paper, we focus on stable points because at equilibrium, $J_p$ depends directly on $\theta^\*$ by Eq. (8), while model coupling affects it indirectly through the equilibrium $\theta^*$. This let us isolate the long-run impact of model coupling more cleanly. Along the trajectory, $J_p$ depends directly on both $\theta$ and $\phi$, making the analysis more complex; formally studying this is an interesting direction for future work.
>
> > Strong convexity assumption.
>
> This assumption is only used for deriving theorems and does not necessarily hold in experiments. For example, in the CIFAR-10 experiments, the models are trained with a noise-prediction MSE objective, which is not strongly convex in the parameters. For the Qwen experiments, we use standard token-level cross-entropy loss, which also violates the strong convexity condition.
>
> Even for theory, there is a possibility to relax this assumption. Mofakhami et al. [3] show that strong convexity is not the only route to stability guarantees under performative prediction---an optimization framework where the model's parameters influence the training data distribution, creating a feedback loop closely related to ours. Extending such analyses to coupled multi-model self-consuming systems remains an interesting direction for future work.
>
> > Whether the key quantities can be measured or estimated in practice.
>
> In small-scale settings, these quantities can be measured with reasonable fidelity, directly compared against exact calculations and used for mechanism validation in [5]. For large models, however, precise estimation is much harder, like in prior work [3, 4]: both the assumption-level quantities and the Section 4 matrices are affected by sampling noise, finite-sample statistical error, and approximation error from large-scale high-order matrix estimation. As a result, for large models these quantities are currently more useful as qualitative/local diagnostics than as precise numerical predictors. The development of accurate and scalable estimators for large models is also an important future research direction [4]. The main contribution of our paper is the structural characterization of the coupled mechanism and the identification of the quantities that govern self-influence and cross-influence.
>
> [1] Zhang R., Guo L., Huang S., Wen B. ReLLIE: Deep Reinforcement Learning for Customized Low-Light Image Enhancement, 2021.
>
> [2] Park J., Lee J.-Y., Yoo D., Kweon I.S. Distort-and-Recover: Color Enhancement Using Deep Reinforcement Learning, 2018.
>
> [3] Mofakhami M., Mitliagkas I., Gidel G. Performative Prediction with Neural Networks, 2023.
>
> [4] Basu S., Pope P., Feizi S. Influence Functions in Deep Learning Are Fragile, 2021.
>
> [5] https://anonymous.4open.science/r/Anonymize-C240/fig.pdf

---

> > ### Author Rebuttal · Reviewer_jaJH · 2026-04-04
> >
> > I thank the authors for the rebuttal and encourage them to incorporate all the discussed points into the final version.
> >
> > Regarding the hue-based rewards: I understand this serves as an empirical result supporting the theory. On the empirical side, could the authors provide more real-world examples of where this problem arises in practice, with more natural preference differences? More discussion here would help the community identify real-world cases that fall under this scenario.

---

> > > ### Author Response · Authors · 2026-04-07
> > >
> > > Thank you for the helpful follow-up. We agree that adding more real-world examples would strengthen the practical motivation. Below we provide several examples, some of which further elaborate the examples (line 10-20) in Section 1.
> > >
> > > A first example is modern alignment and safety pipelines, where preference signals are explicitly heterogeneous. Safe RLHF [1] formulates a tension between helpfulness and harmlessness, while Safety-Tuned LLaMAs [2] shows that emphasizing helpfulness alone can produce unsafe behavior, whereas stronger safety tuning can induce exaggerated refusals on benign inputs. This is precisely the kind of naturally occurring preference conflict that motivates our multi-model view: different models in the pipeline have different preferences (helpfulness/harmlessness), and their interaction can shape the final model behavior in nontrivial ways.
> > >
> > > A second example is instruction-tuning/model-distillation pipelines such as Self-Instruct [3] and Alpaca [4], where one language model generates instruction-response data, the outputs are filtered/curated, and another model is then fine-tuned on the resulting synthetic data. In such settings, the relevant preference differences are not artificial: they can correspond to natural trade-offs such as concise vs. detailed responses, style/formality preferences, or general helpfulness vs. safety-oriented behavior.
> > >
> > > A third example is web-scale multimodal data construction. CapsFusion [5] uses LLMs to consolidate and refine web image-text pairs and image-only data into synthetic captions for future multimodal model training, while datasets such as JourneyDB [6] further illustrate that large-scale generated image–text ecosystems already involve content- and style-sensitive curation. In such settings, natural preference differences can arise between literal captioning vs. richer descriptive captioning, style fidelity vs. semantic coverage, or aesthetic preference vs. downstream task utility.
> > >
> > > Another realistic case is AI assisted news production in a politically polarized media ecosystem. Different outlets serve audiences with systematically different political leanings (right/liberal), while major news organizations such as AP and Reuters publicly state that generative AI is already used in news workflows, including summaries, headlines, writing, editing, and publishing [7,8]. In such an environment, different outlets may naturally prefer different political framings of the same event; for example, emphasizing border security and law-and-order versus civil rights and inclusion. If AI tools are used to draft, summarize, rewrite, or prioritize content under these different editorial preferences, then even mild framing differences can affect which narrative preferences are preserved, amplified, and reused downstream. This concern is also consistent with prior work [9], which shows that recursive synthetic training can amplify political bias across generations.
> > >
> > > We will add these discussions in the revision to clarify that the paper is not only about a stylized toy conflict, but about a broader and practically relevant class of synthetic-data ecosystems in which multiple models interact through generated and curated data, and where natural preference differences can propagate through the loop.
> > >
> > > [1] Dai J., Pan X., Sun R., Ji J., Xu X., Liu M., Wang Y., Yang Y., Safe RLHF: Safe Reinforcement Learning from Human Feedback, 2023.
> > >
> > > [2] Bianchi F., Suzgun M., Attanasio G., Röttger P., Jurafsky D., Hashimoto T., Zou J. Safety-Tuned LLaMAs: Lessons From Improving the Safety of Large Language Models that Follow Instructions, 2023.
> > >
> > > [3] Wang Y., Kordi Y., Mishra S., Liu A., Smith N.A., Khashabi D., Hajishirzi H. Self-instruct: Aligning Language Models with Self-Generated Instructions, 2023.
> > >
> > > [4] Taori R., Gulrajani I., Zhang T., Dubois Y., Li X., Guestrin C., Liang P., Hashimoto T.B. Alpaca: A Strong, Replicable Instruction-Following Model, 2023.
> > >
> > > [5] Yu Q., Sun Q., Zhang X., Cui Y., Zhang F., Cao Y., Wang X., Liu J. CapsFusion: Rethinking Image-Text Data at Scale, 2024.
> > >
> > > [6] Sun K., Pan J., Ge Y., Li H., Duan H., Wu X., Zhang R., Zhou A., Qin Z., Wang Y., Dai J., Qiao Y., Wang L., Li H. JourneyDB: A Benchmark for Generative Image Understanding, 2023.
> > >
> > > [7] https://www.ap.org/the-definitive-source/behind-the-news/updates-to-generative-ai-standards
> > >
> > > [8] https://www.reuters.com/info-pages/reuters-and-ai
> > >
> > > [9] Wang Z., Wu Z., Zhang Y., Guan X., Jain N., Lu Q., Gupta S., Koshiyama A. Bias Amplification: Large Language Models as Increasingly Biased Media, 2025.

---

### Official Review · Reviewer_Bvew · 2026-03-13

**Soundness:** 3
**Presentation:** 4
**Significance:** 2
**Originality:** 3
**Overall Recommendation:** 4
**Confidence:** 2

**Summary:**

This paper theoretically analyzes self-consuming training involving interactions among multiple models. Based on the authors' assumptions and model, the system converges to a stable point, and human curation does not always improve model alignment. The proofs in the paper appear to be solid, but I cannot verify all the details.

**Compliance With Llm Reviewing Policy:**

Affirmed.

**Final Justification:**

I appreciate the clarifications provided. But due to my research background, it is challenging for me to accurately judge the value of the paper's details, so I will keep my original score.

**Key Questions For Authors:**

1. Would incorporating probability distributions into the model change the conclusions of the paper?

2. What assumptions do the authors make about the synthetic data, such as its quality and distribution?

3. Are any of them unexpected results?

**Limitations:**

The authors are encouraged to examine the impact of randomness more carefully.

**Strengths And Weaknesses:**

**Strengths**

1. The authors provide clear definitions for the model construction and offer rigorous explanations in the subsequent analysis. The theoretical framework of the article is also relatively clear, which is is supported by illustrative figures.

2. Model collapse and alignment discussed by the authors are important in both research and practice, and the framework in which different models interact, as considered by the authors, is also novel.

**Weaknesses**

1. There remains a gap between the setting assumed by the authors and real-world conditions. For instance, LLM outputs should follow a probability distribution that satisfies certain constraints rather than being deterministic. Thus, in theory, the system may converge to a stable distribution (or possibly something else) rather than a fixed point.

2. While it may be challenging to establish this theoretically when more factors are considered, the authors can support their arguments with further experiments. For example, the quality of the synthetic data could also affect the conclusions, and the authors do not seem to have taken this into account.

---

> ### Author Rebuttal · Authors · 2026-03-31
>
> We thank the reviewer for the constructive comments. We appreciate the recognition of our theoretical framework and the contributions to model collapse and alignment. We now address the reviewer's concerns:
>
> >  Probability distributions of LLM outputs
>
> Our framework is already distributional rather than deterministic: each model is defined via a conditional distribution (e.g., Eqs. (1)–(2)), and the retraining dynamics are defined on the induced mixture distributions. The "stable point" in our paper refers to a fixed point of the parameter update map with stationary induced training distributions, not to deterministic outputs for a fixed input.
>
> > Assumptions on synthetic data
>
> Our framework makes minimal assumptions on synthetic data: the only requirement is that the input/output data types of $\theta_t$ and $\phi_t$ are compatible (lines 69–70), and that synthetic data follows the models' output distributions, entering training through the mixture distributions $P(\theta,\phi)$ and $Q(\theta,\phi)$. The framework otherwise supports arbitrary data distributions.
>
> Regarding quality assumptions: to isolate the effect of cross-model interactions on the dynamics, our theoretical analysis assumes optimal model updates and abstracts away sampling noise, finite-sample error, and stochastic optimization noise. This is a deliberate modeling choice that allows us to precisely understand the coupling mechanism. We have discussed in Appendix B.1 that this population-level analysis does not yet quantify robustness under these additional sources of randomness; however, our experiments already operate in the finite-sample, stochastic-training regime and exhibit the same qualitative trend analyzed in theory, suggesting the theoretical insights transfer to practice. We will also clarify this theory–experiment distinction more explicitly.
>
> Beyond these practical considerations, we also think explicitly studying how synthetic-data quality interacts with the theory is an interesting direction for future work. There is already a natural connection in our framework: model quality affects the synthetic data it generates, and model-related quantities also determine the real-data threshold $\tau$ in Proposition 3.4 (lines 2113–2116) that guarantees convergence. This provides an indirect but theoretically meaningful link between synthetic-data quality and stability: if a stronger model generates higher-quality synthetic data, this can naturally affect how much synthetic data can be incorporated before the retraining dynamics risk losing stability.

---

> > ### Author Rebuttal · Reviewer_Bvew · 2026-04-04
> >
> > I appreciate the authors' reply.
> >
> > I would like to point out that this paper does not fall within the main research area. I suggest that the authors exercise greater caution when selecting the Primary Area and Keywords for similar work in future submissions.
> >
> > Thus, my main concern remains the connection between the framework presented in the paper and real-world scenarios. I still feel that the paper relies on strong assumptions, for instance, have the authors considered scenarios with multiple fixed points?

---

> > > ### Author Response · Authors · 2026-04-07
> > >
> > > We thank the reviewer for the helpful follow-up.
> > >
> > > We chose *Social Aspects -> Alignment* because the paper studies the long-run effect of human curation, a common alignment mechanism, on model behavior under cross-model interactions. Our goal is to emphasize the question related to alignment; namely, when human curation intended to improve alignment may instead have unintended long-run effects in coupled multi-model systems. In that sense, we viewed the work as closely connected to alignment, and to how human preference data shapes iterative training dynamics. That said, we will be more careful about area and keyword selection for similar work in future submissions.
> > >
> > > Regarding multiple fixed points: we have in fact thought about this case. In a non-contractive setting, the equilibrium may depend on initialization, and practical stochastic effects such as sampling noise, finite-sample error, or stochastic optimization noise. Characterizing such richer regimes would require additional analysis beyond the scope of the present paper. This is also consistent with prior works [1,2] on self-consuming / iterative retraining dynamics, which typically focus on controlled regimes that enable a clean stability analysis rather than attempting to cover all non-contractive or multi-equilibrium settings at once.
> > >
> > > In our case, we deliberately focus on a **stability regime** in which the induced distribution sensitivity is sufficiently controlled (equivalently, real-data anchoring is sufficiently strong), so that the coupled retraining dynamic has a **unique stable point**. Thus, multiple fixed points are not ruled out in general; they are ruled out only under the assumptions of Proposition 3.4 / Theorem 3.6. Our point is not that all real-world systems satisfy these assumptions, but rather that even in a clean regime already favorable to convergence, cross-model coupling can still produce the non-monotonic long-run effects of curation studied in this paper.
> > >
> > > We also note that our CIFAR10 and Qwen experiments **do not satisfy** the assumptions of the theory: they involve finite-sample effects, sampling randomness, and optimization noise. Nevertheless, they exhibit the same **qualitative phenomena** predicted by the analysis. In addition, our experiments [3] further support this point: they allow a clearer comparison between noiseless theoretical quantities and their empirical noisy counterparts, and show that the practical finite-sample results remain closely aligned with the theoretical prediction at the qualitative level ([3]C/D, [3]G/H).
> > >
> > > Due to space limitations and, more importantly, to isolate the main effect studied in this paper that cross-model coupling and how human curation can have non-monotonic long-run consequences, we leave the analysis of multiple fixed points, initialization dependence, and noise-induced equilibrium selection to the discussion and future work, rather than making them core ingredients of the present paper.
> > >
> > > [1] Ferbach D., Bertrand Q., Bose A.J., Gidel G., Self-Consuming Generative Models with Curated Data Provably Optimize Human Preferences, 2024.
> > >
> > > [2] Bertrand Q., Bose A.J., Duplessis A., Jiralerspong M., Gidel G., On The Stability of Iterative Retraining of Generative Models on Their Own Data, 2024.
> > >
> > > [3] https://anonymous.4open.science/r/Anonymize-C240/fig.pdf

---

### Decision · Program_Chairs · 2026-04-30

**Decision:**

Accept (regular)

**Comment:**

This paper studies the dynamics of training generative models on a mix of human-curated and self-generated data, focusing on a setting where multiple models exist and affect each other. The paper proposes a simplified model of such interactions, and presents a comprehensive theoretical analysis of their evolution.

All reviewers appreciated the importance of the task and the significance of the results. The proposed model is simple yet natural, and the theoretical results on convergence and its implications should be of interest to the community. While most design choices are reasonable in the context of the paper and as a starting point for initial results, reviewers raised some concerns regarding their realism and whether analysis in the paper's simplified setting and under their (presumably strong) assumption support the authors' conceptual claims. Some reviewers also found that the experiments could have been more convincing, and designed differently to support the more general claims; e.g., by using them to stress-test the model under violation of its assumptions.

Nonetheless, this is a strong paper that makes is likely to make an impact in its domain. The authors are encouraged to incorporate the reviewers' feedback, which will likely serve to strengthen the paper even further.